# Neural Processes with Stochastic Attention: Paying more attention to the context dataset

**Mingyu Kim, Kyeongryeol Go, Se-Young Yun**
KAIST AI
`{callingu, kyeongryeol.go, yunseyoung}@kaist.ac.kr`

## Abstract

Neural processes (NPs) aim to stochastically complete unseen data points based on a given context dataset. NPs essentially leverage a given dataset as a context representation to derive a suitable identifier for a novel task. To improve the prediction accuracy, many variants of NPs have investigated context embedding approaches that generally design novel network architectures and aggregation functions satisfying permutation invariant. In this work, we propose a stochastic attention mechanism for NPs to capture appropriate context information. From the perspective of information theory, we demonstrate that the proposed method encourages context embedding to be differentiated from a target dataset, allowing NPs to consider features in a target dataset and context embedding independently. We observe that the proposed method can appropriately capture context embedding even under noisy data sets and restricted task distributions, where typical NPs suffer from a lack of context embeddings. We empirically show that our approach substantially outperforms conventional NPs in various domains through 1D regression, predator-prey model, and image completion. Moreover, the proposed method is also validated by MovieLens-10k dataset, a real-world problem.

## 1 Introduction

Neural processes (NPs) have been in the spotlight as they stochastically complete unseen target points considering a given context dataset without huge inference computation (Garnelo et al., 2018a;b; Kim et al., 2019). NPs leverage neural networks to derive an identifier suitable for a novel task using context representation, which contains information about given context data points. These methods enable us to handle considerable amounts of data points, such as in image-based applications, that the Gaussian process cannot naively deal with. Many studies have revealed that the prediction performance relies on the way of context representation (Kim et al., 2019; Volpp et al., 2021). The variants of NPs have mainly investigated on context embedding approaches that generally design novel network architectures and aggregation functions that are permutation invariant. For example, Garnelo et al. (2018a;b) developed NPs using MLP layers to embed context set via mean aggregation. Bayesian aggregation satisfying permutation invariance enhanced the prediction accuracy by varying weights for individual context data points (Volpp et al., 2021). From the prospective of network architectures to increase model capacity, Kim et al. (2019) suggested the way of local representation to consider relations between context dataset and target dataset using deterministic attention mechanism (Vaswani et al., 2017). For robustness under noisy situations, bootstrapping method was proposed to orthogonally apply to variants of NPs (Lee et al., 2020).

Despite many appealing approaches, one significant drawback of previous NPs is that they still underfit when confronted with noisy situations like real-world problems. This manifests as inaccurate predictions at the locations of the context set as seen in Figure 1a. Additionally, in Figure 1b, the attentive neural process (Kim et al., 2019) fails to capture contextual embeddings because the attention weights of all target points highlight on the lowest value or the maximum value in the context dataset. In the case of the Bootstrapping ANP (Lee et al., 2020) and ANP with information dropout, the quality of heat-map is slightly improved, but it still falls short of ours. This indicates that the present NPs are unable to properly exploit context embeddings because the noisy situations impair the learning of the context embeddings during meta-training.

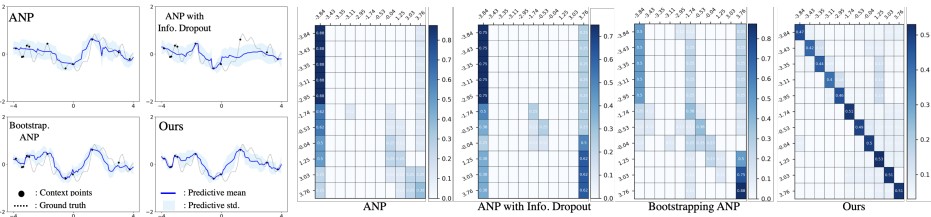

(a) 1D regression predcitions      (b) Heatmaps of asscociated attention weights

Figure 1: Comparison of 1D regression predictions and asscociated attention weights as specified by ANP (Kim et al., 2019), ANP with Information dropout, Bootstrapping ANP (Lee et al., 2020) and ours. The training data for 1D regression is fairly noisy. (a) Ours more accurately captures the context datasets and significantly better predicts than baselines. (b) The horizontal axis indicates the value of features in the context dataset, while the vertical axis indicates the value of features in the target dataset in these heatmaps. The best pattern for this heat-map is diagonal because all feature values are arranged in ascending order. Among all models, ours comes closet to the ideal. The detailed analysis is shown in Appendix G

To address this issue, we propose a newly designed neural process to fundamentally improve performance by paying more attention to the context dataset. The proposed method expedites a stochastic attention to adequately capture the dependency of the context and target datasets by adjusting stochasticity. It results in improving prediction performance due to maintained context information. We observe that our proposed algorithm works well by utilizing contextual information in an intended manner as seen in Figure 1. This method outperforms current NPs and their regularization methods in all experiment settings. Thus, this paper clarifies the proposed method as regularized NPs in terms of information theory, explaining that the stochastic attention encourages the context embedding to be differentiated from the target dataset. This differentiated information induces NPs to become appropriate identifiers for target dataset by paying more attention to context representation. To summarize, we make the following contributions:

- We propose the novel neural process that pay more attention to the context dataset. We claims for the first time, using the information theory framework, that critical conditions for contextual embeddings in NPs are independent of target features and close to contextual datasets.

- Through comprehensive analyses, we illustrate how stochastic local embeddings are crucial for NPs to focus on capturing the dependencies of context and target datasets. Even when context dataset contains noise or is somewhat different from target dataset, as shown in Lee et al. (2020), the proposed method is capable of adapting a novel task while preserving predictive performance. Particularly, this method significantly enhances performance without additional architectures and data augmentation compared to the attentive neural process (Kim et al., 2019).

- The experimental results show that the proposed model substantially outperforms conventional NPs in typical meta-regression problems. For instance, the proposed method achieves to obtain the state of the art score in the image completion task with the CelebA dataset. Especially, the proposed model maintains performance in the limited task distribution regimes such as the MovieLenz-10k dataset with a small number of users.

This paper is organized as follows. We introduce background knowledge in section 2. section 3 presents a neural process with stochastic attention and related works illustrated in section 5. Experimental results are shown in section 4, concluding with section 6.

## 2 BACKGROUND

### 2.1 NEURAL PROCESSES

Suppose that we have an observation set $X = \{x_i\}_{i=1}^n$ and a label set $Y = \{y_i\}_{i=1}^n$. NPs (Garnelo et al., 2018a;b; Kim et al., 2019) are designed to obtain the probabilistic mapping from the observation set to the label set $p(Y|X, X_c, Y_c)$ given a small subset $(X_c, Y_c) = (x_j, y_j)_{j=1}^m$. Basically, it is built

upon the neural network with an encoder-decoder architecture where the encoder $f_\phi$ outputs a task representation by feed-forwarding $(X_c, Y_c)$ through permutation-invariant set encoding (Zaheer et al., 2017; Edwards & Storkey, 2017) and the decoder $f_\theta$ models the distribution of $Y$ (e.g Gaussian case : estimating $\mu$, $\sigma$) using $X$ along with the encoder outputs. Its objective is to maximize the log likelihood over the (unknown) task distribution $p(\mathcal{T})$. All tasks provided by the data generating process are considered as Monte Carlo samples, respectively:

$$\sum_{\mathcal{T}_k \sim p(\mathcal{T})} \log p(Y|X, X_c, Y_c) = \sum_{\mathcal{T}_k \sim p(\mathcal{T})} \sum_{i=1}^{n} \log \mathcal{N}(y_i|\mu_i, \sigma_i)$$
$$\text{where} \quad (\mu_i, \sigma_i) = f_\theta(x_i, r), \quad r = f_\phi(\{x_j, y_j\}_{j=1}^{m}) \tag{1}$$

The set encoding architecture differentiates the type of NPs. In Conditional neural process(CNP) (Garnelo et al., 2018a), the representation $r$ is a deterministic variable such that, by using mean aggregation, the encoder maps the context set into a single deterministic representation $r = 1/m \sum_{j=1}^{m} f_\phi(x_j, y_j)$. Neural process (NP) (Garnelo et al., 2018b) introduces a probabilistic latent variable $z$ to model functional uncertainty as a stochastic process such that the parameters of output distribution may change according to the sampled value of $z$. Due to the intractable log-likelihood, a training objective is derived based on variational inference which can be decomposed into two terms, reconstruction term and regularization term:

$$\log p(Y|X, X_c, Y_c) \geq \mathbb{E}_{q_\phi(z|X,Y)} \left[ \log \frac{p_\theta(Y|X, r, z)p_\theta(z|X_c, Y_c)}{q_\phi(z|X, Y)} \right]$$
$$= \mathbb{E}_{q_\phi(z|X,Y)} \left[ \log p_\theta(Y|X, r, z) \right] - \text{KL}(q_\phi(z|X, Y)||p_\theta(z|X_c, Y_c)) \tag{2}$$

For simplicity, the prior distribution $p_\theta(z|X_c, Y_c)$ is approximated by $q_\phi(z|X_c, Y_c)$. However, as pointed out in Kim et al. (2019), the mean aggregation over the context set is too restrictive to describe the dependencies of the set elements. To enhance the expressiveness of the task representation, Attentive neural process(ANP) accommodates an attention mechanism (Vaswani et al., 2017) into the encoder, which generates the local deterministic representation $r_i$ corresponding to a target data point $x_i$, and addresses the underfitting issue in NP. Although additional set encoding methods such as the kernel method and the bayesian aggregation considering task information have been suggested (Xu et al., 2020; Volpp et al., 2021), the attentive neural process is mainly considered as the baseline in terms of model versatility.

## 2.2 BAYESIAN ATTENTION MODULE

Consider $m$ key-value pairs, packed into a key matrix $K \in \mathbb{R}^{m \times d_k}$ and a value matrix $V \in \mathbb{R}^{m \times d_v}$ and $n$ queries packed into $Q \in \mathbb{R}^{n \times d_k}$, where the dimensions of the queries and keys are the same. Attention mechanisms aim to create the appropriate values $O \in \mathbb{R}^{n \times d_v}$ corresponding to $Q$ based on the similarity metric to $K$, which are typically computed via an alignment score function $g$ such that $\Phi = g(Q, K)$. Then, a softmax function is applied to allow the attention weight $W \in \mathbb{R}^{n \times m}$ to satisfy the simplex constraint so that the output features can be obtained by $O_{i,j} = W_i \cdot V_j$:

$$W_{i,j} = \frac{\exp(\Phi_{i,j})}{\sum_{j=1}^{m} \exp(\Phi_{i,j})} \tag{3}$$

Note that there are many options for the alignment score function $g$ where a scaled dot-product or a neural network are widely used.
Bayesian attention module (Fan et al., 2020) considers a stochastic attention weight $W$. Compared to other stochastic attention methods (Shankar & Sarawagi, 2018; Lawson et al., 2018; Bahuleyan et al., 2018; Deng et al., 2018), it requires minimal modification to the deterministic attention mechanism described above so that it is compatible with the existing frameworks, which can be adopted in a straightforward manner. Specifically, un-normalized attention weights $\hat{W}$ such that $W_{i,j} = \hat{W}_{i,j}/\sum_{j=1}^{m} \hat{W}_{i,j}$ are sampled via the variational distribution $q_\phi(\hat{W}|Q, K)$, which can be trained via amortized variational inference:

$$\log p(O|Q, K, V) \geq \mathbb{E}_{q_\phi(\hat{W}|Q,K)} \left[ \log p_\theta(O|V, \hat{W}) \right] - \text{KL}(q_\phi(\hat{W}|Q, K)||p_\theta(\hat{W})) \tag{4}$$

Considering the random variable $\hat{W}$ to be non-negative such that it can satisfy the simplex constraint by normalization, the variational distribution $q_\phi(\hat{W}|Q, K)$ is set to Weibull$(k, \lambda)$ and the prior distribution $p_\theta(\hat{W})$ is set to Gamma$(\alpha, \beta)$. This $\lambda$ can be obtained by the standard attention mechanism, on the other hands, $\alpha$ can be either a learnable parameter or a hyper-parameter depending on that the model follows key-based contextual prior described in the following paragraph. The remaining variables $k$ and $\beta$ are regarded as user-defined hyper-parameters. By introducing Euler–Mascheroni constant $\gamma$ (Zhang et al., 2018; Bauckhage, 2014), the KL divergence in Equation 4 can be computed in an analytical expression. The detailed derivation is shown in Appendix A.

$$\text{KL}(\text{Weibull}(k, \lambda)||\text{Gamma}(\alpha, \beta)) = \frac{\gamma\alpha}{k} - \alpha \log k + \beta\lambda\Gamma(1 + \frac{1}{k}) - \gamma - 1 - \alpha \log \beta + log\Gamma(\alpha) \quad (5)$$

Samples from Weibull distribution can be obtained using a reparameterization trick exploiting an inverse CDF method: $\lambda(-\log(1 - \epsilon))^{\frac{1}{k}}$, $\epsilon \sim \text{Uniform}(0, 1)$. Note that mean and variance of Weibull$(k, \lambda)$ are computed as $\lambda\Gamma(1 + 1/k)$ and $\lambda^2 \left[\Gamma(1 + 2/k) - (\Gamma(1 + 1/k))^2\right]$. It can be observed that the variance of obtained samples decreases as $k$ increases.

**Key-based contextual prior** To model the prior distribution of attention weights, key-based contextual prior was proposed. This method allows the neural network to calculate the shape parameter $\alpha$ of the gamma distribution as a prior distribution. This leads to the stabilization of KL divergence between standard attention weights and sampled attention weights, and it prevents overfitting of the attention weights (Fan et al., 2020). In this paper, we explain the reason why the key-based context prior is important for capturing an appropriate representation focusing on a context dataset from the prosepective of information theory.

## 3 NEURAL PROCESSES WITH STOCHASTIC ATTENTION

We expect NPs to appropriately represent the context dataset to predict target data points in a novel task. However, if the context dataset has a limited task distribution and noise like real-world problems, conventional NPs tend to sensitively react to this noise and maximize the objective function including irreducible noise. These irreducible noises do not completely correlate with context information, so that this phenomenon derives meaningless set encoding of the context dataset in training phase and hinders adaptation to new tasks. In other words, the output of NPs $p_\theta(y_i|x_i, z, r_i)$ does not depend on $(z, r_i)$. To preserve the quality of context representation, we propose a method that better utilizes the context dataset by exploiting stochastic attention with the key-based contextual prior to NPs. We show that the proposed method enables to create more precise context encoding by adjusting stochasticity, even in very restricted task distribution regimes.

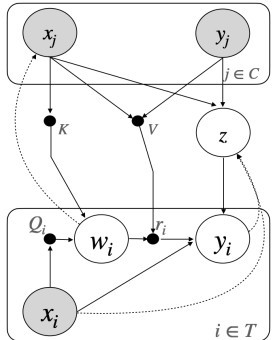

Figure 2: Probabilistic graphical models for the proposed method

### 3.1 GENERATIVE PROCESS

As with the Attentive neural process(Kim et al., 2019), the proposed method consists of the two types of encoders and a single decoder architecture. The first encoder embeds $(X_c, Y_c)$ to a global representation $z$ and the second encoder makes a local representation $r_i$ by $(x_i, X_c, Y_c)$. The global representation $z$ serves to represent entire context data points $(X_c, Y_c)$, whereas the local representation $r_i$ is in charge of fine-grained information between target data $x_i$ and context $(X_c, Y_c)$ for predicting the output distribution $p(y_i|x_i, (z, r_i))$. Unlike the ANP, the proposed model considers all intermediate representations $(z, r_i)$ as stochastic variables. We exploit the Bayesian attention module to create a local representation $r_i$ by the stochastic attention weights $w_i$ with the key-based contextual prior (Fan et al., 2020). To use the reparameterization trick for sampling stochastic variables, we draw random noise samples $\epsilon_1 \sim \text{Unif}(0, 1), \epsilon_2 \sim \mathcal{N}(0, 1)$. First, as mentioned in section 2, the stochastic attention weights $\{w_i\}_{i=1}^N$ are obtained via the inverse CDF of the Weibull distribution with random noise $\epsilon_1$. For amortized variational inference, the prior distribution $q_\phi(\{w\}_{i=1}^N|K(X_c))$ is also derived with context

dataset $X_c$ for implementing key-based contextual prior. Meanwhile, $\epsilon_2$ is used to obtain the global representation $z$ sampled from normal distribution. The entire scheme is shown in Figure D.2c. As shown in section 2, we can calculate all the KL divergences of $z$ and $\{w\}_{i=1}^n$ as closed-form solutions. The decoder follows the standard neural processes $p(y_i|x_i, z, r_i) = \mathcal{N}(y_i|\mu_\phi(x_i, z, r_i), \sigma_\phi^2(x_i, z, r_i))$. See Appendix D for implementation details.

## 3.2 LEARNING AND INFERENCE

Unlike the Attentive neural process(Kim et al., 2019), the proposed method regards all representations $z$ and $W = \{w_i\}_{i=1}^n$ as stochastic variables, so that we clearly drive the objective function according to amortized variational inference. Based on the objective function of Neural process(Garnelo et al., 2018b), the proposed method adds KL divergence of stochastic attention weight $W = \{w_i\}_{i=1}^n$. Assuming the independence between $z$ and $\{w_i\}_i^n$ such that individual $w_i$ is only dependent on $x_i$ and $(X_c, Y_c)$ as seen in Figure 2, the objective function for each task $\mathcal{T}_k$ is presented as follows:

$$\mathcal{L}_{\mathcal{T}_k}(\phi, \theta) = \sum_{i=1}^N \Big[ \log p_\theta(y_i|x_i, z, r_i) - \mathrm{KL}\big(q_\phi(w_i|x_i, X_c)|q_\phi(w_i|X_c)\big) \Big] \\ - \mathrm{KL}\big(q_\phi(z|X, Y)|q_\phi(z|X_c, Y_c)\big) \tag{6}$$

Note that $(X, Y)$ follows a task $\mathcal{T}_k$ and each task $\mathcal{T}_k$ is drawn from task distribution $p(\mathcal{T})$. The final objective function is $\mathcal{L}(\phi, \theta) = \mathbb{E}_{\mathcal{T}_k}[\mathcal{L}_{\mathcal{T}_k}(\phi, \theta)]$. From the perspective of amortized variational inference, the prior distributions of $p(z)$ and $p(\{w_i\})_{i=1}^n$ should be defined. With regard to $z$, we follow the standard neural processes wherein $p(z)$ is defined as $q_\phi(z|\{X_c, Y_c\})$ (Garnelo et al., 2018b). In the case of $\{w_i\}_{i=1}^N$, we introduce the strategy of Bayesian attention modules (Fan et al., 2020). The prior distribution of $\{w_i\}_{i=1}^N$ defined as a key-based contextual prior, $q_\phi(w_i|X_c)$, not only stabilizes the KL divergence but also activates the representation $w_i$ to pay more attention to the context dataset. In the next section, this objective function can be described in terms of information theory as role of regularization to pursue the original goal of NPs, which is deriving appropriate identifiers for target datasets in novel tasks.

## 3.3 DISCUSSION OF OBJECTIVE FUNCTION FROM VIEWS OF INFORMATION THEORY

A novel part of the proposed method is the use of the stochastic attention mechanism to more leverage context information, thereby stably capturing the dependency of context and target datasets in noisy and very limited task distributions. In this subsection, based on information theory, we elaborate the goal of NPs and our novel stochastic attention mechanism explained as the regularization of latent variables to pay more attention to context datasets. This enables us to understand latent variables and their requirements semantically. According to subsection 3.2, the objective function is categorized into two terms: a reconstruction $\log p_\theta(y_i|x_i, z, r_i)$ and regularization as two KL Divergences. Suppose that $\{x_i, y_i\}$ is a data point in a target dataset, $\mathcal{D}$ is a given context dataset, latent variables $z$ and $r_i$ are considered as $Z$, which is defined as the information bottleneck for $\mathcal{D}$, we suggest that maximizing a reconstruction term corresponds to increase $I(y_i, \mathcal{D}|x_i)$ and minimizing two KL Divergence means to decrease $I(Z, x_i|D)$.

First, the mutual information of the target value $y_i$ and the context dataset $\mathcal{D}$ given the input feature $x_i$, $I(y_i, \mathcal{D}|x_i)$, is the metric to identify that NPs is adapted to a novel task. Suppose $p(y_i|x_i, \mathcal{D})$ is output value of NPs and $p(y_i|x_i) = \mathbb{E}_D\big[p(y_i|x_i, \mathcal{D})\big]$ is output value not conditioned on context dataset $\mathcal{D}$. If NPs completely fail to adapt new tasks using the context dataset $\mathcal{D}$, the suggested metric $I(y_i, \mathcal{D}|x_i)$ goes to 0.

$$I(y_i, \mathcal{D}|x_i) = \mathbb{E}_{y_i, \mathcal{D}} \left[ \log \frac{p_\theta(y_i|x_i, \mathcal{D})}{p_\theta(y_i|x_i)} \right] \tag{7}$$

Assumed that target data point $\{x_i, y_i\}$ is sampled from $\mathcal{T}_k$, which is uncontrollable data generating process, the reconstruction term $\sum_{\{x_i, y_i\}} \log p(y_i|x_i, Z)$ can be regarded to $I(y_i; Z|x_i)$ based on information bottleneck theorem; $I(y_i; \mathcal{D}|x_i) \geq I(y_i; Z|x_i)$ holds. The detailed explanation is given

in Appendix C. The objective function should be designed to increase $I(y_i, \mathcal{D}|x_i)$ by an appropriate identifier for a novel task considering a context dataset $\mathcal{D}$. However, as mentioned in the previous sections, NPs suffer from irreducible noises and limited task distribution in meta training. This phenomena means that NPs increase the objective function by learning the way to directly map from $x_i$ to $y_i$ including noises and some features only in meta-training, instead of taking into consideration of the context $\mathcal{D}$. To make latent variables $Z$ capture proper dependencies, we define the regularization accounting for latent variables to pay more attention to the context dataset.

This regularization is the mutual information of latent variable $Z$ and input $x_i$ given context dataset $\mathcal{D}$ to indicates how similar information latent variable $Z$ and target $x_i$ contain. We regard $p(Z|x_i, D)$ as the distribution of latent variables that considers both context dataset $D$ and target $x_i$, meanwhile $p(Z|D) = \mathbb{E}_{x_i}\big[p(Z|x_i, D)\big]$ as the distribution of latent variables only depending on the context dataset $\mathcal{D}$. If latent variables $Z$ have totally different information against the target $x_i$ given the context dataset $\mathcal{D}$, It can be $I(Z, x_i|\mathcal{D}) = 0$

$$I(Z, x_i|\mathcal{D}) = \mathbb{E}_{Z,x_i} \left[ \log \frac{p(Z|x_i, \mathcal{D})}{p(Z|D)} \right] \tag{8}$$

We bring $I(y_i; \mathcal{D}|x_i) - I(Z, x_i|\mathcal{D})$ to be maximized instead of $I(y_i, \mathcal{D}|x_i)$ so as to model adequately dependency between context and the target dataset. By decreasing $I(Z, x_i|\mathcal{D})$ and increasing $I(y_i, \mathcal{D}|x_i)$ in meta-training, we expect that the model learns the way to differentiate $Z$ and $x_i$ and construct an identifier which can further consider $Z$ and $x_i$ together. Note that $\phi$ is the parameter of the function to transform $\mathcal{D}$ to the information bottleneck $Z$ and $\theta$ is the parameter of the identifier for target data point $y_i$, we show that $I(y_i; \mathcal{D}|x_i) - I(Z, x_i|\mathcal{D}) \geq I_{\phi,\theta}(y_i; Z|x_i) - I_\phi(Z, x_i|\mathcal{D})$ if $\phi, \theta$ is perfectly able to leverage all information about $Z$ and $x_i$. We assume that neural network architectures can perform an intended manners.

**Theorem 1.** *Let $Z$ be the representation of the context dataset $\mathcal{D}$ and it follows an information bottleneck. The following equation holds when the latent variable $Z$ can be split into $\{z, w_i\}$ and $z$ is only dependent on $\mathcal{D}$, but $w_i$ is dependent on $\mathcal{D}$ and $x_i$,*

$$\mathcal{L}_{\mathcal{T}_k}(\phi, \theta) \leq I(y_i; \mathcal{D}|x_i) - I(Z, x_i|\mathcal{D}) \tag{9}$$

*where $w_i$ is obtained by the Bayesian attention modules (Fan et al., 2020), and $\mathcal{L}_{\mathcal{T}_k}$ is defined in Equation 6 and its probabilistic graphical model follows Figure 2.*

As mentioned, if the neural network architectures $\phi, \theta$ are able to completely represent to $Z$ and $y_i$ given $D$ and $x_i$, $\mathcal{L}_{\mathcal{T}_k}(\phi, \theta) = I_{\phi,\theta}(y_i; Z|x_i) - I_\phi(Z; x_i|\mathcal{D})$ ideally holds. In this subsection, we reveal that stochastic attention mechanism satisfy on this regularization as mentioned. $p(Z|x_i, \mathcal{D})$ is regarded as $q_\phi(w_i|x_i, \{X_c, Y_c\})$ and $p(Z|\mathcal{D})$ is the key-based contextual prior $q_\phi(w_i|K(X_c))$. Therefore, the stochastic attention mechanism with key-prior that requires $\mathcal{D}$ and $x_i$ as well as being able to this regularization for latent variables to pay more attention to the context dataset in Equation 8 is considered. By Theorem 1, the network parameters $\phi$ and $\theta$ are updated by the gradient of the objective function in Equation 6 semantically improves $I(y_i; \mathcal{D}|x_i) - I(Z, x_i|\mathcal{D})$. Therefore, we present the experimental results in the next section. They indicate that we can implement a model that can complete unseen data points in novel tasks by fully considering the context representations. The detailed analysis is presented in Appendix G and Appendix H.

## 4 EXPERIMENT

In this section, we describe our experiments to answer three key questions: *1) Are existing regularization methods such as weight decay, importance-weighted ELBO as well as recent methods(Information dropout, Bootstrapping, and Bayesian aggregation) effective to properly create context representations ?; 2) Can the proposed method reliably capture dependencies between target and context datasets even under noisy situations and limited task distributions ?; 3) Is it possible to improve performance via the appropriate representation of the dataset ?*

To demonstrate the need for neural process with stochastic attention, we choose CNP, NP (Garnelo et al., 2018a;b), and ANP (Kim et al., 2019) as baselines, which are commonly used. For all baselines, we employ weight decay and importance-weighted ELBO that requires several samples against one

Table 1: Results of likelihood values on the synthetic 1D regression experiment. The test samples are drawn from GPs with various kernel functions; All methods are trained on the samples from the RBF GP function. To consider noisy setting, we artificially generate a periodic noise in the training step. **Bold entries** indicates the best results.

| | RBF kernel GP(*noises*) | | RBF kernel GP | | Matern kernel GP | | Periodic kernel GP | |
|---|---|---|---|---|---|---|---|---|
| | context | target | context | target | context | target | context | target |
| CNP | 0.233±0.036 | -0.478±0.034 | 0.440±0.013 | 0.026±0.014 | 0.246±0.021 | -0.544±0.024 | 0.176±0.022 | -0.978±0.033 |
| NP | -0.151±0.012 | -0.690±0.059 | 0.107±0.018 | -0.177±0.024 | -0.235±0.032 | -0.918±0.029 | -0.400±0.046 | -1.321±0.047 |
| ANP | 0.228±0.021 | -0.603±0.036 | 0.405±0.0010 | -0.097±0.024 | 0.254±0.006 | -0.488±0.014 | 0.111±0.034 | -0.951±0.070 |
| *(Weight decay; $\lambda = 0.001$)* | | | | | | | | |
| CNP | 0.240±0.041 | -0.471±0.037 | 0.460±0.016 | 0.045±0.020 | 0.278±0.035 | -0.472±0.041 | 0.185±0.012 | -0.970±0.037 |
| NP | -0.153±0.033 | -0.709±0.070 | 0.122±0.026 | -0.178±0.036 | -0.228±0.034 | -0.936±0.026 | -0.401±0.047 | -1.315±0.072 |
| ANP | 0.957±0.015 | -0.442±0.030 | 1.050±0.011 | 0.053±0.034 | 1.014±0.009 | -0.209±0.036 | 0.926±0.007 | **-0.644±0.027** |
| *(Importance Weighted ELBO; $s = 5$)* | | | | | | | | |
| CNP | 0.271±0.024 | -0.442±0.041 | 0.478±0.012 | 0.059±0.029 | 0.321±0.019 | -0.460±0.036 | 0.231±0.017 | -0.957±0.041 |
| NP | -0.155±0.014 | -0.633±0.026 | 0.067±0.016 | -0.213±0.030 | -0.238±0.015 | -0.779±0.018 | -0.330±0.008 | -1.094±0.088 |
| ANP | 0.771±0.012 | -0.470±0.034 | 0.895±0.016 | -0.031±0.026 | 0.800±0.031 | -0.324±0.027 | 0.704±0.026 | -0.687±0.039 |
| *(Bayesian Aggregation)* | | | | | | | | |
| CNP | 0.351±0.049 | -0.508±0.084 | 0.575±0.016 | 0.112±0.025 | 0.349±0.018 | -0.569±0.097 | 0.278±0.018 | -1.026±0.051 |
| NP | -0.406±0.035 | -0.723±0.067 | -0.201±0.024 | -0.389±0.024 | -0.450±0.014 | -0.837±0.036 | -0.537±0.011 | -0.877±0.021 |
| *(Functional representation)* | | | | | | | | |
| ConvCNP | 1.314±0.007 | -0.428±0.033 | 1.326±0.010 | 0.084±0.020 | 1.319±0.007 | **-0.119±0.023** | 1.358±0.003 | **-0.502±0.022** |
| ConvNP | 0.873±0.154 | -0.469±0.041 | 0.729±0.131 | 0.053±0.017 | 0.832±0.005 | -0.163±0.025 | 0.953±0.077 | -0.522±0.025 |
| *(Regularization for local representation)* | | | | | | | | |
| ANP *(dropout)* | 0.158±0.019 | -0.593±0.035 | 0.372±0.026 | -0.163±0.025 | 0.136±0.020 | -0.575± 0.014 | 0.014±0.017 | -0.877±0.036 |
| Bootstrapping ANP | 0.754±0.018 | -0.407±0.036 | 0.872±0.009 | 0.043±0.023 | 0.788±0.010 | -0.303± 0.032 | 0.711±0.025 | -0.717±0.033 |
| Ours | **1.374±0.004** | **-0.337±0.027** | **1.363±0.010** | **0.244±0.026** | **1.365±0.009** | -0.175±0.031 | **1.372±0.005** | -0.612±0.029 |

data point and aggregated by $\log \sum_i \exp \mathcal{L}_k$. Recent regularization methods can be also considered. Information dropout(Kingma et al., 2015; Achille & Soatto, 2018), Bayesian aggregation (Volpp et al., 2021) and Bootstrapping (Lee et al., 2020) are chosen. Bayesian aggregation is used for CNP and NP according to the original paper, meanwhile Information dropout and Bootstrapping are used for the ANP to verify regularization of local representations. When training, all methods follow their own learninig policy; however, for fair comparison, all methods perform one Monte-Carlo sampling across a test data point during testing. We respectively conduct 5 trials for each experiment and report average scores. As a metric to evaluate the model's predictive performance, we use the likelihood of the target dataset, while the likelihood of the context dataset is considered as a metric to measure the extent to which models can represent a context dataset. Since these metrics are proportional to model performance, a higher value indicates a better performance. In these experiments, we show that the proposed method substantially outperforms various NPs and their regularization methods.

## 4.1 1D REGRESSION UNDER NOISY SITUATIONS

We conduct synthetic 1D regression experiments to test whether our model adequately captures the dependency of a dataset under noisy situation. We trained all models with samples from the RBF GP functions. We tested them with samples from various kernel functions (Matern, Periodic inclduing RBF) to identify that models is capable of adaptatation to new tasks. To consider noisy environments, we artificially generate noises into the training dataset. Our model maintains outstanding performance compared to other baselines. In particular, when the test data comes from periodic kernel GP, which is the most different from the RBF GP, the proposed model utilizes context datasets appropriately regardless of the test data sources, whereas other models do not properly use the context data. When comparing the likelihood values of context datasets, the proposed method preserves equivalent scores in all cases, but the others do not. As shown in Figure 1a, the proposed method captures context points better than other methods, as only it method correctly captures the relationship even under noisy situations. These results support our claim, the proposed method utilizes context datasets in prediction. We report graphical explanation and the additional experimental result of 1D regression without noises in Appendix E.

## 4.2 PREDATOR-PREY MODEL AND IMAGE COMPLETION

We apply NPs to a domain shift problem called the predator-prey model, proposed in the ConvCNP(Gordon et al., 2019). In this experiment, we train models with the simulated data and applied models to predict the real-world data set, Hudson's bay the hare-lynx. Note that the hare-lynx dataset

Table 2: Results of likelihood values on the Sim2Real experiment using predator-prey simulation and image completion with the CelebA dataset. In the image completion, there are several experimental cases; The number of context points is in $\{50, 100, 300, 500\}$. We report that likelihood of context points is averaged over all experiment cases and elaborate individual likelihood values of target points. **Bold entries** indicates the best results.

| | Sim2Real : Predator-Prey | | | | Image Completion : CelebA | | | | |
| | Simulation | | Real | | context | Target | | | |
| | context | target | context | target | | 50 | 100 | 300 | 500 |
| CNP | 0.395±0.000 | 0.274±0.000 | -2.645±0.085 | -3.120±0.245 | 3.452±0.002 | 2.662 | 3.077 | 3.359 | 3.414 |
| NP | -0.259±0.004 | -0.369±0.002 | -2.526±0.030 | -2.816±0.088 | 3.072±0.003 | 2.483 | 2.786 | 2.999 | 3.042 |
| ANP | 1.211±0.001 | 0.961±0.001 | -1.756±0.187 | -3.742±0.448 | 3.100±0.012 | 2.492 | 2.806 | 3.02 | 3.06 |
| *(Weight decay; $\lambda = 0.001$)* | | | | | | | | | |
| CNP | 0.393±0.000 | 0.265±0.000 | -2.661±0.023 | -3.049±0.096 | 2.969±0.014 | 2.21 | 2.6 | 2.866 | 2.918 |
| NP | -0.069±0.003 | -0.185±0.001 | -2.571±0.054 | -2.914±0.108 | 2.429±0.008 | 1.878 | 2.17 | 2.368 | 2.406 |
| ANP | 1.520±0.001 | 1.180±0.002 | 0.212±0.129 | -2.807±0.665 | 2.832±0.025 | 2.339 | 2.611 | 2.795 | 2.829 |
| *(Importance Weighted ELBO; $s = 5$)* | | | | | | | | | |
| CNP | 0.476±0.000 | 0.337±0.000 | -2.596±0.048 | -3.038±0.183 | 3.474±0.002 | 2.67 | 3.091 | 3.379 | 3.436 |
| NP | 0.055±0.002 | -0.067±0.002 | -2.514±0.048 | -2.760±0.107 | 3.179±0.023 | 2.49 | 2.856 | 3.112 | 3.163 |
| ANP | 1.004±0.001 | 0.775±0.001 | -1.800±0.203 | -3.728±0.486 | 3.131±0.002 | 2.477 | 2.826 | 3.057 | 3.101 |
| *(Bayesian Aggregation)* | | | | | | | | | |
| CNP | 0.551±0.000 | 0.429±0.000 | -2.654±0.156 | -2.952±0.112 | 3.583±0.013 | **2.733** | 3.18 | 3.474 | 3.53 |
| NP | 0.332±0.003 | 0.192±0.005 | -2.489±0.065 | -3.009±0.315 | 3.483±0.032 | 2.529 | 3.072 | 3.4 | 3.46 |
| *(Functional representation)* | | | | | | | | | |
| ConvCNP | 2.509±0.000 | 2.139±0.000 | 1.758±0.089 | **-0.431±0.457** | - | - | - | - | - |
| ConvNP | 2.467±0.000 | 2.124±0.000 | 1.652±0.089 | -1.180±0.439 | - | - | - | - | - |
| *(Regularization for local representation)* | | | | | | | | | |
| ANP *(dropout)* | 1.492±0.003 | 1.134± 0.004 | 1.068±0.039 | -6.215±1.577 | 3.078±0.006 | 2.501 | 2.801 | 3.005 | 3.043 |
| Bootstrapping ANP | 2.537±0.001 | 2.166± 0.001 | **2.451±0.019** | -3.382±1.288 | 3.172±0.009 | 2.453 | 2.837 | 3.095 | 3.145 |
| Ours | **2.711±0.000** | **2.297±0.000** | 2.429±0.031 | -1.766±0.885 | **4.119±0.010** | 2.653 | **3.21** | **3.787** | **3.948** |

is very noisy, unlike the simulated data. The detailed explanation of datasets is written in Appendix F. Recent regularization methods such as Bayesian aggregation, Information dropout, and Bootstrapping are effective to improve performance. In particular, Bootstrapping method significantly influences predictive performance compared of other ANPs as mentioned in the original paper(Lee et al., 2020). However, we observe that the proposed method is superior to the other models in both simulated test set and real-world dataset. In particular, all baselines drastically decrease the likelihood values due to noises during the test, while the proposed model preserves the performance because it is robust to noises. See the left side of Table 2. We report the additional experimental result with periodic noises and the graphical explanation is presented in Appendix F.

Second, we conduct the image completion task in which the models generate images under some given pixels. The experiment setting follows the previous experiments (Garnelo et al., 2018a;b; Kim et al., 2019; Lee et al., 2020). To conduct fair comparison, all models employ only MLP encoders and decoders and the multi-head cross attention is used for local representations; the variant of ANPs. We indicate that the proposed method records the best score as seen in the right side of Table 2. The best baseline records 3.53 with 500 context pixels in Table 2, whereas our method attains 2.653 of likelihood even with 50 context pixels and grows to 3.948 with 500 context pixels. Referred to Lee et al. (2020)'s paper, the previous highest score is 4.150 of likelihood for context pixels and 3.129 of likelihood for target pixels. As Kim et al. (2019) mentioned that the more complex architecture like the self-attention mechanism enhances the completion quality than the MLP encoders, these scores were obtained by ANP with self-attention encoders and employing bootstrapping. However, we achieves to obtain comparable results for context points and exceed the previous highest score for target points by a significant margins without exhausted computations and complicated architectures. The completed images by ours and baselines are shown in Appendix J.

### 4.3 MOVIELENS-100K DATA

We demonstrate the robustness and effectiveness of our method using a real-world dataset, the Movie-Lenz dataset, which is commonly used in recommendation systems. This setting has an expensive data collection process that restricts the number of users. In this section, we report how well the proposed method can be generalized to novel tasks using limited tasks during the meta-training. To

train NPs, we decide to split this dataset according to user ID and regard the rating samples made by one user as a task. The purpose of this experiment is that NP provides an identifier to serve for new visitor using very few rating samples. We follow the setting used in the existing work(Galashov et al., 2019). The proposed model performs better than the other methods. As mentioned in subsection 4.2, it captures the information from the context dataset, while other methods suffer from noise in the data and lack of users in meta-training. The experiment result of comaprison with baselines is reported in Appendix I. To validate use of real applications, we compare with existing studies.

According to Movielens-100k benchmarks, the state-of-the-art RMSE score is about 0.890, which can be obtained using graph structures about users and films(Rashed et al., 2019). For fair comparison, we train and test the proposed method on the same setting, named as U1 splits. When evaluate baselines on the u1.test dataset, we randomly draw samples of corresponding users from the u1.base used in the meta-training and regard as context data points. Although we do not use graph structures, as seen in Table 3, The ANP points the comparable result of 0.909, meanwhile the proposed model attains a promis-

Table 3: Comparison of RMSE scores on u.test in MoveLens-100k

|  | RMSE |
| --- | --- |
| GLocal-K (Han et al., 2021) | 0.890 |
| GraphRec + Feat (Rashed et al., 2019) | 0.897 |
| GraphRec(Rashed et al., 2019) | 0.904 |
| GC-MC + Feat(Berg et al., 2017) | 0.905 |
| GC-MC (Berg et al., 2017) | 0.910 |
| ANP (contexts: 10)(Kim et al., 2019) | 0.909 |
| Ours (contexts: 10) | 0.895 |

ing result, 0.895 of the RMSE value. This experiment indicates that the proposed method can reliably adapt to new tasks even if it provides small histories, and we identify again that our model can properly work on noisy situations.

## 5 RELATED WORKS

The stochastic attention mechanism enables the capturing of complicated dependencies and regularizing weights based on the user's prior knowledge. However, such methods cannot utilize back-propagation because they do not consider the reparameterization trick to draw samples (Shankar & Sarawagi, 2018; Lawson et al., 2018; Bahuleyan et al., 2018; Deng et al., 2018). Even if these methods employ a normal distribution as a posterior distribution of latent variables, satisfying the simplex constraints, which sum to one (Bahuleyan et al., 2018), is impossible. Recently, the Bayesian attention module suggests that the attention weights are samples from the Weibull distribution whose parameters can be reparameterized, so this method can be stable for training and maintaining good scalability (Fan et al., 2020).

Since the Conditional neural process have been proposed(Garnelo et al., 2018a;b), several studies have been conducted to improve the neural processes in various aspects. The Attentive neural process modify the set encoder as an cross-attention mechanism to increase the performance of the predictability and interpretability (Kim et al., 2019). Some studies investigate NPs for sequential data (Yoon et al., 2020; Singh et al., 2019). Trials have been combined with optimization-based meta-learning to increase the performance of NPs for the image classification task with a pre-trained image encoder (Rusu et al., 2018; Xu et al., 2020). The convolutional network can be used for set encoding to obtain translation-invariant predictions with the assistance of supplementary data points (Gordon et al., 2019; Foong et al., 2020). Similar to this study, the Bayesian context aggregation suggests the importance of context information over tasks(Volpp et al., 2021) and the Bootstrapping NPs orthogonally improves the predictive performance under the model-data mismatch. (Lee et al., 2020). Unfortunately, there is no clear explanation of how stochasticity has helped improve performance and does not show a fundamental approach to enhancing dependency between the context and the target dataset in terms of set encoding.

## 6 CONCLUSION

In this work, we propose an algorithm, neural processes with stochastic attention that effectively leverages context datasets and adequately completes unseen data points. We utilize a stochastic attention mechanism to capture the relationship between the context and target dataset and adjust stochasticity during the training phase for making the model insensitive to noises. We demonstrate that the proposed method can be explained based on information theory. The proposedregularization

leads representations paying attention to the context dataset. We conducted various experiments to validate consistent enhancement of the proposed model. We identify that the proposed method substantially outperforms the conventional NPs and their recent regularization methods by substantial margins. This evidence from this study suggests that the proposed method can provides a better identifier to the novel tasks that the model has not experienced.

## ACKNOWLEDGMENTS

This work was conducted by Center for Applied Research in Artificial Intelligence (CARAI) grant funded by DAPA and ADD [UD190031RD].

## REPRODUCIBILITY STATEMENT

Our code is available at `https://github.com/MingyuKim87/NPwSA`. For convenience reproducibility, both training and evaluation codes are included.

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

# A  CLOSED FORM SOLUTION FOR THE KL DIVERGENCE BETWEEN WEIBULL AND GAMMA DISTRIBUTION

The generalized gamma distribution contains both Weibull and gamma distribution as special cases. We are concerned with the three-parameter version of generalized gamma distribution introduced in Stacy (Stacy et al., 1962). Its parameters can be categorized into one scale parameter $a$ and two shape parameters $d$ and $p$. Its probability density function is defined for $x \in [0, \infty)$ and given by

$$f(x|a, d, p) = \frac{p}{a^d} \frac{x^{d-1}}{\Gamma(d/p)} \exp\left[-(\frac{x}{a})^p\right] \tag{A.1}$$

where $\Gamma(\cdot)$ is the gamma function, and parameters $a, d, p > 0$. For $d = p$, it corresponds to the Weibull distribution, and if $p = 1$, it becomes the gamma distribution. Bauckhage (Bauckhage, 2014) derives a closed form solution for the kullback-leibler divergence between two generalized gamma distribution as follow.

$$
\begin{aligned}
\mathrm{KL}(f_1|f_2) &:= \int_0^\infty f_1(x|a_1, d_1, p_1) \log \frac{f_1(x|a_1, d_1, p_1)}{f_2(x|a_2, d_2, p_2)} dx \\
&= \log\left(\frac{p_1 a_2^{p_2} \Gamma(d_2/p_2)}{p_2 a_1^{p_1} \Gamma(d_1/p_1)}\right) + \left[\frac{\psi(d_1/p_1)}{p_1} + \log a_1\right](d_1 - d_2) \\
&\quad + \frac{\Gamma(\frac{d_1+p_2}{p_1})}{\Gamma(\frac{d_1}{p_1})} \left(\frac{a_1}{a_2}\right)^{p_2} - \frac{d_1}{p_1}
\end{aligned}
\tag{A.2}
$$

where $f_1, f_2$ are the generalized gamma distributions, and $\psi(\cdot)$ is digamma function. The Weiull distribution with scale parameter $\lambda$ and shape parameter $K$ coincides with the generalized gamma distribution $a = \lambda$ and $d, p = K$. For gamma distribution with scale parameter $\beta$ and shape parameter $\alpha$, it becomes the generalized gamma distribution $a = {}^1\!/\!{}_\beta$, $d = \alpha$ and $p = 1$. The KL divergence between the Weibull distribution $(\lambda, k)$ and the gamma distribution $(\alpha, \beta)$ amounts to

$$
\begin{aligned}
\mathrm{KL}(f_1|f_2) = {}&\log k - \alpha \log \beta + \log \Gamma(\alpha) \\
&+ \psi(1) - \frac{\alpha}{ks}\psi(1) - \alpha \log \lambda + \lambda\beta\Gamma(1 + {}^1\!/\!{}_K) - 1
\end{aligned}
\tag{A.3}
$$

We introduce $\psi(1) = -\gamma \approx 0.5772$ as Euler constant. Finally, we obtain Equation 5

$$
\begin{aligned}
\mathrm{KL}(\mathrm{Weibull}(k, \lambda) || \mathrm{Gamma}(\alpha, \beta)) = {}&\frac{\gamma\alpha}{k} - \alpha \log \lambda + \log k + \beta\lambda\Gamma\left(1 + \frac{1}{k}\right) \\
&- \gamma - 1 - \alpha \log \beta + \log \Gamma(\alpha)
\end{aligned}
\tag{A.4}
$$

# B ELBO DERIVATIONS

In this section, we derive the objective function in the manuscript of this paper. Without loss of generality, target dataset $\{X, Y\} = \{x_i, y_i\}_{i=1}^N$ and context dataset $\mathcal{D} = (X_c, Y_c) = \{x_i, y_i\}_{j=1}^M$ such that $N \gg M$. Let the log-likelihood of target data points in a task $T_k$ be $\log p(Y|X, D)$. We begin to maximizing the log-likelihood of target data points based on context dataset and representations. We suppose that it follows the graphical model in Figure D.2c

$$\log p(Y|X, D) = \log \int p(Y, Z|X, D) dz \tag{B.5}$$

$$= \log \int p(Y, Z|X, D) \frac{q(Z|X, D)}{q(Z|X, D)} dz \tag{B.6}$$

$$= \log \int \frac{p(Y, Z|X, D)}{q(Z|X, D)} q(Z|X, D) dz \tag{B.7}$$

$$= \log \int \prod_{i=1} \frac{p(y_i|x_i, Z_i)}{q(Z_i|x_i, \mathcal{D})} p(z_i|x_i, \mathcal{D}) q(z_i|x_i, \mathcal{D}) dz \tag{B.8}$$

$$= \log \mathbb{E}_{q(z)} \Big[ \prod_{i=1} \frac{p(y_i|x_i, Z_i)}{q(Z_i|x_i, \mathcal{D})} p(z_i|x_i, \mathcal{D}) \Big] \tag{B.9}$$

Applying Jensen's inequality to

$$\log p(Y|X, D) \geq \mathbb{E}_{q(z)} \Big[ \log \prod_{i=1} \frac{p(y_i|x_i, Z_i) p(Z_i|x_i, \mathcal{D})}{q(z_i|x_i, \mathcal{D})} \Big] \tag{B.10}$$

$$\geq \mathbb{E}_{q(z)} \Big[ \sum_{i=1} \log p_\theta(y_i|x_i, Z_i) - \log \frac{q(Z_i|x_i, \mathcal{D})}{p(Z_i|x_i, \mathcal{D})} \Big] \tag{B.11}$$

From $Z_i = \{z, w_i\}$ in Figure D.2c, we define $z$ is dependent on context dataset $\mathcal{D}$ and $w_i$ is dependent on target $x_i$ and context dataset $\mathcal{D}$

$$\log p(Y|X, D) \geq \mathbb{E}_{q(z)} \Big[ \sum_{i=1} \log p_\theta(y_i|x_i, Z_i) - \log \frac{q(w_i|x_i, \mathcal{D})}{p(w_i|x_i, \mathcal{D})} - \log \frac{q(z|\mathcal{D})}{p(z|\mathcal{D})} \Big] \tag{B.12}$$

We assume that $p(w_i|x_i, \mathcal{D}) \approx q_{\phi_{1_2}}(w_i|K(X_c))$ by regularization of latent variables to pay attention on context dataset. It also follows the strategy of the standard neural processes : posterior distribution of $z$ is $q_{\phi_2}(w_i|\{X_t, Y_t\})$ and prior distribution of $z$ is $q_{\phi_2}(w_i|\{X_c, Y_c\})$. Thereby, we show that ELBO become the objective function of the proposed method in Equation 6.

$$\log p(Y|X, D) \geq \mathbb{E}_{q(z)} \Big[ \sum_{i=1} \log p_\theta(y_i|x_i, Z_i) - \log \frac{q_{\phi_{1_1}}(w_i|x_i, \mathcal{D})}{q_{1_2}(w_i|, K(X_c))} - \log \frac{q_{\phi_2}(z|\{X_t, Y_t\})}{q_{\phi_2}(z|\{X_c, Y_c\})} \Big] = \mathcal{L}_{T_k} \tag{B.13}$$

## C    PROOF OF THEOREM 1

We present that $I(y_i; \mathcal{D}|x_i)$ has relation to the objective function of the proposed method. Suppose that $Z$ is information bottleneck for $\mathcal{D}$ and all variables follow the graphical models Figure 2. As described in Equation 7, the mutual information $I(y_i, \mathcal{D}|x_i)$ is to measure dependency of $y_i$ and context dataset $\mathcal{D}$. We should maximize this information to generalize the novel task. For simplicity, we set all latent variables as information bottleneck $Z$ for the context dataset $\mathcal{D}$. Note that $Z = (\{w_i\}_{i=1}^n, z)$ is information bottleneck corresponding to context representations in NPs, where $W = \{w_i, \ldots, w_n\}$ is an attention weights and $X, Y = \{(x_1, y_1), \ldots, (x_n, y_n)\}$ is respectively data points in the target data set. $\phi$ is regarded as the parameter of the function to transform $\mathcal{D}$ to the information bottleneck $Z$ and $\theta$ is the parameter of the identifier for target data point $y_i$. By information bottleneck theorem, Equation C.14 holds.

$$I(y_i; \mathcal{D}|x_i) \geq I_{\phi,\theta}(y_i; Z|x_i) = -H_{\phi,\theta}(y|x_i, Z) + H(y|x_i) \tag{C.14}$$

The target data point $\{x_i, y_i\}$ is drawn by the data generating process $\mathcal{T}_k \sim p(\mathcal{T})$, so that $H(y|x_i)$ can be regarded as a constant value due to uncontrollable factors. As a result, $-H_{\phi,\theta}(y|x_i, Z)$ is written as

$$-H_{\phi,\theta}(y_i|x_i, Z) = \mathbb{E}_{(x,y)}\big[-\log p_\theta(y_i|x_i, Z)\big] \tag{C.15}$$

We employ $\sum_{\{x_i, y_i\}} \log p_\theta(y_i|x_i, Z_\phi)$, which is unbiased estimate for $-H_{\phi,\theta}(y_i|x_i, Z)$ based on Monte Carlo sampling. Given that the concrete information bottleneck $Z$ is given, we expect that the $Z$ allows us to generalize a novel task $\{x_i, y_i\}$ by enhancing $\sum_{\{x_i, y_i\}} \log p_\theta(y_i|x_i, Z_\phi)$. However, in practice, $Z$ induced by neural networks is more likely to memorize the training dataset including irreducible noises when the number of tasks is insufficient. The conventional methods accomplished maximizing $\sum_{\{x_i, y_i\}} \log p_\theta(y_i|x_i, Z_\phi)$ by simply finding the relation between $y_i$ and $x_i$ only in the training dataset. It means that the information in representation $Z$ becomes increasingly identical to the information in $x_i$ in the meta training dataset. It does not satisfy the condition of information bottleneck for the context dataset $\mathcal{D}$.

To avoid this issue, we introduce $I(Z, x_i|\mathcal{D})$ as a regularization. By reducing dependencies of $Z$ and $x_i$, we can make the information bottleneck $Z$ focus on $\mathcal{D}$. It means that as $p(Z|x_i, \mathcal{D})$ and $P(Z|\mathcal{D})$ get closer, the target input $x_i$ less influences $Z$, but $\mathcal{D}$ has a high correlation with $Z$. In other words, minimizing $I(Z, x_i|\mathcal{D})$ is to make $Z$ and target $x_i$ more independent given context dataset $\mathcal{D}$. It encourages latent variables $Z = (w_i, z)$ to pay more attention to context dataset $\mathcal{D}$

$$
\begin{aligned}
I(Z, x_i|\mathcal{D}) &= \mathbb{E}_{x_i}\left[\mathbb{E}_{Z_i}\left[\mathrm{KL}(q(Z_i|x_i, \mathcal{D})||q(Z_i||\mathcal{D}))\right]\right] \\
&= \mathbb{E}_{x_i}\Big[\mathbb{E}_{w_i}\left[\mathrm{KL}(q(w_i|x_i, \mathcal{D})||q(w_i|\mathcal{D}))\right] + \mathbb{E}_z\left[\mathrm{KL}(q(z|x_i, \mathcal{D})||q(z|\mathcal{D}))\right]\Big]
\end{aligned} \tag{C.16}
$$

where, the $q(w_i|x_i, \mathcal{D})$ follows the Weibull distribution and $q(w_i|\mathcal{D})$ follows the gamma distribution. In this study, $q(w_i|x_i, \mathcal{D})$ can be modeled as the stochastic attention weights and $q(w_i|\mathcal{D})$ can be modeled as the key-based contextual prior $q_\phi(w_i|K)$. We can factorize all latent variables because the attention weight $w_i$ does not have a dependency on data points except for the specific data point $\{x_i, y_i\}$. We denote that $w_i$ has conditional independence over $\{w_j\}_{i \neq j}$ given $x_i$. For the global representation; $z$, to follow objective function of neural processes, we assume $q(z|x_i, \mathcal{D}) = q_\phi(z|X, Y)$ and $q_\phi(z|\mathcal{D}) = q(z|\mathcal{D})$. As mentioned in section 3, we suppose that the global representation $z$ follows normal distribution. Therefore, we derive the regularization term $I(Z, x_t|\mathcal{D})$ have relation to KL Divergence terms in our loss function. From this fact, we recognize KL Divergence terms in our objective function helps latent variables give attention on context dataset.

To summarize the mutual information between $y_i$ and $\mathcal{D}$, and the newly designed regularization, we found that

$$
\begin{aligned}
I(y_i; \mathcal{D}|x_i) &\geq I_{\phi,\theta}(y_i; Z|x_i) - I(Z, x_i|\mathcal{D}) \\
&\geq -H_{\phi,\theta}(y|x_i, Z) - I(Z, x_i|\mathcal{D}) + \mathrm{Constant}
\end{aligned} \tag{C.17}
$$

Based on the graphical models in Figure 2 and assumptions of representations $z$ and $w_i$, we identify that this equation has relation to our objective function $\mathcal{L}_{\mathcal{T}_k}$

$$
\begin{aligned}
I_{\phi,\theta}(y_i; Z|x_i) - I(Z, x_i|\mathcal{D}) &= \sum_{\{x_i, y_i\}} \log p_\theta(y_i|x_i, Z_\phi) - \mathbb{E}_z\left[\text{KL}(q_\phi(z|X, Y)|q_\phi(z|X_c, Y_c))\right] \\
&\quad - \mathbb{E}_W\left[\text{KL}(q_\phi(W|X, \mathcal{D})|q_\phi(W|K(X_c)]\right] + \text{Constant} \\
&= \mathcal{L}_{\mathcal{T}_k} + \text{Constant}
\end{aligned}
$$

(C.18)

Finally, we can derive Theorem 1 as below

$$
\mathcal{L}_{\mathcal{T}_k}(\phi, \theta) \leq I(y_i; \mathcal{D}|x_i) - I(Z, x_i|\mathcal{D})
$$

(C.19)

From Equation C.19, the gradients of $\phi$ and $\theta$ with respect to $\mathcal{L}_{\mathcal{T}_k}$ can be regarded as the direction of increasing $I(y_i; \mathcal{D}|x_i) - I(Z, x_i|\mathcal{D})$, where, $\mathcal{L}_{\mathcal{T}_k}$ is defined as the target likelihood and KL divergence of $z$ and $\{w_i\}_{i=1}^n$ in a single task.

$$
\begin{aligned}
\mathcal{L}_{\mathcal{T}_k}(\phi, \theta) &= \sum_{i=1}^N \left[ \log p_\theta(y_i|x_i, z, r_i) - \text{KL}\left(q_\phi(w_i|x_i, \mathcal{D})|q_\phi(w_i|X_\mathcal{D})\right) \right] \\
&\quad - \text{KL}\left(q_\phi(z|X, Y)|q_\phi(z|\mathcal{D})\right)
\end{aligned}
$$

(C.20)

where $X_\mathcal{D}$ is input features in the context dataset and $q_\phi(w_i|X_\mathcal{D})$ follows key-based contextual prior as described in section 3.

## D  IMPLEMENT DETAILS

We referred to most of the architectures from the paper(Kim et al., 2019) and their released source code[1]. The information dropout and importance weighted ELBO were respectively borrowed from these papers(Kingma et al., 2015; Burda et al., 2016)[2][3]. The stochastic attention can be implemented based on Bayesian attention modules(Fan et al., 2020)[4]. We migrated and revised all codes to meet our purpose. In this chapter, we follow the notation of this paper (Lee et al., 2020).

### D.1  (ATTENTIVE) NEURAL PROCESS

**MLP Encoder**  We suppose that multi-layers modules(MLP) have the structure as Equation D.21, where $l$ is the number of layers, $d_{\text{in}}$ is the dimension of input features, $d_{\text{hidden}}$ is the dimension of hidden units and $d_{\text{out}}$ is the dimension of outputs.

$$
\begin{aligned}
\text{MLP}(l, d_{\text{in}}, d_{\text{hidden}}, d_{\text{out}}) &= \text{Linear}(d_{\text{hidden}}, d_{\text{out}}) \\
&= \circ \underbrace{(\text{ReLU} \circ \text{Linear}(d_{\text{hidden}}, d_{\text{hidden}}) \cdots)}_{l-2} \\
&\circ (\text{ReLU} \circ \text{Linear}(d_{\text{in}}, d_{\text{hidden}}))
\end{aligned}
\tag{D.21}
$$

The variants of neural processes have two types of MLP encoders: The deterministic path is used in the conditional neural process(Garnelo et al., 2018a), and the stochastic path is employed in the neural process(Garnelo et al., 2018b), attentive neural process(Kim et al., 2019) and ours. The deterministic path is to aggregate all hidden units by the MLP encoder.

$$
\begin{aligned}
h &= \frac{1}{|c|} \sum_{j \in c} \text{MLP}(l_{\text{pre}}, d_x + d_y, d_h, d_h)([x_j, y_j]) \\
r &= \text{MLP}(l_{\text{post}}, d_h, d_h)(h)
\end{aligned}
\tag{D.22}
$$

Instead, the stochastic path is to aggregate all hidden units and then feed-forward a single network to generate $\mu_{\phi_1}$ and $\sigma_{\phi_1}$. We obtain the stochastic path $r$ via the reparameterization trick.

$$
\begin{aligned}
h &= \frac{1}{|c|} \sum_{j \in c} \text{MLP}(l_{\text{pre}}, d_x + d_y, d_h, d_h)([x_j, y_j]) \\
\mu_{\phi_1}, \sigma_{\phi_1} &= \text{MLP}(l_{\text{post}}, d_h, d_h)(h) \\
r &= \mu_{\phi_1} + \epsilon_2 * \sigma_{\phi_1}
\end{aligned}
\tag{D.23}
$$

**Attention encoder**  We introduce cross-attention to describe the dependency of context and target dataset. Let MHA be a multi-head attention (Vaswani et al., 2017) computed as follows :

$$
\begin{aligned}
Q' &= \{\text{Linear}(d_q, d_h)(q)\}_{q \in Q} \\
K' &= \{\text{Linear}(d_k, d_h)(k)\}_{k \in K} \\
V' &= \{\text{Linear}(d_v, d_h)(v)\}_{v \in V} \\
H &= \text{softmax}(Q'K'/\sqrt{d_h})V' \\
\text{MHA}(Q, K, V) &= \text{LayerNorm}(H)
\end{aligned}
\tag{D.24}
$$

Where, $\{d_q, d_k, d_v\}$ are respectively the dimensions of query, key and value components. The LayerNorm is the layer normalization in terms of heads.

**MLP Decoder**  The architecture of MLP decoder is similar to the stochastic encoder. In case of CNP(Garnelo et al., 2018a) and NP(Garnelo et al., 2018b), this decoder transforms all representation $z$ and the input feature $\{x_i\}_{i \in T}$ to target distribution, the normal distribution, $\{\mu_i, \sigma_i\}_{i \in T}$.

---

[1]https://github.com/deepmind/neural-processes

[2]https://github.com/kefirski/variational_dropout

[3]https://github.com/JohanYe/IWAE-pytorch

[4]https://github.com/zhougroup/BAM

$$(\mu_y, \sigma_y) = \text{MLP}(l_{\text{dec}}, d_x + d_h, d_h, 2d_y)([z, r_i, x_i]) \tag{D.25}$$

Meanwhile, in case of attentive neural process (Kim et al., 2019) and ours, the inputs of this decoder are the global representation $z$, local representation $\{r_i\}_{i \in T}$ and the input feature $\{x_i\}_{i \in T}$.

$$(\mu_y, \sigma_y) = \text{MLP}(l_{\text{dec}}, d_x + d_h + d_h, d_h, 2d_y)([z, r_i, x_i]) \tag{D.26}$$

The detailed information of each architecture is described in Table D.1.

Table D.1: Architecture details and hyperparameters for the neural processes. Attention indicates variants of attentive neural processes used. Encoder and decoder indicate the MLP network sizes used.

| Models | MLP Encoder | Attention Encoder | MLP Decoder | Weight decay | MC samples | Rep. by functions | Information dropout | Stochastic attention |
|---|---|---|---|---|---|---|---|---|
| CNP | 3×128 | - | 3×128 | 0.001 (for *CNP_WD*) | 5 (*IWAE*) | - | - | - |
| NP | 3×128 | - | 3×128 | 0.001 (for *NP_WD*) | 5 (*IWAE*) | - | - | - |
| CNP_BA | 3×128 | - | 3×128 | - | 5 | - | - | - |
| NP_BA | 3×128 | - | 3×128 | - | 5 | - | - | - |
| ConvCNP | (U_NET) : 12 layers | - | 1×16 | - | - | *RBF kernel* | - | - |
| ConvNP | (U_NET) : 12 layers | - | 1×16 | - | 5 (For Sim2Real : 2)* | *RBF kernel* | - | - |
| ANP | 3×128 | Multi-heads : 8 | 3×128 | 0.001 (for *ANP_WD*) | 5 (*IWAE*) | - | - | - |
| ANP *(dropout)* | 3×128 | Multi-heads : 8 | 3×128 | - | - | - | ✓ | - |
| Bootstrapping ANP | 3×128 | Multi-heads : 8 | 3×128 | - | 5 | - | - | - |
| Ours | 3×128 | Multi-heads : 8 | 3×128 | - | - | - | - | ✓ |

* : In the case of ConvNP, we proceeded with 2 samples due to lack of memory.

Table D.2: The number of network parameters required for all experimental cases.

| Models | 1D regression | Sim2Real | MovieLenz-100k | Image Completion |
|---|---|---|---|---|
| CNP | 99,842 | 100,228 | 110,850 | 1,583,110 |
| NP | 116,354 | 116,740 | 127,362 | 1,845,766 |
| CNP_BA | 116,354 | 116,740 | 127,362 | 1,845,766 |
| NP_BA | 116,354 | 116,740 | 127,362 | 1,845,766 |
| ConvCNP | 50,612 | 50,655 | N/A | N/A |
| ConvNP | 51,156 | 51,199 | N/A | N/A |
| ANP | 595,714 | 596,228 | 617,730 | 9,466,886 |
| ANP *(dropout)* | 628,610 | 629,124 | 650,626 | 9,991,686 |
| Bootstrapping ANP | 628,610 | 629,124 | 650,626 | 9,991,686 |
| Ours | 597,015 | 597,529 | 619,031 | 9,472,027 |

Table D.3: Time required for inference in all experiment cases. (*1 epoch*).

| Models | 1D regression | Sim2Real | MovieLenz-10k | Image Completion : CelebA | | | |
|---|---|---|---|---|---|---|---|
| | | | | 50 | 100 | 300 | 500 |
| CNP | 0.860 | 0.896 | 0.829 | 1.274 | 1.321 | 1.390 | 1.390 |
| NP | 1.411 | 1.422 | 1.356 | 1.975 | 1.990 | 1.959 | 2.081 |
| CNP_BA | 1.143 | 1.215 | 2.149 | 1.629 | 1.644 | 2.091 | 2.221 |
| NP_BA | 1.930 | 2.003 | 3.198 | 3.074 | 3.106 | 3.709 | 4.017 |
| ConvCNP | 2.788 | 4.128 | N/A | N/A | N/A | N/A | N/A |
| ConvNP | 2.907 | 4.289 | N/A | N/A | N/A | N/A | N/A |
| ANP | 2.442 | 2.500 | 2.222 | 5.294 | 5.710 | 9.010 | 11.740 |
| ANP *(dropout)* | 3.283 | 3.328 | 2.967 | 5.961 | 6.376 | 9.926 | 12.710 |
| Bootstrapping ANP | 6.474 | 6.968 | 6.250 | 26.186 | 30.012 | 60.619 | 88.332 |
| Ours | 3.152 | 5.848 | 3.121 | 11.350 | 19.217 | 57.898 | 95.031 |

Unit : second

In this paper, we set the dimension of all latent variables as 128, namely $d_h = 128$. The number of heads in multi-head attention is 8, which is the same as the original paper of attentive neural processes (Kim et al., 2019). For all models and all experiments, we use the Adam optimizer(Kingma & Ba, 2015) with the learning rate $0.001$, and we set the number of update steps as 200000.

For training and evalation, we used AMD Ryzen 2950X(16-cores), RAM 64GB and RTX2080Ti. The 1D regressions has a batch size of 1 and a total of 160 batches. The Sim2Real configures the batch size to be 10 and the total number of batches to be 16. The MovieLenz-100k has a batch size of 1 and the number of batches to be 459. The image completion task consists of 227 batches and its size is 4.

## D.2 MODEL ARCHITECTURE

We graphically show the architecture of the proposed method in Figure D.1. This model has two types of encoder as attentive neural process(Kim et al., 2019). The encoder parameters $\phi$ consists of $\phi_1, \phi_2$. The $\phi_1$ is responsible for the local representation $r_i$ and the $\phi_2$ is responsible for global representation $z$. The encoder of global representation is same as neural process(Garnelo et al., 2018b), however, the encoder of local representation is different from the standard cross attention(Vaswani et al., 2017). After obtaining standard attention weight $w_{standard}$, we introduce reparameterization trick for $w_i$, which follows the Weiubll distribution(Fan et al., 2020). The important thing is that the key conceptual prior can be made by $\text{MLP}_{\phi_{1_3}}(\{x_j\}_{j \in C})$. The decoder is the same as the attentive neural process(Kim et al., 2019).

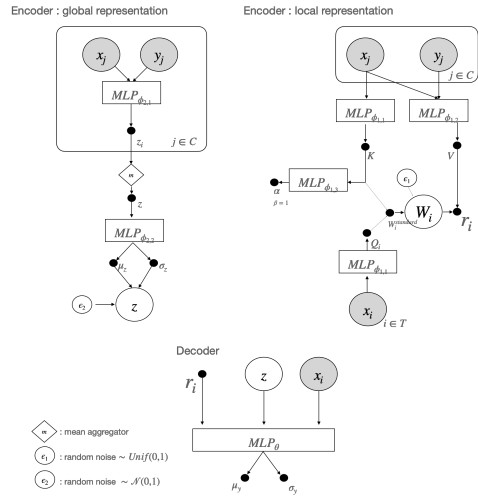

Figure D.1: Model Architectures of the proposed method

## D.3 ALGORITHM

The proposed method requires hyper-parameters $k$ and $\beta$ for reparameterization of the Weibull distribution and KL divergence between the Weibull and the gamma distribution. We conduct the grid searches for $k$ to find the best value. We identify that the proposed method with $k = 300$ adequately captures the dependency and generates noises to avoid memorization for all experiments. In the case of $\beta$, we follow the setting of the Bayesian attention module(Fan et al., 2020). We suggest the entire procedure of our algorithm as follows:

## D.4 COMPARISON OF PROBABILISTIC GRAPHICAL MODELS FOR VARIANTS OF NPS

We describe NPs with probabilistic graphical models to differentiate the variants of NPs. The Neural process employs mean aggregate function and reparameterization trick to obtain a global context representation $z$(Garnelo et al., 2018b). This model follows Figure D.2a. The Attentive neural process expedites the multi-head cross attention to obtain local representation $r_i$ with global context representation $z$. We show that the graphical model for ANP is the middle of Figure D.2, all variables in the attention mechanism are regarded as the determinstic variable.

In this work, we design that all latent variables contain stochasticity and achieves by the reparameterization trick. By Bayesian attention module, we present our graphical model as shown in Figure 2

---

**Algorithm 1:** Neural Process with stochastic attention

---

**Input:** Task distribution $p(\mathcal{T})$, the shape parameters of weibull distribution and gamma
      distribution : $k = 300$(Image completion task : $k = 100$), $\beta = 1$, Stepsize : $\gamma$

**output:** encoder parameters : $\phi = \{\phi_1, \phi_2\}$, decoder parameters : $\theta$

Initialization : $\phi$ and $\theta$ randomly;

**while** *not converged* **do**

    Sample tasks $\{\mathcal{T}_k\}_{k=1}^K$ from $p(\mathcal{T})$

    **for** *all* $\mathcal{T}_k \ni \{\mathcal{T}\}$ **do**

        Sample context dataset $(X_c, Y_c) = \{x_j, y_j\}_{j=1}^M$ and target dataset

        $(X_t, Y_t) = \{x_i, y_i\}_{i=1}^N$ from the task $\mathcal{T}_k$

        Sample random noises $\epsilon_1 \sim \text{Unif}(0, 1)$ and $\epsilon_2 \sim \mathcal{N}(0, 1)$

1         **Local representation** $\{r_i\}_{i=1}^N$ **:**

           Q, K, V $\leftarrow f_{\phi_1}(X_t, (X_c, Y_c))$

           $w_{\text{standard}} \leftarrow \text{Softmax}(QK^T / \sqrt{d_k})$

           *(Reprameterization sampling of weibull distribution for W )*

               $\lambda \leftarrow w_{\text{standard}} * \Gamma(1 + 1/k)$

               $W = \{w_1, \cdots, w_n\} \leftarrow \lambda(-\log(1 - \epsilon_1))^{1/k}$

               $W = \{w_1, \cdots, w_n\} \leftarrow \{\frac{w_i}{\sum W_i}, \cdots, \frac{w_n}{\sum W_i}\}$

           *(Parameters of the prior distribution for W )*

               $\alpha \leftarrow f_{\phi_1}(K)$

           $\{r_i\}_{i=1}^N \leftarrow \{w_i V, \cdots, w_n V\}$

2         **Global representation** $z$ **:**

           $\mu_{\phi_1}, \sigma_{\phi_1} \leftarrow \text{Aggregator}(f_{\phi_2}(\{x_j, y_j\}_{j=1}^M))$

           *(Reprameterization sampling of normal distribution for z)*

               $z = \mu_{\phi_2} + \epsilon_2 * \mu_{\phi_2}$

3         **Decode** $p(Y_t | X_t, z, \{r_i\}_{i=1}^N)$ **:**

           $\mu_y, \sigma_y \leftarrow (f_\theta(X_t, z, \{r_i\}_{i=1}^N))$

4         **Evaluation loss :**

           $\mathcal{L}_{\phi,\theta}^{\mathcal{T}_k} = \sum_{i=1}^N \left[ \log p_\theta(y_i | \mu_{y,i}, \sigma_{y,i}) - \text{KL}\big(q_{\phi_2}(z | \{X_t, Y_t\}) | q_{\phi_2}(z | \{X_c, Y_c\})\big) - \right.$

           $\left. \text{KL}\big(q_{\phi_1}(w_i | x_i, X_c) | q_{\phi_1}(w_i | X_c)\big) \right]$

    **end**

    Update $\phi \leftarrow \phi + \gamma \nabla_\phi \frac{1}{|\mathcal{T}|} \sum_{\mathcal{T}_k} \mathcal{L}^{\mathcal{T}_k}$

    Update $\theta \leftarrow \theta + \gamma \nabla_\phi \frac{1}{|\mathcal{T}|} \sum_{\mathcal{T}_k} \mathcal{L}^{\mathcal{T}_k}$

**end**

---

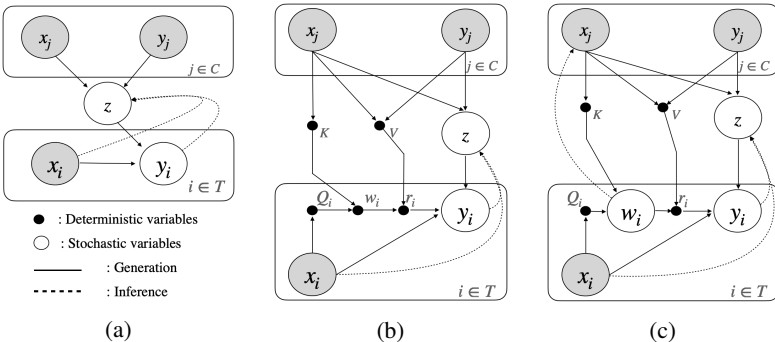

(a)             (b)             (c)

Figure D.2: Comparison of graphical models; (a) Neural process(Garnelo et al., 2018b), (b) Attentive neural process(Kim et al., 2019) and (c) Proposed method

# E   1D REGRESSIONS

For the synthetic 1D regression experiment, we set $d_x = 1, d_y = 1$. The number of layers in encoder and decoder in all baselines is respectively $l_{enc} = \{l_{pre} = 3, l_{post} = 1\} = 4$ and $l_{dec} = 3$. We set the dimension of latent variable as $d_h = 128$.

When it comes to data generation process, the training data is generated from the Gaussian process with RBF kernel. For each task, we randomly generate $x \sim Unif(-2, 2)$ and then $y$ is the function value by the Gaussian process with RBF kernel, $k(x, x') = s^2 \cdot \exp(-\|x-x'\|^2/2l^2)$. To validate our model, we establish two types of datasets in the this 1D regression experiment.

First, we set the parameters of the RBF kernel as $s = 3$ and $l = 3$. Hence, as shown in Figure E.3a, all datasets are drawn from continuous and smooth functions. Second, we consider the noisy situations in the first scheme. We intend that the suggested scheme represents the actual real-world situation. Unlike the existing studies (Garnelo et al., 2018a; Kim et al., 2019; Lee et al., 2020), we modify the RBF kernel function adding a high frequency periodic kernel function $k(y, y') = s^2 \cdot \exp(-2\sin^2(\pi\|x - x'\|^2/p)l^2)$. Lee et al. (2020) proposed a noisy situation by using random noises sampled by t-distribution; however, these noises often have exaggerated values so that the generated function does not have any tendency and seems to be entire noises. On the other hand, the function generated by Gaussian processes with a high frequent periodic kernel is smooth but is satisfied with a random function every trial. Thus, this function does not interfere with the smoothness of the RBF GP function and maintains the smoothness of all support ranges. Therefore, we decide to use the dataset generated by the RBF GP function with periodic noises to synthetically test all baselines and the proposed method for the robustness of noises. The sampled dataset is graphically presented in Figure E.3b.

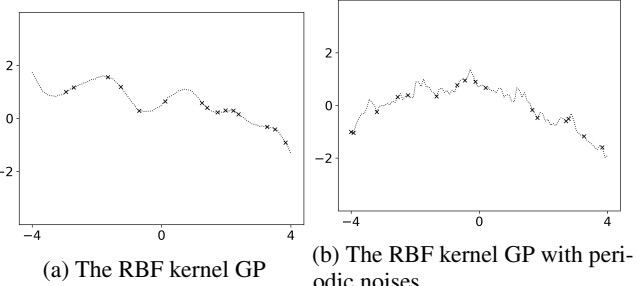

(a) The RBF kernel GP

(b) The RBF kernel GP with periodic noises

Figure E.3: Comaprison of RBF GP functions in 1D regression problems

To generate function values of the Gaussian process, the GPy library[5] provides various functions compatible for PyTorch[6]. In the GPy, $\|x - x'\|^2$ can be regarded as the pre-defined vector $\{1, \ldots, \text{freq}\}^T\{1, \ldots, \text{freq}\}$. We set the parameters of the periodic kernel as freq $= 30$, $p = 2\pi$, $s = 1$. The generated functions for training datasets are shown in Figure E.3.

To test generalization to a novel task, we introduce other functions such as the Matern32 kernel GP and the periodic kernel GP as shown in Lee et al. (2020). In this experiment, we expect the trained models to capture context points despite different functions. Naturally, all models perform well on the RBF GP function because the meta-training set is generated from RBF GP functions; instead, we test all baselines on other types of functions such as the Matern32 kernel GP and the Periodic kernel GP as well as the RBF kernel GP. For the Matern32 kernel $k(x, x') = s^2(1+\sqrt{3}\|x-x'\|)\exp(-\sqrt{3}\|x-x'\|)$ and the periodic kernel is same as mentioned earlier. The shared parameters of these three GPs are same as $s = 3, l = 3$. The periodic kernel requires another parameter freq $= 10$, which is the default value in the GPy.

To elaborate the meta-training framework, the minimum number of context points is 3, and the minimum number of target points is 6, including the context data points. The maximum number of context points and target points is respectively 97 and 100. In the following subsection, we report experimental results, including clean and noisy situations.

---

[5]https://github.com/SheffieldML/GPy

[6]https://pytorch.org

Table E.4: Results of likelihood values on the synthetic 1D regression without noises. **Bold entries** indicates the best results.

| | RBF kernel GP | | Matern kernel GP | | Periodic kernel GP | |
|---|---|---|---|---|---|---|
| | context | target | context | target | context | target |
| CNP | 1.136±0.044 | 0.762±0.090 | -0.465±0.106 | -3.684±0.222 | -2.354±0.218 | -9.612±0.0287 |
| NP | 0.687±0.030 | 0.312±0.127 | -1.222±0.055 | -3.546±0.268 | -2.252±0.115 | -6.038±0.195 |
| ANP | 1.326±0.004 | 0.819±0.023 | 0.963±0.039 | -1.538±0.083 | -0.193±0.219 | -7.447±0.278 |
| *(Weight decay; $\lambda = 1.0e - 3$)* | | | | | | |
| CNP | 1.167±0.025 | 0.827±0.051 | -0.310±0.067 | -3.135±0.178 | -2.278±0.170 | -8.840±0.259 |
| NP | 0.695±0.031 | 0.297±0.051 | -1.297±0.201 | -3.583±0.536 | -2.577±0.223 | -6.489±0.268 |
| ANP | 1.355±0.002 | 0.924±0.035 | 1.077±0.023 | -1.753±0.089 | 0.379±0.080 | -7.663±0.325 |
| *(Importance Weighted ELBO; $k = 5$)* | | | | | | |
| CNP | 1.170±0.023 | 0.803±0.083 | -0.443±0.130 | -3.704±0.296 | -2.422±0.151 | -9.639±0.330 |
| NP | 0.728±0.028 | 0.255±0.156 | -1.476±0.054 | -4.208±0.175 | -3.042±0.399 | -7.463±0.624 |
| ANP | 1.344±0.004 | 0.889±0.037 | 0.854±0.050 | -1.854±0.114 | -0.483±0.198 | -7.900±0.201 |
| *(Bayesian Aggregation)* | | | | | | |
| CNP | 1.282±0.014 | 0.978±0.019 | -0.141±0.140 | -3.443±0.294 | -2.016±0.168 | -9.703±0.221 |
| NP | 0.379±0.023 | 0.069±0.051 | -0.751±0.074 | -1.811±0.0066 | -1.465±0.092 | **-3.308±0.177** |
| *(Functional representation)* | | | | | | |
| ConvCNP | **1.351±0.002** | 0.891±0.024 | **1.336±0.008** | -1.941±0.066 | **1.020±0.062** | -9.634±0.249 |
| ConvNP | 1.318±0.012 | 0.905±0.029 | 1.287±0.015 | -1.806±0.169 | 0.953±0.057 | -9.580±0.306 |
| *(Regularization for local representation)* | | | | | | |
| ANP *(dropout)* | 1.348±0.001 | 0.866±0.029 | 0.902±0.051 | -1.753±0.084 | -0.520±0.211 | -8.052± 0.324 |
| Bootstrapping ANP | 1.347±0.005 | 0.895±0.026 | 0.790±0.044 | -1.834±0.084 | -0.983±0.309 | -8.463± 0.343 |
| Proposed Method | **1.343±0.006** | **0.937±0.040** | 1.170±0.013 | **-1.708±0.043** | 0.681±0.052 | -7.807±0.399 |

## E.1 EXPERIMENT RESULT ON RBF GP FUNCTIONS

We train all baselines on the RBF GP functions without noises. We demonstrate experimental results in terms of predictability and context set encoding.

As shown in Table E.4, all models are capable of fitting the RBF GP function; meanwhile, all models are degraded in cases of Matern kernel GP and Periodic GP. Unlike all baseline models, of which performances drop substantially, the proposed method has relatively small degradation. Particularly, the proposed method records that the likelihood value of context points at Matern kernel GP and Periodic GP is respectively 1.170 and 0.680. Meanwhile, the best performance among baselines is 1.077 and 0.379 by ANP with weight decay. This result shows that the proposed method performs better than all baselines. The graphical results are described in Figure E.4.

## E.2 EXPERIMENT RESULT ON RBF GP FUNCTIONS WITH NOISES

As mentioned early in the current section, we train all models on the RBF GP functions adding the random periodic noise. Looking at the column of RBF kernel GP in Table 1, most of the baselines are degraded due to noisy situations. However, the performances of all baselines are improved in Matern GP and Periodic GP. We guess that this phenomenon can be explained as mitigating memorization issues by injecting random noise to the target value (Rajendran et al., 2020). We present how effectively random noises improve generalization performances for all baselines and our method in this experiment. However, we can recognize that injecting random noises cannot be a fundamental solution. As seen in Appendix G, we emphasize that capturing and appropriately utilizing context information when predicting target values is a fundamental solution to avoid memorization in a given situation. The experimental result is described in Table 1 on manuscript and the detailed graphical explanation is suggested in Figure E.5.

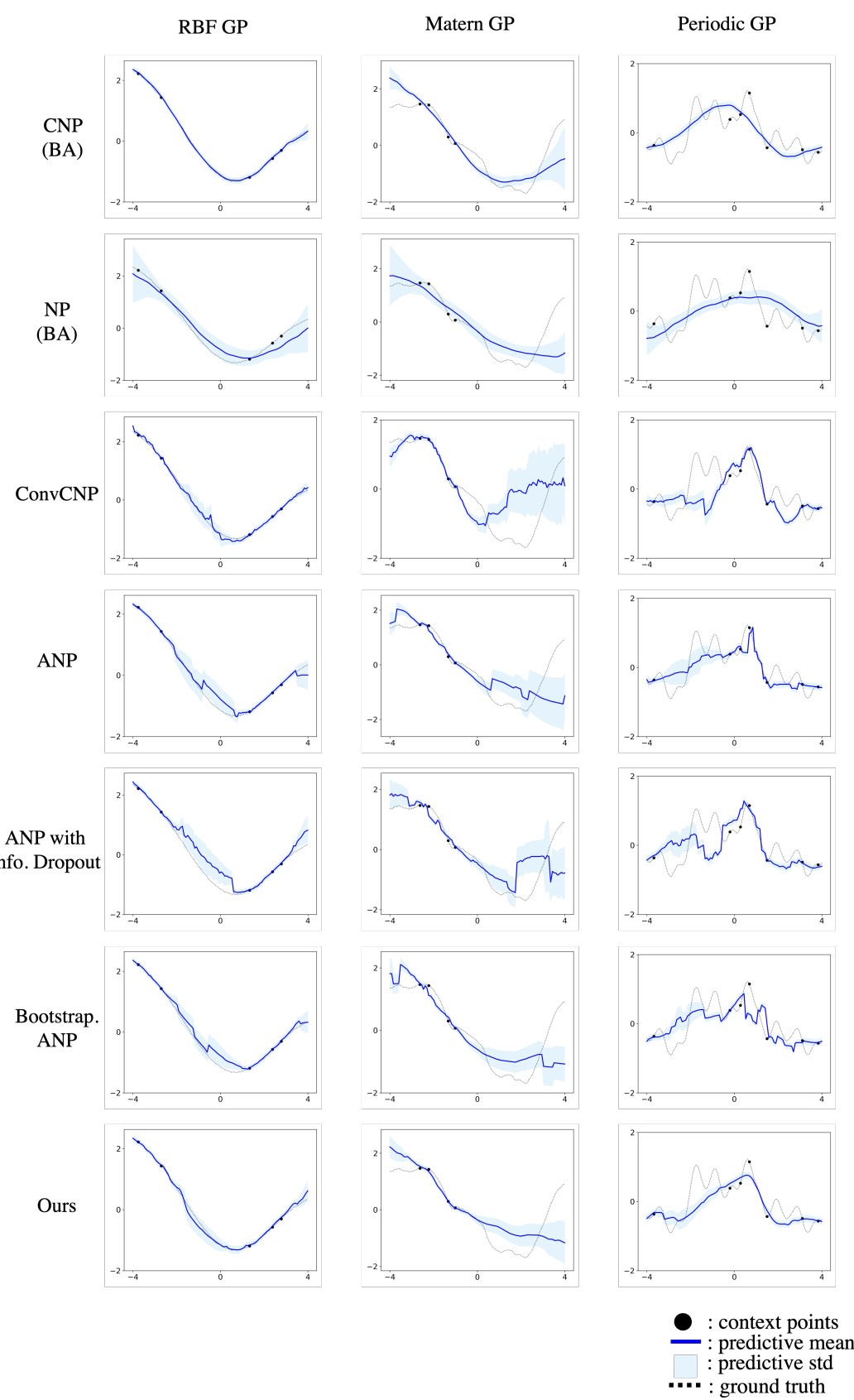

Figure E.4: 1D regression results by NPs trained on the RBF kernel function without periodic noises. All baselines and ours fit the RBF GP function, but degrade their performances in cases of Matern and Periodic kernel GP.

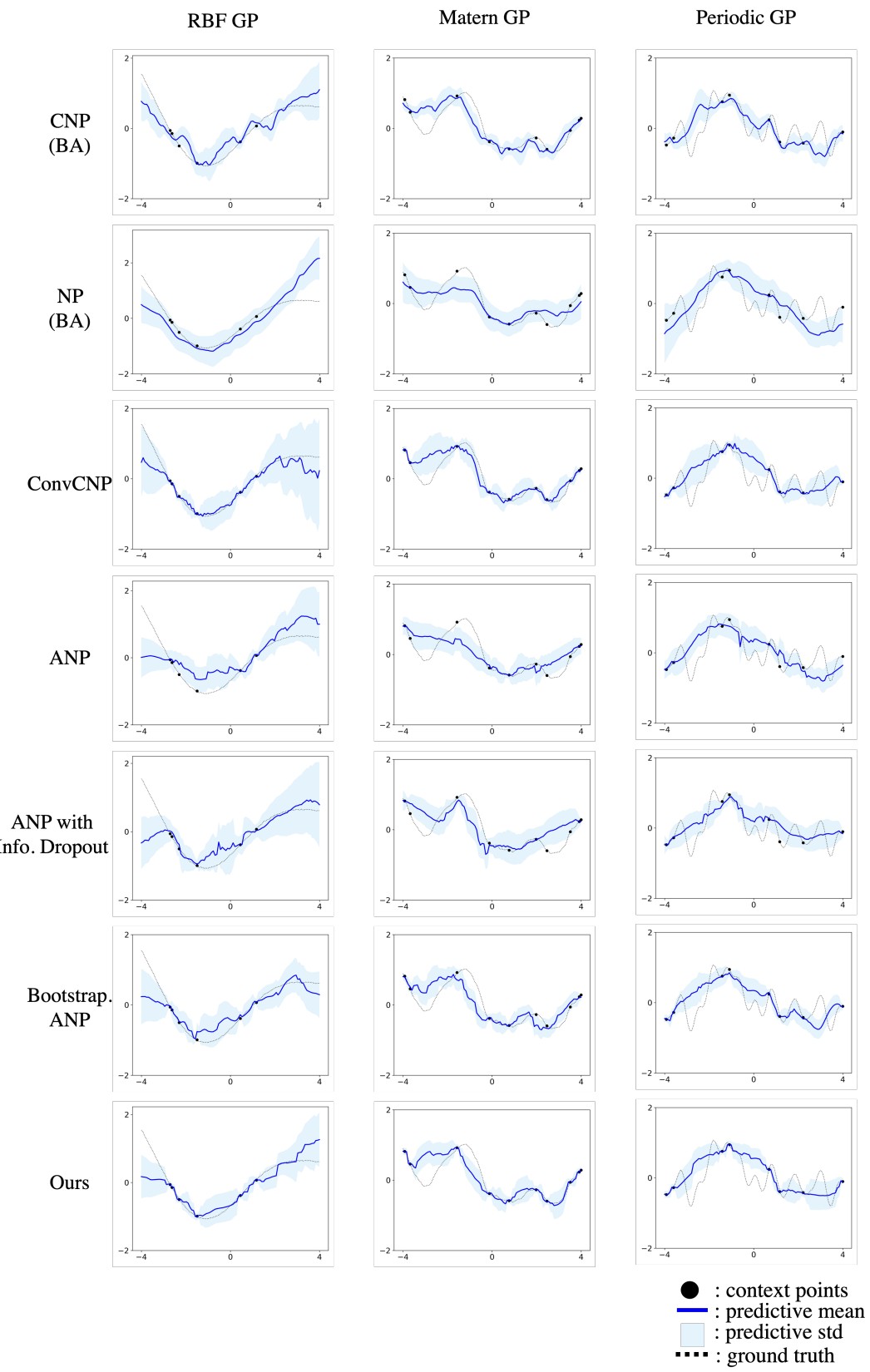

Figure E.5: 1D regression results by NPs trained on the RBF kernel function with periodic noises. All models improve performance in Matern and Periodic GP than Figure E.4. The proposed method outperforms baselines and regularization methods. Even this model achieves better performance than the result on RBF GP without noises.

# F PREDATOR PREY MODEL

We follow the experimental detail in this paper(Lee et al., 2020; Gordon et al., 2019). This experiment is designed to evaluate all baselines capable of adapting new tasks, which have slightly different task distributions. It is called "Sim2Real". We assume that, in this situation, we easily obtain simulation data, but it requires a high expense to collect real-world data points. We try to train all baselines on the dataset generated by simulations and test them on real data sets that are relatively small compared to the simulation data.

The way of simulated data is generated from the Lotka-Volterra model(LV model)(Wilkinson, 2018). Note that $X$ is the number of predators, and $Y$ is the number of prey at any time step in our simulation. According to the explanation, one of the following four events should occur. The following events require the parameter $\theta_1 = 0.01, \theta_2 = 0.5, \theta_3 = 1, \theta_4 = 0.01$

1. A single predator is born according to rate $\theta_1 \cdot X \cdot Y$, increasing $X$ by one.
2. A single predator dies according to rate $\theta_2 \cdot X$, decreasing $X$ by one.
3. A single prey is born according to rate $\theta_3 \cdot Y$, decreasing $Y$ by one.
4. A single prey dies(or is eaten) according to rate $\theta_4 \cdot X \cdot Y$, decreasing $Y$ by one.

The initial $X$ and $Y$ are randomly chosen. On the other hand, Hudson's bay lynx-hare dataset follows a similar tendency to the LV model; however, it is oscillating time series because it has been collected in real-world records. There were unexpected outliers, unexplained events, and noise. For the detailed explanation, refer Gordon et al. (2019)'s work. All simulation codes are available on this URL[7]. The generated simulation data can be graphically shown as the left side of Figure F.6.

In this experiment, we set $d_x = 1$, $d_y = 2$, and all remaining settings are the same as the 1D regression experiment except that the number of context points is at least 15 and the extra target points is also at least 15. When training models, we use the dataset generated by the LV model and test on the lynx-hare dataset. The experimental result is shown in Table 2, and the prediction performance can be graphically shown in Figure F.6.

We conduct additional experiments to validate whether the random periodic noise positively influences the performance. As conducted in the 1D regression experiment, we utilize the periodic kernel GP function, which has very high frequency freq = 100 to add noises to the training dataset. As shown in Table F.5 compared to Table 2, We recognize that adding random periodic noise empirically improves all baselines and our method. Among models, our models perform best due to the full representation of context and target datasets. The prediction results can be graphically shown in Figure F.6b

Table F.5: Predator-prey model results. All models are trained on data with periodic noises.

| | Simulation | | Real data | |
|---|---|---|---|---|
| | context | target | context | target |
| CNP | -0.181 | -0.308 | -2.420 | -2.789 |
| NP | -0.641 | -0.739 | -2.464 | -2.710 |
| ANP | 0.645 | 0.290 | -0.634 | -1.962 |
| *(Importance weighted ELBO (s = 5))* | | | | |
| CNP | -0.284 | -0.390 | -2.484 | -2.864 |
| NP | -0.527 | -0.628 | -2.335 | -2.641 |
| *(Bayesian Aggregation)* | | | | |
| CNP | -0.060 | -0.171 | -2.393 | -2.776 |
| NP | -0.357 | -0.458 | -2.260 | -2.618 |
| *(Functional representation)* | | | | |
| ConvCNP | 2.395 | 1.679 | 1.879 | -0.205 |
| ConvNP | 2.368 | 1.687 | 1.691 | -0.521 |
| *(Regularization for local representation)* | | | | |
| ANP *(information dropout)* | -0.749 | -0.990 | -1.766 | -2.458 |
| Bootstrapping ANP | 0.326 | -0.008 | -1.183 | -2.008 |
| Ours | **2.745** | **1.819** | **2.699** | **-0.076** |

---

[7]https://github.com/juho-lee/bnp

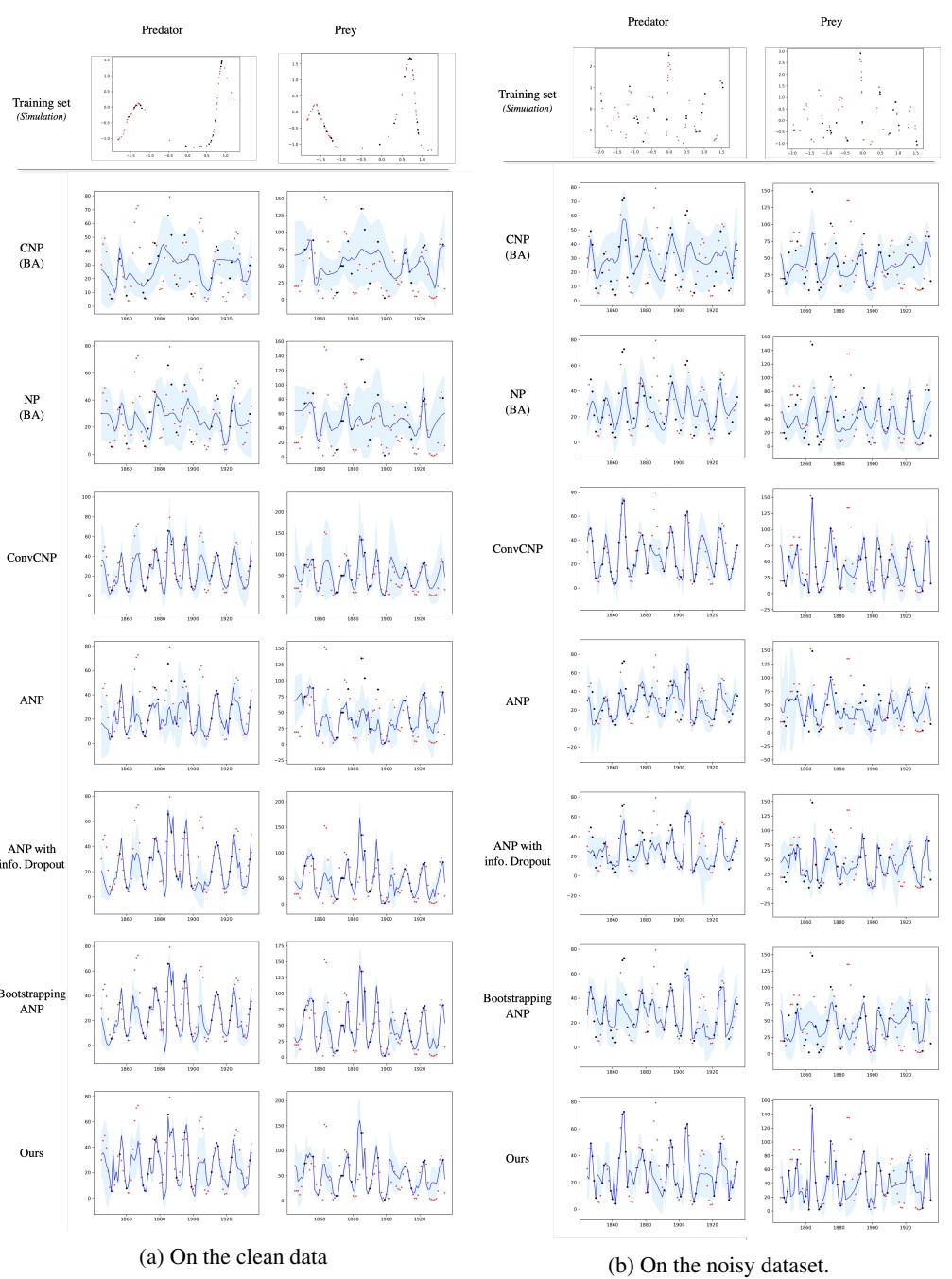

(a) On the clean data

(b) On the noisy dataset.

Figure F.6: Prediction results on the hare-lynx data. (*left*) All models are trained on the clean dataset. (*right*) All models are trained on the noisy dataset.

# G ANLAYSIS OF CONTEXT EMBEDDINGS VIA ATTENTION WEIGHTS IN 1D REGRESSIONS AND THE LYNX-HARE DATASET

The attention mechanism can explicitly display the distance between context and target data points via attention weights. Especially, when the dimension of features is one, $x \in \mathbb{R}^1$, the distance between features is simply calculated and sorted, so allowing for straightforward comprehension when the attention weights are shown in heat-maps. As a result, we present several heatmaps to compare ours with baselines in both clean and noisy 1D regression datasets. The horizontal axis in these graphs represents the value of features in the context dataset, while the vertical axis represents the value of features in the target dataset. The best pattern for this heat-map is diagonal because all feature values are arranged in ascending order. Additionally, we will provide the simplified heat-map that take into account only target points with same value in the context dataset. Due to the fact that the original version has one-hundred target points, labels and ticks are necessarily small. We guess that readers may be unable to decipher the detailed information such as labels and attention scores. This simplified version allows the readers to instantly grasp what we attempt to convey by exhibiting a more distinct diagonal pattern. From Figure G.7 to Figure G.9, on the left side, we show a simplified version of the attention score; on the right side, we show the result for the entire target dataset. Plus, we provide Sim2Real's attention heat-maps in both clean and noisy environments. As a result, our discovery appears to be consistent.

First, the heatmaps of the 1D regression problem without perioidc noises are shown in Figure G.7. In this graph, all models including ours can accurately depict the similarity between the context and targete dataset. Although the attention scores for each model vary, it is clear that the majority pattern of all models is diagonal. Therefore, all models are trained using the clean dataset in the intended manner. However, other phenomena are found in the noisy data. We identify that the attentive neural process Kim et al. (2019) fails to capture contextual embeddings because the attention weights of all target points highlight on the lowest value or the maximum value in the context dataset. In the case of the Bootstrapping ANP Lee et al. (2020) and ANP with information dropout, the quality of heat-map is slightly improved, but it still falls short of ours. This indicates that the present NPs are unable to properly exploit context embeddings because the noisy situations impair the learning of the context embeddings during meta-training. On the other hands, even in the noisy situation, the attention score in ours still appear clearly. Second, we include heatmaps for the Sim2Real problem. As seen in Figure F.6, the context and target datapoints are much too many for the label information and ticks to be recognized. Hence, we recommend that readers verify the presence of diagonal patterns rather than examining detailed numerical values. As seen in Figure G.9a, all models are capable of capturing properly similarity between the context and target datasets. However, as with 1D regression, the diagonal pattern of the baselines is disrupted as seen in the left graph of Figure G.9b; nevertheless, ours retains the ideal pattern in both clean and noisy situations.

When comparing our model's heat-map pattern in clean and noisy environments, there is a noteworthy point. Our model is capable of learning adaptively how to focus on certain context datapoints depending on the extent of dirty data in the meta-train dataset. The model is trained on the clean dataset to take into consideration nearby points as well as the corresponding point in the context dataset, hence, the heat-map gradually changes. This is because the clean dataset has smooth values, $y_{i,j}$, near a certain feature $x_{i,j}$. Meanwhile, in the noisy dataset, the model is trained to focus exclusively on corresponding points to the context dataset. Hence, the heat-map in Fig E.5 (b) indicates that attention score of target datapoints that includes the context dataset has a high value, whilst the remainder points treat all context datapoints uniformly. This phenomena occurs because there is less correlation between adjacent features $x_{i,j}$ and its labels $y_{i,j}$ in the noisy situation during the meta-training.

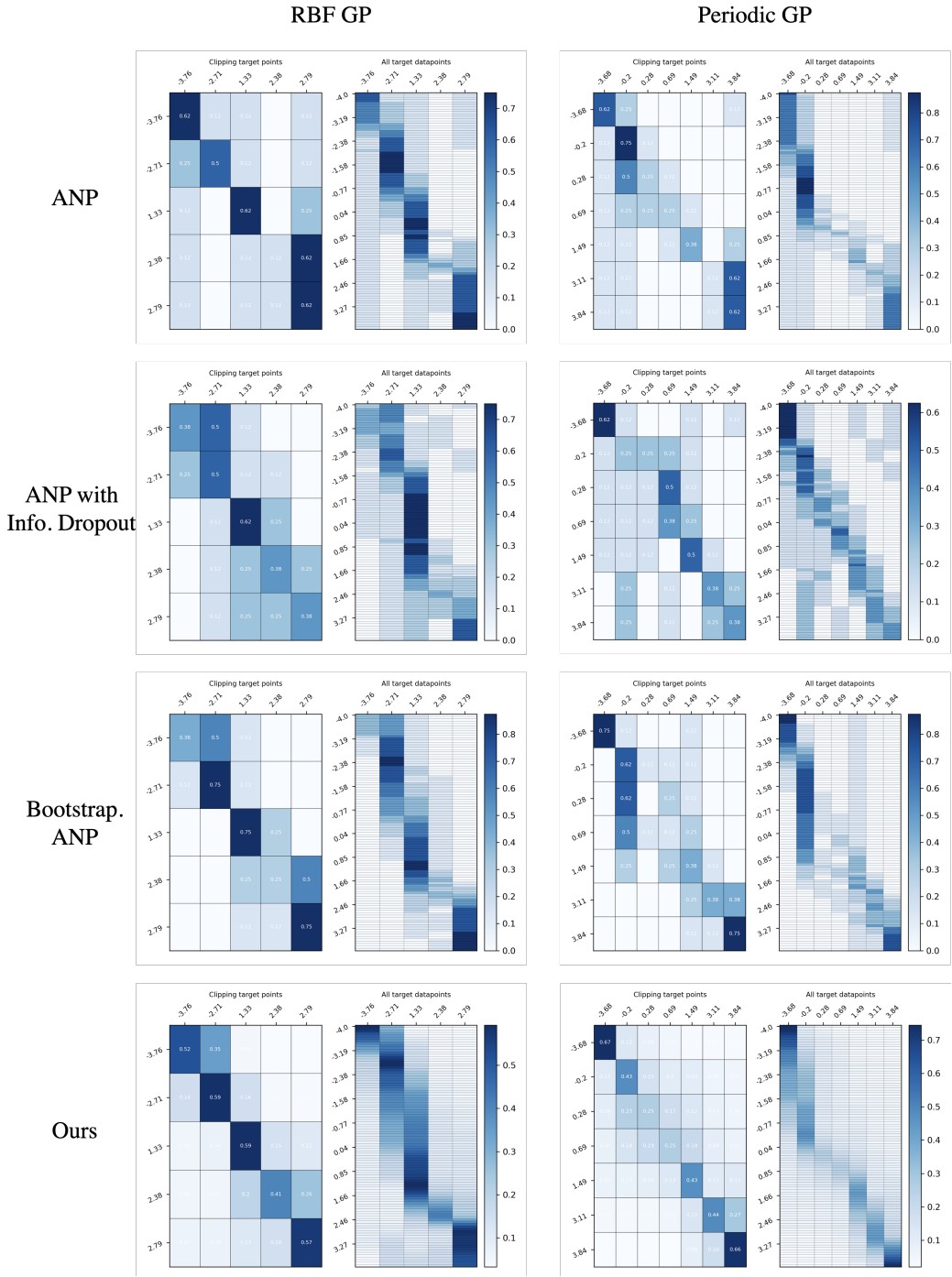

Figure G.7: Heatmaps of asscociated attention weights in 1D regression problems when test datasets are sampled from RBF and Periodic kernel functions. All models are trained on the clean dataset. (*Left*) A simplified heat-map that takes into account only target datapoints with same value in the context dataset. (*Right*) A heat-map that takes into account entire target datapoints.

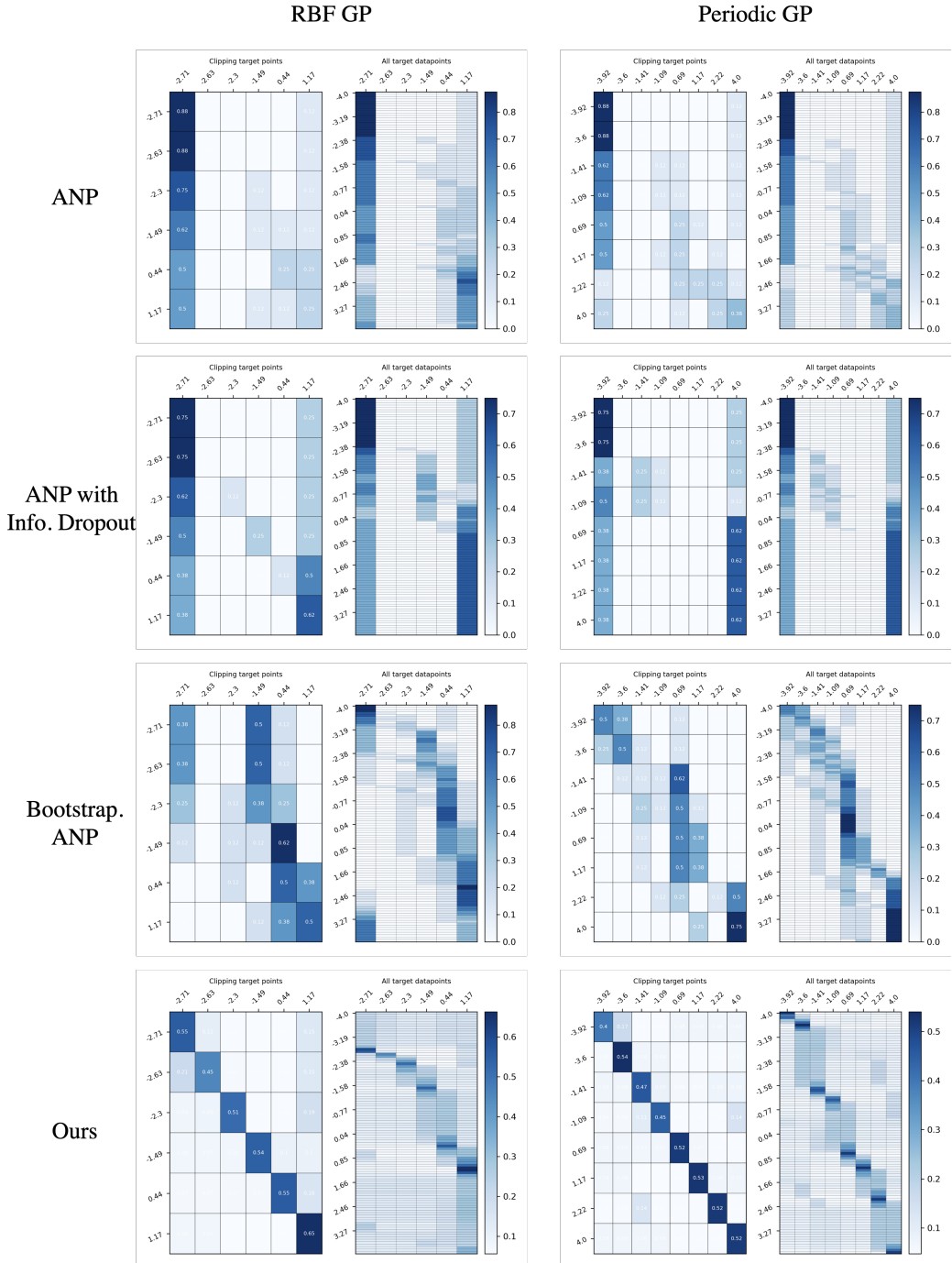

Figure G.8: Heatmaps of asscociated attention weights in 1D regression problems when test datasets are sampled from RBF and Periodic kernel functions. All models are trained on the data with periodic noises. (*Left*) A simplified heat-map that takes into account only target datapoints with same value in the context dataset. (*Right*) A heat-map that takes into account entire target datapoints.

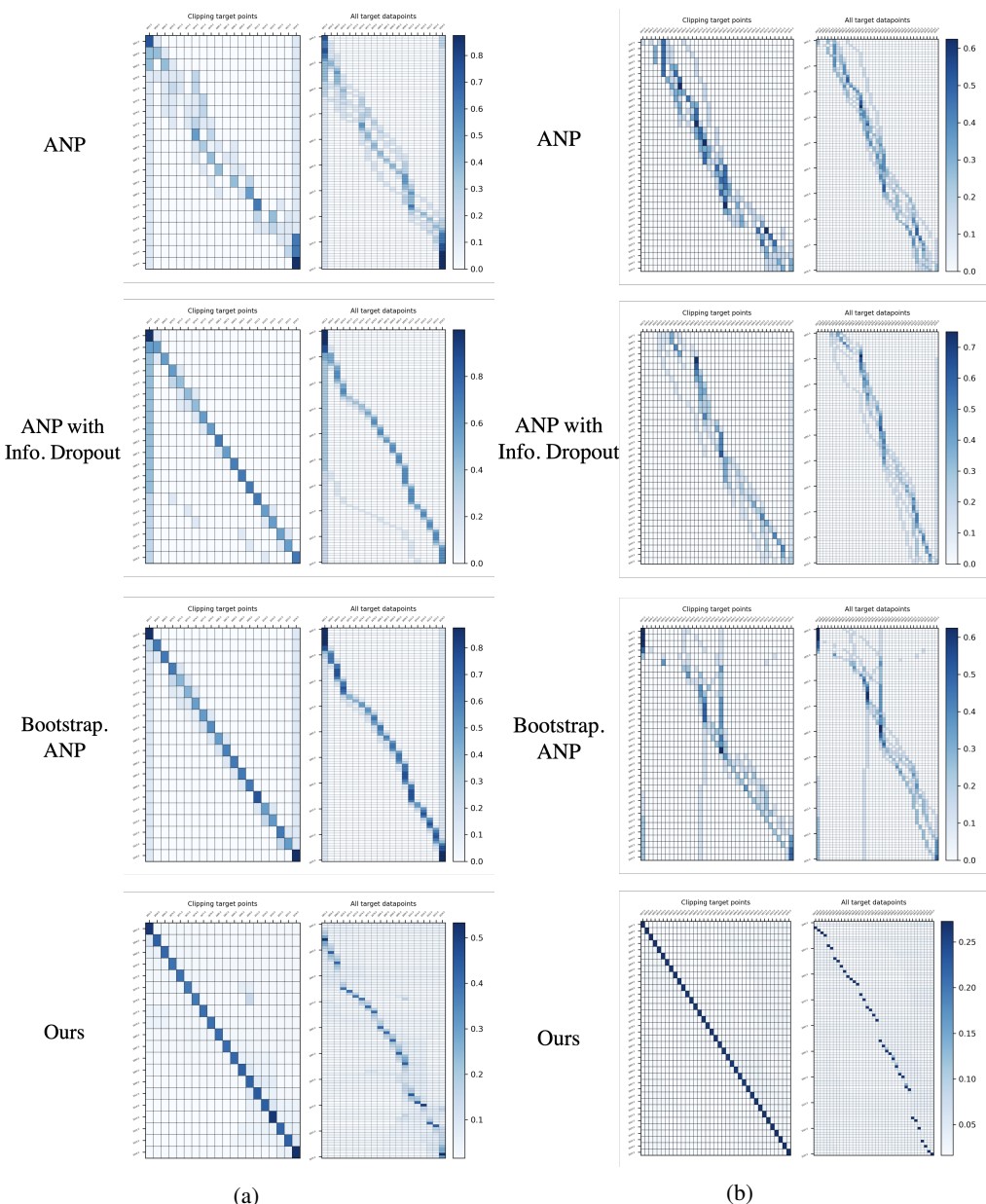

Figure G.9: Heatmaps of asscociated attention weights in the hare-lynx dataset. (*Left*) A simplified heat-map that takes into account only target datapoints with same value in the context dataset. (*Right*) A heat-map that takes into account entire target datapoints. (a) All models are trained on the clean dataset by the lotka-volterra simulation (b) All models are trained on the lotka-volterra simulation with periodic noises.

# H ABLATION STUDY : RELATIONSHIP BETWEEN REGULARIZATION OF PAYING MORE ATTENTION TO THE CONTEXT DATASET AND CONTEXT EMBEDDINGS

This section discusses the effect of suggested regularization, *paying more attention to the context dataset* on attention scores. We demonstrated how the proposed regularization term results in an increase in learning stability and embedding quality. Prior to conducting a thorough comparison of stochastic attention with and without regularization, we analyze learning curves between baselines, such as CNP, ConvCNP and ANPs in 1D regression under a noisy situation. Because the proposed regularization naturally incorporates the attention mechanism, we begin by comparing the proposed model to NPs that do not employ the attention mechanism. We choose CNP with Bayesian Aggregation, NP with Bayesian Aggregation(Volpp et al., 2021) and ConvCNP(Gordon et al., 2019) as baselines, taking into account prediction performance as demonstraed in Table 1. As shown in Figure H.10a, the negative loglikelihood of CNP_BA and NP_BA have converged to about $-10.0$ and $0.0$ However, the ConvCNP and ours record values around $-30.0$. In the case on ConvCNP, this method optimally computes similarity between context and augmented data points (called *grid*) that include target points with a high probability using the thereotically well-defined functional representation by RBF kernel function. On the other hand, the proposed regularization approaches the ideal loglikelihood value without the assistance of extra data points. The proposed can train how to establish similarity between context and target points and predict labels of target points during meta-training. Second, we compare training curves between ANP, ANP(*dropout*)(Kim et al., 2019), Bootstrapping ANP(Lee et al., 2020) and ours. This comparison indicates that while all methods make use of the attention mechanism, the chosen baselines exclude stochastic attention and regularization; $I(Z, x_i|D)$. When we look at Figure H.10b, we observe the same results as shown the quality of context embeddings in Appendix G. While the Bootstrapping ANP had superior training curves to ANP and ANP(*dropout*), but it was unable to be match ours.

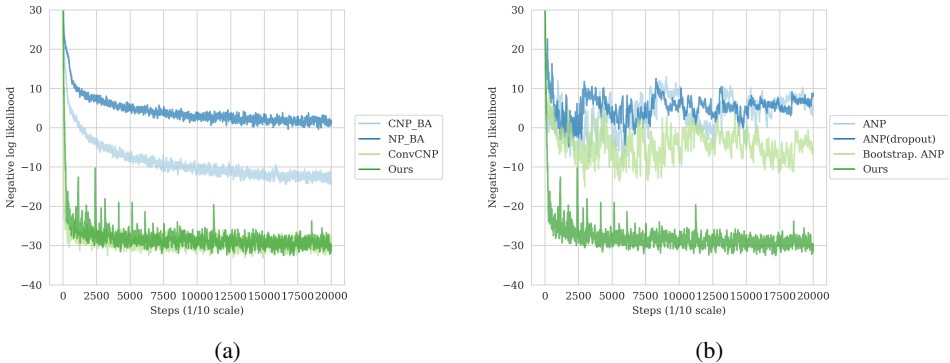

Figure H.10: Comparison of training curves with their respective baselines. (a) The baselines that do not employ the attention mechanism. (b) The baselines that employ the attention mechanism.

We examine the general effect of the proposed regularization to the stochastic attention by altering hyper-parameter $K$. We conduct experiments in which all models use same generative process for stochastic attentions, as desribed in Figure 2, but with varied hyper parameter, $K \in \{1, 10, 20, 30, 40, 50, 60, 70, 80, 100, 300, 500\}$ and the presence of $\mathrm{KL}[q_\phi(w_i|x_i, X_c)|q_\phi(w_i|X_c)]$. To begin, when we analyze Table H.6, we observe that variation of the hyper-parameter $K$ has little influence on the prediction performance of 1D regressions. If the models properly converge, the prediction outcomes should be statistically indistinguishable. However, the proposed regularization has an effect on model's learnability. While all experiments cases that employ the proposed regularization converge, a few models with the hyper-parameter $K \in \{10, 50, 70\}$ that do not use the proposed regularization, do not. Additionally, we demonstrate this fact via negative log likelihood values in the meta-training, as seen in Figure H.11. As Table H.6 indicates, all models that use the proposed regularization have converged, whereas some models that do not use the proposed regularization and use the hyper-parameter $K \in \{10, 50, 70\}$, have significantly different training

curves when compared of converged models. Therefore, we anticipate that the proposed method will ensure learnability and convergence regardless of a hyper-parameter value.

Table H.6: Prediction performance of ours varying the hyper-parameter $K$ and regularization. All models employ the generative process of stocahstic attentions. The hyphen means a model converges.

| | | No Regularization | | | | | Regularization | | | |
| | | RBF | | Periodic | | | RBF | | Periodic | |
| Models | Trainable | context | target | context | target | Trainable | context | target | context | target |
|---|---|---|---|---|---|---|---|---|---|---|
| $K=1$ | - | 1.383±0.000 | 0.421±0.008 | 1.382±0.000 | -0.322±0.028 | - | 1.376±0.001 | 0.405±0.007 | 1.376±0.001 | -0.347±0.030 |
| $K=10$ | ✗ | 0.543±0.007 | 0.199±0.009 | 0.301±0.015 | -0.645±0.030 | - | 1.383±0.000 | 0.417±0.008 | 1.369±0.002 | -0.402±0.025 |
| $K=20$ | - | 1.380±0.001 | 0.431±0.008 | 1.380±0.000 | -0.355±0.029 | - | 1.377±0.003 | 0.408±0.011 | 1.381±0.001 | -0.366±0.017 |
| $K=30$ | - | 1.381±0.001 | 0.356±0.011 | 1.380±0.000 | -0.370±0.027 | - | 1.382±0.000 | 0.452±0.007 | 1.381±0.000 | -0.423±0.026 |
| $K=40$ | - | 1.380±0.001 | 0.395±0.009 | 1.381±0.000 | -0.313±0.031 | - | 1.379±0.002 | 0.446±0.008 | 1.380±0.001 | -0.399±0.021 |
| $K=50$ | ✗ | 0.730±0.012 | 0.267±0.022 | 0.520±0.012 | -0.642±0.043 | - | 1.379±0.002 | 0.353±0.004 | 1.380±0.001 | -0.501±0.028 |
| $K=60$ | - | 1.378±0.003 | 0.433±0.007 | 1.381±0.001 | -0.460±0.029 | - | 1.380±0.001 | 0.349±0.004 | 1.380±0.001 | -0.338±0.025 |
| $K=70$ | ✗ | 0.642±0.005 | 0.235±0.010 | 0.401±0.017 | -0.644±0.029 | - | 1.373±0.001 | 0.429±0.008 | 1.377±0.000 | -0.370±0.024 |
| $K=80$ | - | 1.379±0.002 | 0.423±0.011 | 1.381±0.001 | -0.374±0.034 | - | 1.375±0.002 | 0.443±0.009 | 1.379±0.002 | -0.353±0.030 |
| $K=100$ | - | 1.375±0.002 | 0.373±0.007 | 1.378±0.001 | -0.436±0.030 | - | 1.377±0.001 | 0.457±0.009 | 1.378±0.001 | -0.391±0.034 |
| $K=300$ | - | 1.380±0.002 | 0.393±0.006 | 1.382±0.000 | -0.369±0.026 | - | 1.379±0.002 | 0.417±0.006 | 1.382±0.001 | -0.346±0.024 |
| $K=500$ | - | 1.383±0.000 | 0.359±0.008 | 1.383±0.000 | -0.316±0.030 | - | 1.381±0.001 | 0.402±0.009 | 1.380±0.001 | -0.352±0.032 |

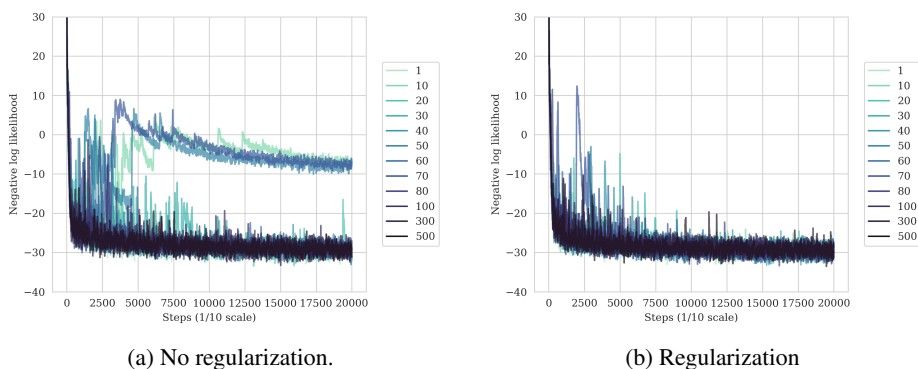

(a) No regularization.          (b) Regularization

Figure H.11: Comparison of training curves varying the hyper parameter $K$. All models employ the generative process of stocahstic attentions.

We study the learned attention scores in detail to verify the proposed regularization's efficiency. To conduct fair comparison, we fix the number of context points and choose target points that have same value as the context dataset. The ideal scenario is that all scores in the diagonal components have identical values and little variation regardless of the placement of target points as the attention mechanism should focus exclusively on context and target dataset.

As seen in the case of no regularization in Figure H.12, despite somewhat varied scores, all converged models exhibit the same diagonal pattern. As a result, these models present similar prediction performance. We argue that 1D regressions do not need precise context embeddings. However, when we compare the attention scores with and without regularization, we discover findings regarding the quality of context embeddings. In the absence of regularization, the values of attention weights changes according to the hyper-parameter $K$. Certain models place a less emphasis on the diagonal components than others, while others excessively place a greater emphasis on them. Without the regularization, the attention scores of diagonal components range from 0.3 to 0.64. We instantly notice that without regularization, the attention heat-map of stochastic attention has an inconsistent degree of brightness in relation to the hyper parameter $K$. In particular, the heatmaps of $K = 60, 100$ is too bright in comparison to other heat-maps, whilst the heatmaps of $K = 40$ is too dark. While the diagoanl pattern is presented, there is no consistency in the brightness of any heat-maps.

On the other hand, the attention scores of regularized models exhibit little fluctuation. Regardless of the hyper parameter $K$, these scores are closed to 0.5. Table H.7 contains the quantitative result. In the regularized model, the variance of attention scores for diagonal components is 0.003 and the mean value of attention scores is 0.469. Nonetheless, the mean of the model without the regularization is 0.367 and the variance is 0.026. This demonstrate that regularized models consistently

| | Diagonal | | Off-Diagonal | |
|---|---|---|---|---|
| | Mean | Variance | Mean | Variance |
| No Regularization | 0.367 | 0.026 | 0.059 | 0.001 |
| Regularization | 0.469 | 0.003 | 0.070 | 0.002 |

Table H.7: Descriptive statistics of attention scores between diagonal and off-diagonal compoenents in cases of regularization, $I(Z, x_i|D)$ and no-regularization

properly emphasize on the context dataset according to property of context and target dataset. The regularization cases in Figure H.12 illustrate this fact graphically. All heatmaps of stochastic attention that employ the proposed method regularization have very a comparable brightness. We identify that the proposed regularization ensures learning stability by ensuring the quality of context embeddings remains consistent indepedent of the hyper-parameter $K$. As a consequence, we declare that the proposed method increases model's insensitivity to hyper-parameters $K$ by preserving the context embeddings consistently.

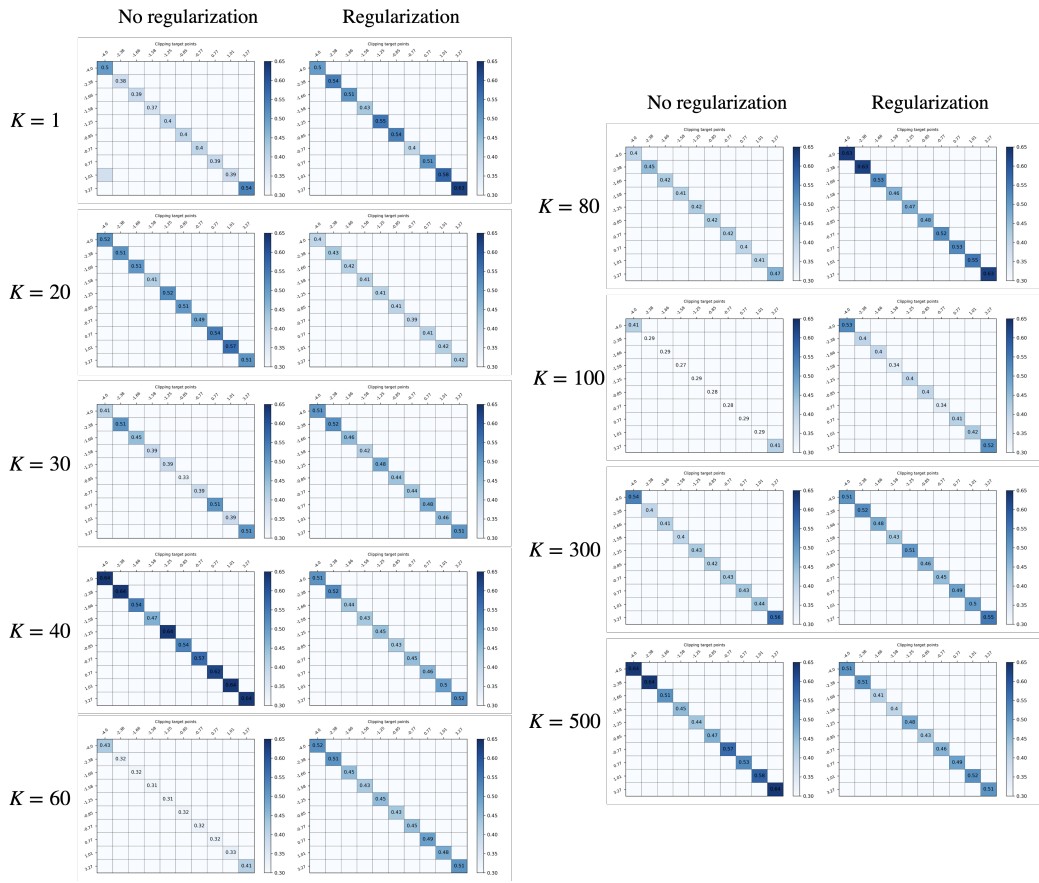

Figure H.12: Comparison of attention scores varying the hyper-parameter, $k$ and the presense of regularization, in 1D regressions drawn from the RBF kernel GP.

The intriguing fact is that the difference in attention scores for the diagonal component is rather significant within the same models. Since the values of target points are same as the context dataset, the attention scores should be constant. For instance, when $K = 40$ and no regularization is used, the attention scores are as $\{0.64, 0.64, 0.54, 0.47, 0.64, 0.54, 0.57, 0.62, 0.64, 0.64\}$. The attention scores for the same model with regularization are $\{0.51, 0.52, 0.44, 0.43, 0.45, 0.43, 0.45, 0.46, 0.5, 0.52\}$. The variance in absence of regularization is $0.003$, whereas the variance in the presence of regulariza-

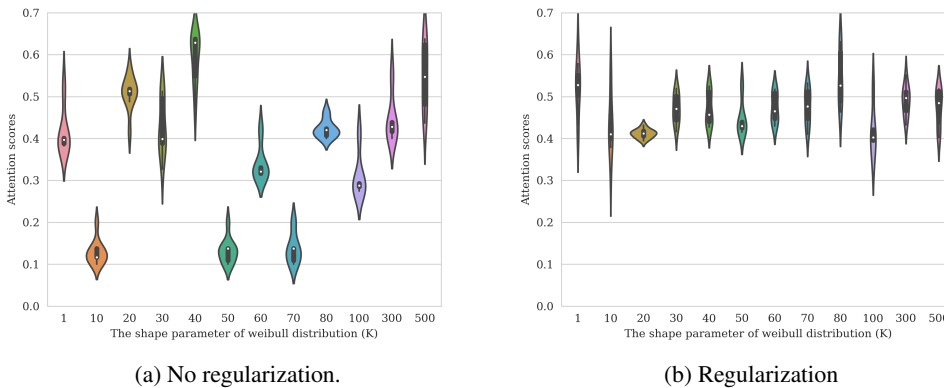

(a) No regularization.
(b) Regularization

Figure H.13: Mean and variance for attention scores of diagonal components varying the hypere-parameter $K$ and regularization.

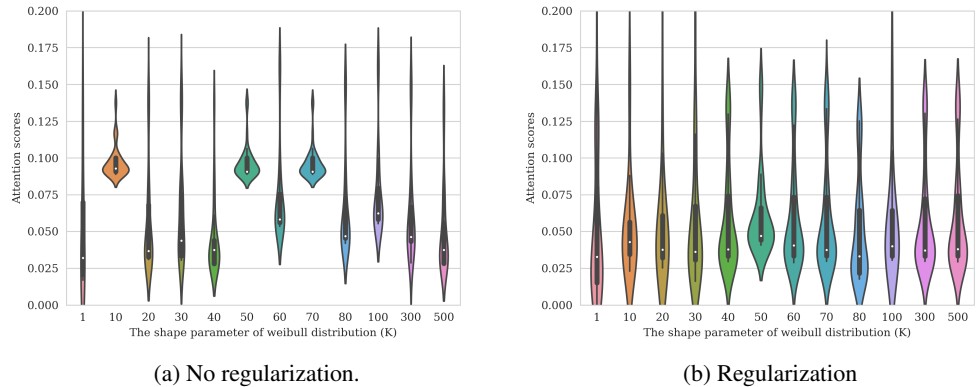

(a) No regularization.
(b) Regularization

Figure H.14: Mean and variance for attention scores of off diagonal components varying the hypere-parameter $K$ and regularization.

tion is $0.001$. As a result of the consistent scores across all target points ranges, we realize that the regularized model improves context embeddings.

The detailed analysis is shown in Figure H.14. While the mean values of attention scores vary between individuals, the proposed regularization assures that attention scores are constantly near $0.5$ value. From the perspective of NP's aim, the proposed regularization enbales models to achieve those objectives. The attention scores of all target points must not be influenced by feature magnitudes and their regions. Consequently, this method has to concentrate only on the similarity between features. The regularization aims to derive an appropriate attention mechanism that evenly weights across important regions and eliminates biases during the meta-training. It leads to achieving *task adaptation*, as it equally prioritizes context data points regardless of features magnitude. Whenever the novel arises, the regularization ensures that the models retain the quality of their context embeddings.

We investigate more research into the cases of not converged models. As seen in Figure H.15, there are completely distinct phenomena from what we have observed so far. When models are not converged, all attention scores remains constantly uniform across all region. As a result, the prediction performance degrades. However, when the models use the proposed regularization, the attention scores are comparable to the others. This means that the regularization is crucial for preventing models from trapped in the local optima. Additionally, as seen in Figure H.11, this regularization enables models can approach to maximize log likelihood values. By examining these phenomenon, it is clear that the approach proposed in this study performed as a regularization role.

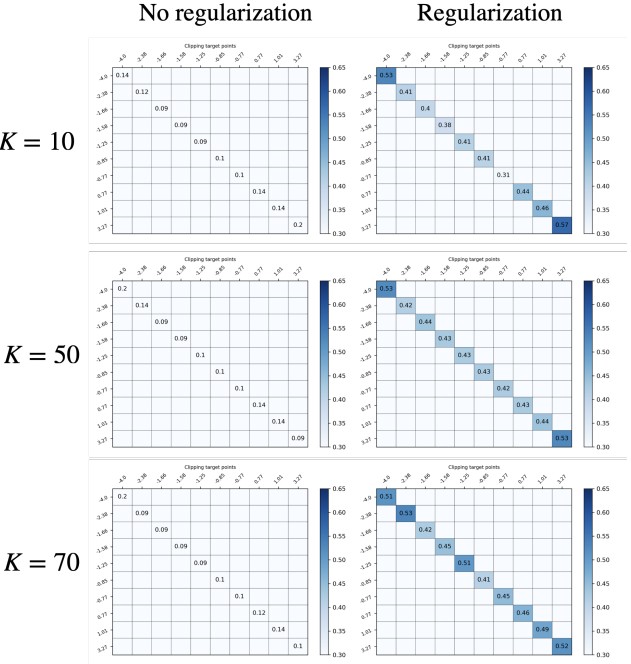

Figure H.15: Comparison of attention scores when the models without regularization are not properly trained, $K \in \{10, 50, 70\}$, in 1D regressions drawn from the RBF kernel GP. *(left)* All methods do not utilize the suggested regularization. They employ the reparameterization trick exclusively for attention scores. *(right)* All methods employ the suggested regularization.

## I  MOVIELENZ-100K DATA

We follow the experimental detail in the paper (Galashov et al., 2019). The way to split into train, test, validation dataset is described in this work[8]. Briefly, all ratings are first divided into two datasets, which respectively have user IDs and are not duplicated each other. We define that one is the meta-training dataset and the other is the meta-test dataset. In this experiment, we evaluate baselines that generalize to a new customer with small history data points. To explain a training regime, the number of context is at least three and the objective function is computed with the remaining data points. All users do not have the same amount of ratings so that the number of data points is varying during meta-training. When testing baselines, we sample a few ratings from unobserved users and then evaluate the likelihood values for the remaining points. To exclude users' unique information, we drop the User ID from the set of features and only use general features such as age, occupation. Lastly, in this experiment, we set $d_x = 44$, $d_y = 1$, and all remaining settings are the same as the 1D regression experiment. The detailed explanation is in Galashov et al. (2019)'s work. The likelihood values on the MovieLens-10k is reported in Table I.8.

Table I.8: Results of likelihood values on the MovieLens-10k. **Bold entries** indicates the best results.

|  | context | Target |
|---|---|---|
| CNP | -24.718 | -23.908 |
| NP | -23.096 | -22.135 |
| ANP | -1.456 | -2.042 |
| *(Weight decay; $\lambda = 0.001$)* | | |
| CNP | -23.524 | -23.098 |
| NP | -15.876 | -16.109 |
| ANP | -0.891 | -1.865 |
| *(Importance Weighted ELBO; $s = 5$)* | | |
| CNP | -17.276 | -17.799 |
| NP | -25.322 | -25.233 |
| ANP | -17.205 | -16.427 |
| *(Bayesian Aggregation)* | | |
| CNP | -12.472 | -12.141 |
| NP | -11.911 | -11.859 |
| *(Regularization for local representation)* | | |
| ANP *(dropout)* | -1.742 | -2.434 |
| Bootstrapping ANP | -16.267 | -16.191 |
| Ours | **-0.349** | **-1.617** |

To compare RMSE scores with existing methods for Movielens-100k data, we suggest evaluating NPs using u1.base and u1.test. As mentioned, the prediction by NPs is based on given context data points. First, we randomly choose a customer in the u1.test. Second, we randomly sample a small number of ratings from u1.base, which is used as the meta-training data. In this experiment, we decided to draw ten samples from u1.base and employ these points as the context data set. The ten ratings is relatively a short history compared to the size of the u1.base. Third, we perform predictions for the data points in the u1.test based on the context data points. The important thing is that context and target data points are sampled from the same user but the different data set. Plus, we do not touch in information in the test set, at least. Based on the relationship between sampled context points and target points, the performance of NPs is somewhat variable, so we report five runs and average these results. The experimental result is in Table 3.

---

[8]https://tinyurl.com/yyfzlg2x

## J    IMAGE COMPLETION

In the image completion task, we follow the training regime suggested by NPs(Garnelo et al., 2018a;b). In this regime, $d_x$ means the pixel location and $d_y$ means the color RGB data. Namely, we set $d_x = 2$ and $d_y = 3$. For high capacity representation, the dimension of all hidden units is determined as $d_h = 512$. All remaining settings, including layer sizes, are the same as the 1D regression experiments.

When it comes to image completion tasks, according to these papers, NPs are known to complete facial images given partial pixels. Although a few pixels are provided, NPs are capable of understanding the common appearance of a human face. These models generate eye, mouth, and nose even if only a small number of local information is given, which is generally hard to recognize. The Conditional neural process properly performs; however, it often loses its context representations. The Attentive neural process tackles this issue to employ an attention mechanism. Instead of the encoders consisting of MLP layers, the self-attention can improve the quality of pixels generated at the context data points. However, as mentioned on subsection 4.2, scores of generated pixels in unseen regions are still problems. Even though many pixels are provided, the quality of completed images is not improved. In this work, the proposed method concentrates on given contextual pixels and complete images based on that information. Hence, the proposed method completes an image with high uncertainty when few pixels are given. Plus, it can be seen that the higher the pixels information, the higher the quality of generated images. The numerical result is presented in Table 2, the graphical evidences are shown in from Figure J.16 to Figure J.19.

The proposed method allows for completion of images using only partially provided pixels, so it enables to complete larget images than the image size in the training dataset. We produce images of various sizes using the proposed method, which is trained on the CelebA dataset with size of $32 \times 32$. The completed images are presented in Figure J.20. When we qualitatively evaluate the completion results, the proposed method properly perform for generating large images without sacrificing performance. All completed images are quite close to the ground truth. As a result, we conclude that the proposed method can be used for *task adaptation* since it can complete images that the model have never encountered during meta-training and whose sizes differ from the training dataset. In particular, while the percentage of contributed pixels to the total number of pixels remains constant, the quality of completed images are enhanced according to the amount of provided pixels. We understand that the number of pixels is more significant than the ratio of context points. This is because the model has a sufficient pixel counts to work with when images are large. For instance, employing only 300 pixels as a context dataset is insufficient to complete the images. Nonetheless, the completed images with 300 pixels are comparable to those with higher pixel counts and is of higher quality than the baselines in Figure J.18.

There is an interesting discovery of uncertainty values in the image completion task. Basically, we expect that many pixels reduce the uncertainty of generated images. However, the existing methods stably generate facial images, but uncertainty values do not decrease even if available pixels increase. The proposed method makes high variance for a completed image under the small number of context points. As shown in Figure J.21, it can decrease the uncertainty of completed images when the number of context points is large. From this fact, we identify that the proposed method performs in an intended manner.

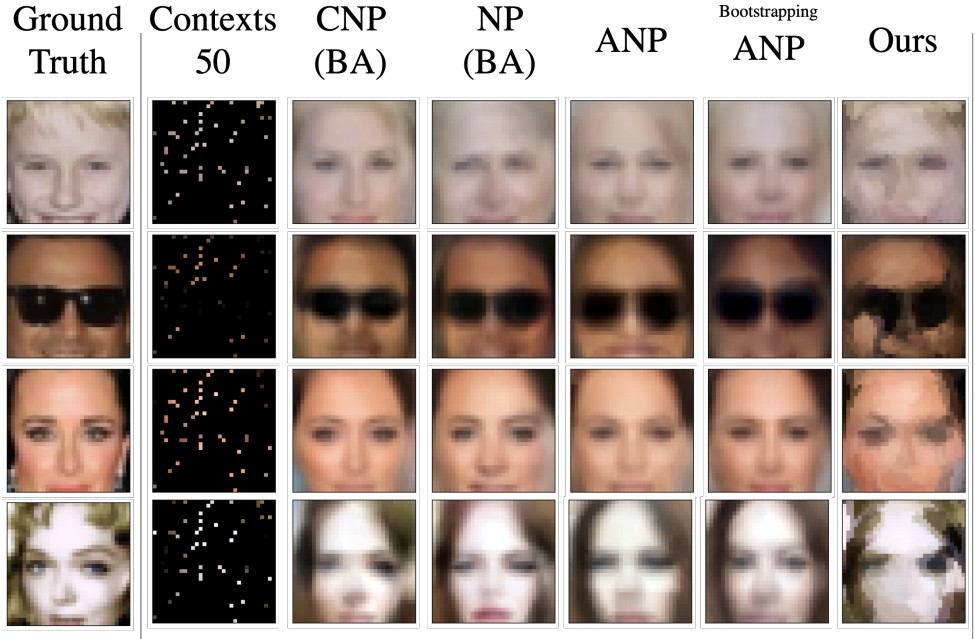

Figure J.16: Completed images using 50 pixels

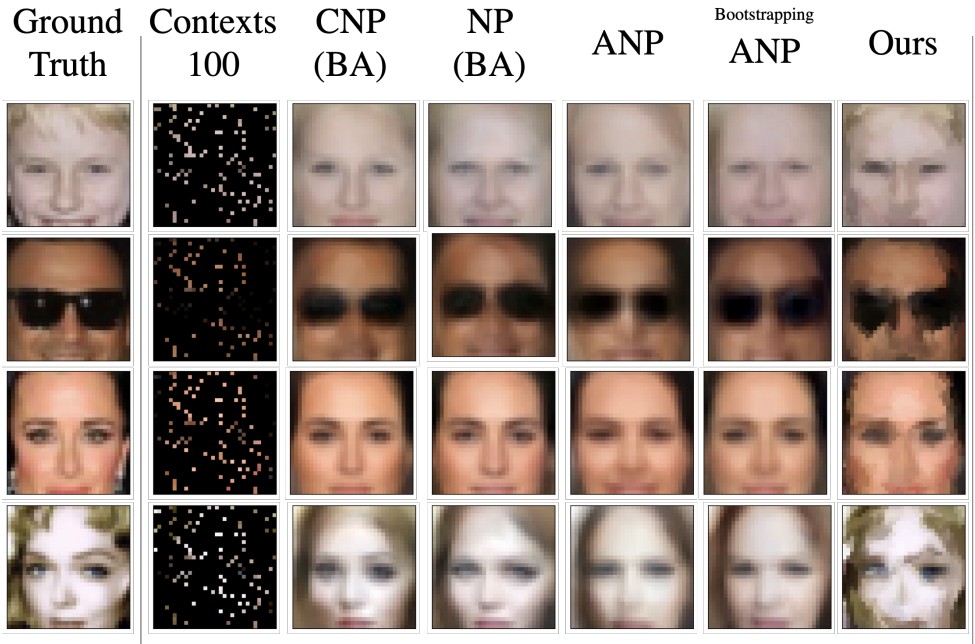

Figure J.17: Completed images using 100 pixels

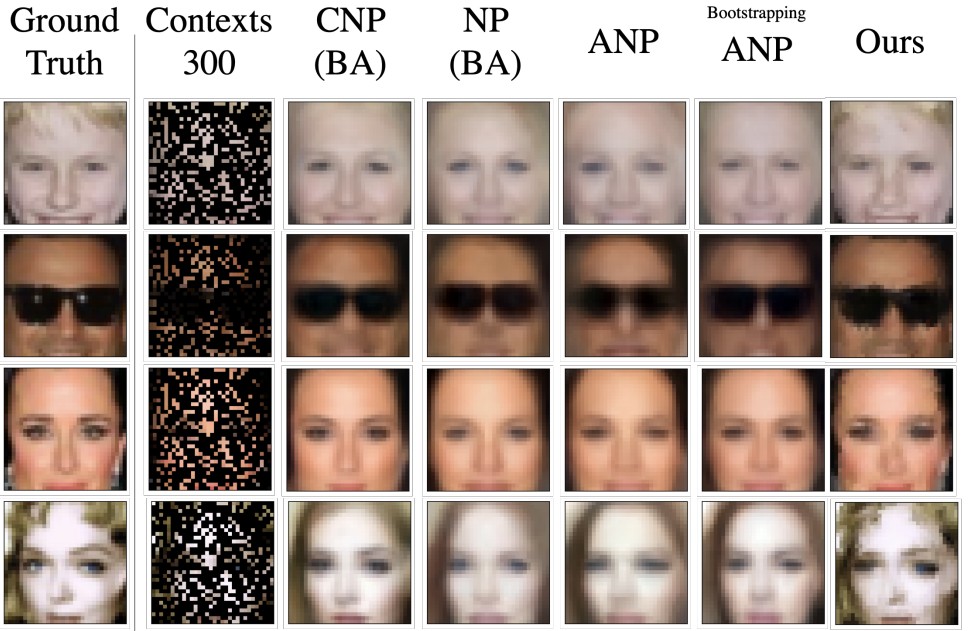

Figure J.18: Completed images using 300 pixels

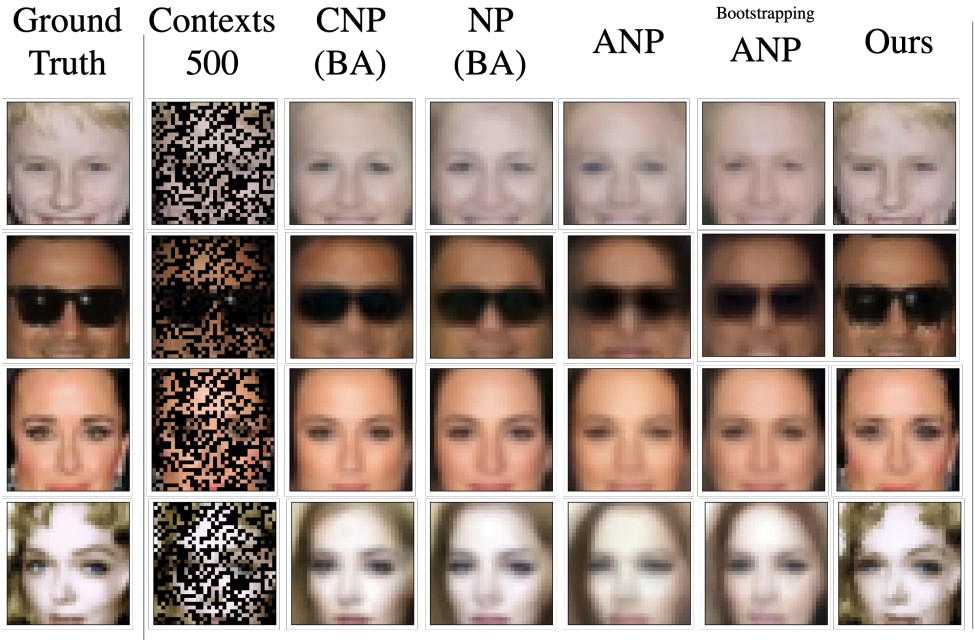

Figure J.19: Completed images using 500 pixels

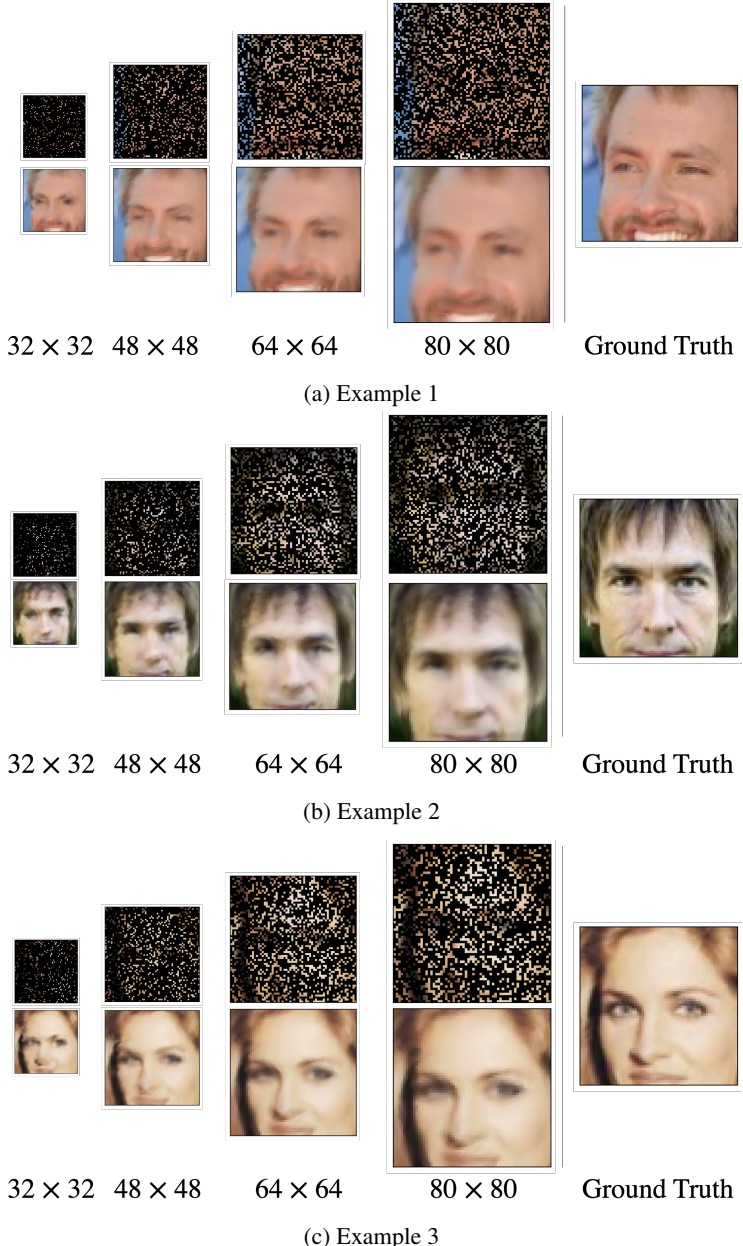

Figure J.20: Our completed images in a variety of sizes. The training dataset consists of the CelebA images with a size of $32 \times 32$. The completed images range in size from $32 \times 32$ to $80 \times 80$. Each image is completed with $0.3$ of the overall pixels. As the image size increases, the following values are used as the number of context points; $\{300, 690, 1230, 1920\}$. Both the context dataset and its completed image are displayed. Additionally the most right column has the ground truth image. Each image is shown in proportion to its image size.

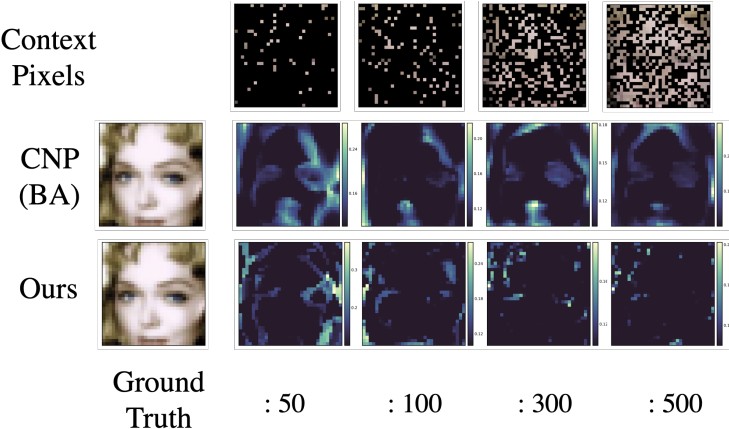

Figure J.21: Comparison of quality of uncertainty by CNP with Bayesian aggregation and Ours.

