# OpenReview forum: "Neural Processes with Stochastic Attention: Paying more attention to the context dataset"
_ICLR.cc/2022/Conference — ICLR 2022 Poster_

### Official Review · Reviewer_fqvy · 2021-10-31

**Correctness:** 3
**Technical Novelty And Significance:** 3
**Empirical Novelty And Significance:** 2
**Recommendation:** 8
**Confidence:** 4

**Main Review:**

Novelty:

In my opinion, the novelty is somewhat limited, as it is a plug-in method (the Bayesian Attention Module), to the well-established Attentive Neural Processes framework. Both methods and the regularization term are already introduced in the literature. While the paper has the merit to propose it on the very specific domain of Neural Processes, its effectiveness against newer methods is not proven enough so as to make the impact outweigh the lack of technical novelty. The development of the Bayesian Attention Module directly replicates that of Fan’s paper, including the mathematical derivation and formalisms.


Presentation:

The writing and presentation are quite inadequate and make the paper hard to be followed and understood. It requires several reads to get a glimpse of the contributions and the figures and results are quite poorly presented (the quality of the images is especially poor). In the same line, the text is more based on suggestions than on actual statements that are linked to the proven concepts, which makes the narrative rather weak. While this is not on itself a heavy point towards my personal evaluation of this paper, I believe that in either case authors should devote a good amount of time to improving the paper readability.


Experiments:

The method only compares against the simple baselines of Attentive Neural Processes and Neural Processes, and does not consider the recent advances in the topic such as the Convolutional Conditional Neural Processes and the ConvNP works of Gordon et al. and Foong et al. These works tackle the problem of domain shift with functional representations and seem to improve over the baselines in a big extend. The authors are encouraged to test their method against the newer NP families to prove the effectiveness of the proposed approach.

I would like to see an analysis of the learned distributions w.r.t. the “distance” between the context and the target points, with a comparison w.r.t. the weights that would be computed by a deterministic attention module, which would act as a baseline.

Other comments:

- It is my understanding that at inference time the weights for the attention module are drawn from the Weibull distribution, meaning that for each query (x_i in the target) there will be N keys (x_i in the context) and hence N weights. In Eqn. (6), q_phi represents that distribution, which is conditioned to x_i on the target, and X_c and Y_c on the context. This seems to indicate that the parameter lambda is determined by an MLP whose inputs are x_i, X_c and Y_c. However, I do not see the role of Y_c on computing such weights, as the text seems to refer to the fact that the weights for x_i, X_c will be computed probabilistically according only to these values, which represent the weights to compute the value r_i for x_i as a linear combination of the r_i in the context, which are now represented by X_c AND Y_c. However, why is Y_c necessary to obtain the KL loss in Eqn. 6? Are the weights computed from x_i on the target and the r_i on the context? Please do clarify as I might have misunderstood something here.
- One of the main problems of NPs is the domain shift, which occurs e.g. when the inputs are farther from the context set. NPs compute the latent variable independently of the target points i.e. they do not account for the distance between the context and the target points. ANPs partially solve this problem, as well as the ConvCNP and ConvNP family. In this paper, this problem is not analysed, and indeed the motivation seems to lie on the idea of making the attention weights be more focused on the context set rather than on the distance between the context and target sets. Again I might be misunderstanding something here, but I would like to ask the authors to elaborate on the similarities and differences w.r.t. the recent works aforementioned. How would the proposed method work in the image completion experiment for images of larger size than those used to train (e.g. 32x32 to 64x64).


- The NP loss is formulated in a similar fashion to Variational Autoencoder, where z is computed during training from the target points. However, as pointed out in Foong et al. this regularization term might not be needed and indeed one can train the Neural Process without it by conditioning z only on the context points. Such approach improves the expressiveness of the NP family. I wonder how would the proposed method perform in such case, i.e. without the regularizations and by computing the latent distributions for the global and local variables directly from the context points only. This should be a necessary experiment towards understanding whether the mutual information gap needs to be maximised, or not. The paper does not provide any experimental analysis on the two terms of the mutual information mentioned in Section 3.3. Proving that this gap is an upper bound might not necessarily entail that indeed I(Z,x_i|D) is being minimized.

In summary, I believe the paper holds an interesting research direction worth exploring in the context of Neural Processes. However, in my opinion, the current manuscript does not meet the standards for acceptance at ICLR 2022. The authors are strongly encouraged to address the limitations and improve readability, as I do believe this is a promising idea.


**Summary Of The Paper:**

The paper proposes a novel method for the Attentive Neural Processes paradigm by adding stochasticity to the weights of the cross-attention module between the context and target representations. These weights are drawn from a proposal distribution which is a Weibull distribution with parameters determined by the context and target input points. A new KL regularization term is added to the total loss to enforce the proposal distribution be close to the gamma distribution determined only by the context points, in a similar fashion to the regularization term for the latent representation between the whole set of target and context points and that of the context points only. The authors include, from Fan’s paper, the closed-form KL divergence between the Weibull and Gamma distributions, making the whole loss be differentiable w.r.t. the network parameters thanks to the reparametrization trick, as in the original NP paper. The new loss is linked to the gain of the target distribution by means of information theory, by making the assumption that the new regularization implies not only that the mutual information between the target distribution and the target points is maximized, but also the mutual information between the representation for the context and target points and the latent representation of the context variables only is minimized, enforcing the latent variable to also consider the context points, and not otherwise as suggested by the original NP formulation.

**Summary Of The Review:**

The paper develops on a promising idea towards incorporating stochastic attention weights to the Attentive Neural Processes framework. However, the current manuscript is far from being ready for its publication.


--- Post rebuttal

After interacting with the authors and considering the comments and revisions, I am happy to raise my score. The paper can make a good contribution to ICLR 2022.

---

> ### Author Response · Authors · 2021-11-14
> **Response to fqvy's review (5/5)**
>
> ### Presentation
>
> **Enhance the quality of figures and provide more figures to substantiate our originality** : In the revised manuscript, we increased the size of all figures. We recognize that the reader may miss our academic contribution under the initial manuscript because all heat-maps are too tiny to comprehend the meaning of axes, labels and ticks. We are revising the manuscript to increase the scale of each heat-map and add this analysis in the appendix. Additionally, we will provide a simplified heat-map that takes into account only target points with the same value in the context dataset. Due to the fact that the original version has one-hundred target points, labels and ticks are necessarily small. As a result, readers may be unable to decipher detailed information such as labels and attention scores. By minimizing the number of target points, we are able to present the simplified version. We believe that the readers will instantly grasp what we convey by exhibiting a more distinct diagonal pattern. Plus, we will provide Sim2Real's attention heat-maps in both clean and noisy environments. As a result, our discovery appears to be consistent.
>
> Notably, we modified the introduction and clarified the unclear wording based on recent reviews. We amended Fig 1 for readers to instantly comprehend our contribution in the introduction; noisy situations degrade context embeddings. By clarifying the ticks, axes, and labels on the heat-maps in Fig 1, as well as all figures of the revised manuscript, the reader will be less likely to be confused about our contribution. Additionally, we have made an effort to improve the readability of this paper by having it reviewed by proofreading service. During this discussion period, we continuously proofread and fix ambiguous terms.
>
>
> **[Reference]**
>
> 1. Lee, Juho, et al. "Bootstrapping neural processes.", NeurIPS2020
> 2. Volpp, Michael, et al. "Bayesian Context Aggregation for Neural Processes.", ICLR2021
> 3. Yao, Huaxiu, et al. "Improving generalization in meta-learning via task augmentation." ICML2021
> 4. Yin, Mingzhang, et al. "Meta-learning without memorization." ICLR2020
> 5. Rajendran, et al. "Meta-learning requires meta-augmentation." NeurIPS2020
> 6. Fan, Xinjie, et al. "Bayesian attention modules." NeurIPS2020
> 7. Gordon, Jonathan, et al. "Convolutional conditional neural processes." ICLR2020
> 8. Foong, Andrew YK, et al. "Meta-learning stationary stochastic process prediction with convolutional neural processes." NeurIPS2020

---

> ### Author Response · Authors · 2021-11-14
> **Response to fqvy's review (4/5)**
>
> ### Background
>
> **Describe the similarity and difference between this study and previous research** : As the reviewer fqvy mentioned, the neural processes(NPs), one of the meta-learning algorithms, is proposed to improve generalization performance using a given context dataset, called *Task adaptation*. The purpose of NP is to use embedding $Z$ to derive an identifier suitable for a novel task. Therefore, the previous studies have been concentrated on how to make appropriate context embeddings even for a novel task. The significance of NPs research is that the meaning of each component is explained in terms of the model's purpose, meanwhile, implementations of existing neural processes, including ours, are straightforward. The ANP shows why attention is required to capture context information in a clean dataset, and ConvCNP[7] and ConvNP[8] explain why a convolutional network is required for domain shift due to their translation invariant features. From this point, our study makes an adequate academic contribution. This is because we propose how context embedding should prioritize larger context datasets in order to preserve embedding quality in noisy environments.
>
> 1. Comparison of our study and ANP (main baseline)
>
> The vanilla neural processes(NP) and conditional neural processes(CNP), which are composed of basic MLP structures, are limited in their ability to capture context embeddings effectively. To improve embedding capacity, the attentive NP(ANP) incorporates an attention mechanism. It is frequently used in large datasets such as natural language processing and computer vision. Regardless of how noisy a dataset is, irreducible noises may be ignored due to the law of large numbers. The authors of the ANP demonstrate significant improvement in 1D regression and image completion, such that this approach can be considered as the baseline of current NP studies. However, we notice that the vanilla attention does not perform as intended in few-shot and even worse in noisy conditions. Additionally, because the ANP employs a deterministic function attention mechanism as a part of context embedding, it deviates from the principle of amortized variational inference. As a result of these considerations, we believe that the ANP still falls short of the goal of NPs, *Task adaptation*.
> Therefore, we seek to conceptually imply, using information theory, that NPs satisfy the aim even in the few-shot regime. According to the above statement, we design a novel method that adheres to the principle of amortized variational inference without sacrificing the performance of context embeddings by introducing the Bayesian Attention Module with key based prior. The novelty is noteworthy that we theoretically establish for the first time the meaning of key based prior in terms of the purpose of NPs for the first time.
>
> 2. Regarding to a family of ConvNP and recent enhancement to NPs
>
> The ConvCNP and ConvNP are attempting to increase extrapolation capabilities, as NPs degrade in areas not covered by training datasets. ConvCNP and ConvNP handle this issue by utilizing good deterministic functions such as the RBF kernel and power function as context embeddings, as well as convolutional neural networks as a decoder to ensure translation invariance. They introduce the Sim2Real problem we have also addressed, as a means of verification.
> The bootstrapping approach is developed to be orthogonally applied to the NPs for enhancing robustness to noise. However, the authors of bootstrapping NP tested their models on noisy datasets when all models are trained on clean datasets. For CNP and NP, the bayesian aggregation function is used instead of the average aggregation function. It is designed to comprehend the importance of each task.
> In summary, previous studies have not shown whether context embeddings are achievable in noisy situations seen in real world problems. Hence, we identify that the previous methods have impaired the quality of context embeddings when a noisy dataset is used instead of a clean dataset as the meta-train dataset.
>
> For a family of ConvCNP, we reported the experimental results in the revised manuscript. (refer to the 2nd response)

---

> ### Author Response · Authors · 2021-11-14
> **Response to fqvy's review (3/5)**
>
> ### Experiments
>
> **Distance between the context and target points** : The initial manuscript already contains attention weights between context and target data points as seen in Fig 1(a) and Fig E.5. As the reviewer fqvy mentioned, the attention mechanism can explicitly display the distance between context and target data points via attention weights. Especially, when the dimension of features is one, $x \in \mathbb{R}^1$, the distance between features is simply calculated and sorted, allowing for straightforward comprehension using the attention heat-maps.
>
> We have increased the size of heat-maps and included clear figures in the revised manuscript in response to your comment. For your convenience, we will refer to the number of figures based on the revised manuscript in the following response. Fig 1 and all figures of Appendix G represent a comparison of baselines and ours in both clean and noisy 1D regression datasets. The x-axis in these graphs represents the value of features in the context dataset, while the y-axis represents the value of features in the target dataset. The best pattern for this heat-map is diagonal because all feature values are arranged in ascending order.
>
> Fig G.8 tells us that all methods exhibit identical patterns like the ideal case when trained on a clean dataset; nevertheless, when all models are trained on a noisy dataset, we see distinct phenomena as illustrated in Fig G.9. The attentive neural process(ANP) fails to capture contextual embeddings because the attention weights of all target points highlight the lowest value or the maximum value in the context dataset. In the case of the Bootstrapping ANP, the quality of heat-map is slightly improved, but it still falls short of ours. Even in the noisy situation, the attention weights of target data points in ours still appear clearly.
>
> When comparing our model's heat-map pattern in clean and noisy environments, there is a noteworthy point. Our model is capable of learning adaptively how to focus on certain context data points depending on the extent of dirty data in the meta-train dataset. The model is trained on the clean dataset to take into consideration nearby points as well as the corresponding point in the context dataset, hence, the heat-map gradually changes. This is because the clean dataset has smooth values, $y_{i,j}$, near a certain feature $x_{i,j}$. Meanwhile, in the noisy dataset, the model is trained to focus exclusively on corresponding points to the context dataset. Hence, the heat-map in Fig G.9 indicates that the attention score of target data points that includes the context dataset has a high value, whilst the remaining points treat all context data points uniformly. This phenomenon occurs because there is less correlation between adjacent features $x_{i,j}$ and its labels $y_{i,j}$ in the noisy situation during the meta-training.
>
> **About ConvCNP and ConvNP as baselines** : The main concern of this paper is to ensure that neural network-based context embedding is learned in noisy environments. However, in the 1D regression example, the ConvCNP and ConvNP additionally employ deterministic functions such as the kernel function for context embeddings. They retained translation invariance by using convolutional networks. We did not consider a family of ConvCNPs as baselines because we concentrated primarily on context embeddings via MLP structures.
>
> As the reviewer fqvy concerned, we tested ConvCNP on the noisy 1D regression, Sim2Real and Movielenz-10k dataset. These results were included the revised manuscript, Tab 1,2,E.4 and F.5 (refer to the 2nd response).
>
> **About the role of key based prior** : To further understand the function of key-based prior, we suggest an information dropout model for the ANP. This approach makes use of a drop-out mask derived from amortized variational inference to a local presentation, $r_i$. At this point, the prior distribution employed is the standard normal distribution, $N(0,1)$. Consequently, the majority of experimental results show that this model does not improve performance except for the Sim2Real problem. In terms of context embedding quality, we also notice that ANP with information dropout is unable to adequately capture the similarity between context and target datasets in Appendix G. We recognize the need of a well-defined stochastic local representation like the key-based prior.
>
> We additionally evaluate the proposed method in the absence of the regularization term; KL loss for the attention weights in order to see whether this model supports the Theorem1. These experimental results are included in the Appendix H (refer to the 2nd response).

---

> ### Author Response · Authors · 2021-11-14
> **Response to fqvy's review (2/5)**
>
> ### Novelty
>
> **Noisy environment impairs the embedding of context dataset in existing neural processes** : The main concern of this paper is to examine how neural processes (NPs) learn to acquire suitable contextual embeddings in noisy situations that have received little attention. The preceding study, the Bootstrapping Neural Process[1], deals with training models on a clean dataset and testing them on a dirty dataset to demonstrate their model's robustness. However, no one has investigated whether NPs can be trained to acquire suitable contextual embeddings given dirty datasets. We discover through experimental results that existing NPs suffer from meaningless context embedding, hence, all existing methods do not perform well prediction in noisy situations. While modern approaches such as Bayesian Aggregation[2] and Bootstrapping[1] have marginally improved performance in 1D regression and Sim2Real experiments, these gains are not consistent across all experimental cases, and these cannot, therefore, be considered solutions.
>
> The initial manuscript includes Figure 1 and Figure E.5(b) to substantiate this statement. We reflect your comment about image quality, so we modified the manuscript to increase image size in order to visually inspect the quality of context embedding. The attention heat-maps for 1D regression and Sim2Real (the hare-lynx dataset) should ideally be a diagonal pattern because the provided features, $x_i \in \mathbb{R}^1$ are ascendingly ordered in both context and target dataset. While heat maps of the existing attentive neural processes are chaotic, the proposed method appears to have a diagonal pattern, indicating that it is far better suited to the ideal case. The clear figures are provided in Appendix G of the revised manuscript.
>
> **New metrics for NPs to be followed for proper context embedding via information theory** : As reviewed by uH8j and DEA4, to address this issue, we claim for the first time, using an information theory framework, that critical conditions for contextual embedding in NPs are independent of target features and close to contextual datasets as seen in Theorem 1. Previous studies[3,4,5] have dealt with information theory setups to reveal memorization phenomena under limited task sizes in the meta-training, meanwhile, nobody has investigated the points at which NPs retain their performance and quality of contextual embedding in severe situations. From this perspective, this work has a significant trajectory in NP domains.
> We theoretically and empirically prove that the proposed algorithm supports this statement. We demonstrate that the Bayesian Attention Module Address with key based prior[6] addresses this issue by leveraging amortized variational inference for attention weights in order to encourage NPs to pay more attention to the context dataset. Therefore, we empirically identify facts that the proposed method consistently improves predictive performance as well as the quality of contextual embeddings, especially in noisy environments.
>
> **Difference from previous works, the Bayesian Attention Modules[6]** : The Bayesian Attention Modules, which is meaningful to propose a concept of fully amortized inference for attention mechanism, examines only the role of the key-based prior via empirical experiments. Regrettably, the experiments of graph embeddings and language models show just improvement in comparison to traditional attention. In addition, there is no theoretical justification about the role of key based prior.
>
> On the other hand, as noted before, this paper establishes that the loss function and all components are constructed to fulfill Theorem 1. Improvement in the quality of contextual embeddings using several heat-maps as ablation studies is detected, which confirms our claim. As a result, we prove that the Bayesian attention Module's cross-attention mechanism is better suited in terms of NP's aim. We believe that significant research achievement has been demonstrated because theoretical claims and all experimental outcomes coincide.

---

> ### Author Response · Authors · 2021-11-14
> **Response to fqvy's review (1/5)**
>
> Thank you for thoughtful comments.
>
> ### Quick answers for the issued questions
>
> **Analysis of the learned attention weights** : The initial manuscript already contained attention weights between context and target data points as seen in Fig 1 and Fig 3.5 in the initial manuscript. In short, the attentive neural process(ANP) fails to capture contextual embeddings because the attention weights of all target points highlight on the lowest value or the maximum value in the context dataset. In the case of the Bootstrapping ANP, the quality of heat-map is slightly improved, but it still falls short of ours. Even in the noisy situation, the attention weights of target data points in ours still appear clearly.
>
> However, all heat-maps were too tiny to comprehend the meaning of axes, labels and ticks. We are revising the manuscript to increase the scale of each heat-map and add this analysis in the appendix. As a result, the heat-maps of Fig 1 and all heat-maps of Appendix G are adjusted in the revised manuscript to account for the similarity between the context and target dataset.
>
> **Regarding to the label $Y_c$ in the KL loss for the attention weight in the Eqn (6)** : As noted by the reviewer fqvy, the posterior distribution of unnormalized attention weight $\lambda$ is calculated using $x_i$ and $X_c$, whereas the prior distribution of unnormalized attention weight based on the key-based prior is calculated with $X_c$. We corrected the KL loss in the Eqn (6). This change is noted in the revised manuscript.
>
> **Novelty** : We summarize our contributions. The details are written in the following replies.
>
> - The main concern of this paper is to examine how neural processes (NPs) learn to acquire suitable contextual embeddings in noisy situations that have received little attention.
> - As stated by the reviewer uH8j and DEA4, we claim for the first time, using an information theory framework, that critical conditions for contextual embedding in NPs are independent of target features and close to contextual datasets as seen in Theorem 1.
> - Through extensive experiments, we identify facts that the proposed method consistently improves predictive performance as well as the quality of contextual embeddings, especially in noisy environments.
>
> **Additional experiments** : Additional experiments are being planned in response to the reviewer's suggestions. The experiment for which we are now planning is stated below.
>
> - The ConvCNP model will be evaluated in the same noisy situations like those discussed in this paper. (**Discussed**, See the 2nd response)
> - We evaluate the proposed method in the absence of the regularization term; KL loss for the attention weights in order to see whether this model supports the Theorem1. (**Discussed**, See the 2nd response)
> - We attempt to complete larger-sized images rather than those utilized in the meta-training for the image completion problem (**Discussed**, See the respond to fqvy's comment (2/2) written in 23th, Nov.)

---

> ### Author Response · Authors · 2021-11-18
> **2nd Response to fqvy's review (2/2)**
>
> ### Comparison between a family of *ConvCNP* and ours
>
> **Bold entries** indicates the best results. The detailed results are presented in the revised manuscript. We describe the location of tables in the revised manuscript in each table. The graphical explanation is also presented in Appendix E and F of the revised manuscript.
>
> (The tables below summarize the likelihood values)
>
> 1. 1D regressions in a noisy situations (Table 1.)
> |   Models  |     Context / RBF(noises)   |     Target / RBF(noises)  |     Context / RBF  |   Target / RBF  |     Context / Matern    |     Target / Matern    | Context / Periodic    |     Target / Periodic    |
> |:----:|:------------:|:------------:|:------------:|:------------:|:------------:|:------------:|:------------:|:------------:|
> | ConvCNP | 1.314 | -0.427 | 1.326 |0.084 | 1.319 |**-0.119** |1.358 |**-0.502** |
> | ConvNP | 0.873 | -0.469 | 0.729 |0.053 | 0.832 |-0.163 |0.953 |-0.522|
> | Ours| **1.374** | **-0.337** | **1.363** | **0.244** | **1.365** | -0.175 |**1.372** | -0.612 |
>
> 2. 1D regressions in a clean situations (Table E.4)
> |   Models  | Context / RBF  |   Target / RBF  |     Context / Matern    |     Target / Matern    | Context / Periodic    |     Target / Periodic    |
> |:----:|:------------:|:------------:|:------------:|:------------:|:------------:|:------------:|
> | ConvCNP | **1.351** | 0.891| **1.336** |-1.941 | **1.020** |-9.634 |
> | ConvNP | 1.318 | 0.905| 1.287 |-1.806 | 0.953 |-9.580 |
> | Ours| **1.343** | **0.937** | 1.170 | **-1.708** | 0.681 | -7.807 |
>
> 3. Sim2Real in a clean situation (Table 2)
> |   Models  | Context / simulation  |   Target / simulation  | Context / the hare-lynx    | Target / the hare-lynx    |
> |:----:|:------------:|:------------:|:------------:|:------------:|
> | ConvCNP | 2.509 | 2.139| 1.758 |**-0.431** |
> | ConvNP | 2.467 |2.124| 1.652 |-1.180 |
> | Ours| **2.711** | **2.297** | **2.429** | -1.766 |
>
> 4. Sim2Real in a noisy situation (Table F.5)
> |   Models  | Context / simulation  |   Target / simulation  | Context / the hare-lynx    | Target / the hare-lynx    |
> |:----:|:------------:|:------------:|:------------:|:------------:|
> | ConvCNP | 2.395 |1.679| 1.879 |-0.205|
> | ConvNP | 2.368 |1.687| 1.691 |-0.521|
> | Ours| **2.745** | **1.819** | **2.699** | **-0.076** |
>
>
> **Discussion of ConvCNP and ours** : Overall, both models perform reliably in all experiment cases. The ConvCNP is also a noise-resistant model as compared to other baselines. This might be explained by the fact that context embedding is carried out using a deterministic function and convolutional networks, which have a small number of parameters, are utilized.
> When investigating the experimental findings in detail, by comparing Table 1 and Table E.4, the behavior of the two models are similar in the clean dataset, when noise is present, however, different phenomena happen. Table 1 illustrate the performance of the proposed method in the presence of noises; except for the likelihood of the target dataset in the Matern and Periodic kernel, the proposed method outperforms remarkably. The Sim2Real problem exhibits the same result. This demonstrates that ConvCNP also uses a U-NET based encoder, implying that noise sensitivity cannot be fully address without regularization. As a result, it is obvious that a more trustworthy approach for paying attention to the context data is necessary when noises occur.
>
> **Limitations of ConvCNP** : The ConvCNP utilizes a functional mapping($\phi$) to construct feature maps similar to images. The official source code implements a functional representation using the RBF kernel function with augmented data points. (called *grid* : the number of points above 500 in 1D regression problems) While the majority of operations are carried out by convolutional neural networks, this model does not require a large number of parameters, nonetheless, obtaining feature maps requires heavy computation time, and there is difficulty with high-dimensional features. (e.g. $x \in \mathbb{R}^{44}$ in MovieLenz-100k; curse of dimensionality). While the *ConvCNP* enhances performance and robustness by using kernel functions, we believe this study is noteworthy for academia since the proposed method establishes that neural networks outperform alternatives that depend on kernel functions.

---

> ### Author Response · Authors · 2021-11-18
> **2nd Response to fqvy's review (1/2)**
>
> ### Quick answers for the issued questions (2)
>
> 1. **Performance comparison between ConvCNP and ours** : Both two models exceed established records by the baselines, While the ConvCNP perform similarly to the proposed method on a clean dataset, the proposed method outperforms the ConvCNP in noisy environments. The numerical evidence is provided in Table 1, Table 2 and Table E.4 in the revised manuscript
>
> 2. **Ablation studies for context embeddings and the proposed regularization** : The ablation study in Appendix H demonstrates how the proposed method encourages to improve context embeddings. If the model can not be trained, this regularization ensures that the models capture the similarity between the context and target dataset. Additionally, this regularization plays a role in ensuring that the learned embedding takes into account all context data points equally. We can meet the objectives of NPs;Task adaptation, because we can consider all data areas without regard for training biases.
>
> ---
>
> ### Ablation study for the proposed regularization
>
> **The effect of the proposed regularization for training NPs** : To facilitate the ablation study, 1D regressions with noisy situations was introduced. As seen in Fig H.11 to Fig H.13 of the revised manuscript, we tested the proposed models varying the hyper-parameter $K$ and the presence of the regularization $I(Z, x_i|D)$. In general, the learned context embeddings vary according to hyper-parameter $k$ and $I(Z, x_i|D)$; however, the majority of models achieves comparable performance as shown in Tab 1 because 1D regression does not require complicated context embeddings, whereas the models with $K=10$, $K=50$ and $K=70$ failed to train in intended manners. The noteworthy point is that while the model with $K=10, 50, 70$ can be adequately trained in the presence of $I(Z, x_i|D)$, it cannot be trained fully in the absence of $I(Z,x_i|D)$. As stated by the reviewer fqvy, maximizing only log-likelihood alone is inefficient in a noisy situation. Rather than that, as suggested, we encourage the models to be trained in an intended manner by maximizing likelihood and decreasing dependency between context embeddings and the target feature $x_i$. Not only does this regularization improve model's learnability, but it also has a positive effect in terms of quality of context embeddings on the models with high numbers of hyper-parameter $K \in \{100,300,500\}$.
>
> **Relationship between the proposed regularization and attention scores** : The proposed regularization encourages attention weights of each target point and retains similar scales and magnitudes. In our experiment setting, we identify that the occurrence of deviation in attention scores was derived from wrong trained models. For example, in noisy situations, the attention weights of most of the target points focus on the lowest or highest value in the context dataset as seen in Fig G.8. At this moment, the features are aligned in an ascending order. The highest attention score of each target point is listed as {0.88, 0.88, 0.75, 0.62, 0.5, 0.5}. However, the ANP trained on the clean dataset, has the {0.62, 0.5, 0.62, 0.62, 0.62} highest attention values of each target point. Thus, the best model, our models in a noisy situation, has the value of the highest score like {0.55, 0.45, 0.51, 0.54, 0.55, 0.65}. From this fact, we identify that the wrong models have been taught to be excessively interested in either feature regions with small numbers or large numbers, with the intermediate section omitted.
> When comparing the stochastic attention with and without the proposed regularization, this phenomena is readily apparent. The proposed method with $k = 40$ and without regularization in the RBF GP problem gets the maximum attention score, 0.64 and the lowest, 0.47. Meanwhile, the thing with regularization records the highest attention score, 0.52 and the minimum value, 0.43.
> This is an important aspect of the \textit{task adaptation}. The attention scores of all target points must not be influenced by feature magnitudes and their regions. Consequently, this method has to concentrate only on the similarity between features. The regularization aims to derive an appropriate attention mechanism that evenly weights across important regions and eliminates biases during the meta-training. It leads to achieving *task adaptation*, as it equally prioritizes context data points regardless of features magnitude.
> The detailed explanation and figures are presented in Appendix H of the revised manuscript.

---

### Official Review · Reviewer_TBTA · 2021-11-01

**Correctness:** 3
**Technical Novelty And Significance:** 3
**Empirical Novelty And Significance:** 3
**Recommendation:** 5
**Confidence:** 4

**Main Review:**

Strength: 1. The paper clearly indicates the idea of incorporating stochastic attention with key-based contextual prior into the neural processing.

Weakness: 1. My biggest concern for this paper is whether there is a novelty of the proposed method. Although I see the benefit of stochastic attention as it is proposed by Fan et al, this paper basically combines neural processing with stochastic attention. Then what is the main novelty here? It is simply A (neural processing) + B (stochastic attention) is better than A. Also the contextual prior, how to optimize and inference, etc are similar as the Fan et al 2020.

2. The information theory as well as the derivation of the equations in the main paper and appendix such as reparameterization, kl, and elbo are all covered in previous or early literature. It is not new.

3. As the key contextual prior is one of the important parts of this paper, it is also less ablation study to show the comparison between including the contextual prior and not including the contextual prior.

4. In the experiment section, the Movielens 10k dataset is a very real-world dataset as claimed by the author. However, the improvements on MovieLens-10k dataset are very marginal and even raise my concern about the practical benefits of this proposed method.


**Summary Of The Paper:**

Neural processes (NPs) aim to stochastically complete unseen data points based on a given context dataset. This paper incorporates the stochastic attention in NPs to capture the context information. We empirically show that our approach on 1D regression, predator-prey model, image completion, and MovieLens-10k dataset.


**Summary Of The Review:**

Please see the details in my main review. In summary, I think this paper clearly indicates their ideas of incorporating stochastic attention into neural processing. However, the novelty as well the comprehensiveness of the study and experiments need more elaborations and works. Thus, I recommended marginally below the acceptance threshold.


----------------------------------Post rebuttal-----------------------------------------------------------------------------------------------------------------
After carefully reading the authors' response, I am still holding the question regarding the novelty of this paper. The theoretical proof in this paper is very limited and basic. And most of them including the contextual prior, stochastic attention, elbo, etc have been fully covered by the bayesian attention modules [Fan et al 2020] paper. Thus I don't agree with the author about the theoretical proof. Thus, this paper can only be seen as a simple implementation of the bayesian attention module in the NP field.
In addition, although I see new results in the rebuttal, my concern is that the accepted version of the paper might not be a substantially improved version of the paper. The rebuttal stage is not similar to a journal revision where you can submit an improved version of your paper. Rather, it should be used to clarify misunderstandings.
Overall, I would like to hold my original score for this paper and still think this paper is not ready for the ICLR conference.

---

> ### Author Response · Authors · 2021-11-14
> **Response to TBTA's review (3/3)**
>
> ### Experiments (cont')
>
> **Analysis of the learned attention weights** : We identify that the proposed method outperforms the baselines in terms of quality of context embeddings. To substantiate this fact, the empirical evidence is presented in Fig 1 and Appendix G in the revised manuscript. The attention mechanism can explicitly display the distance between context and target data points via attention weights. Especially, when the dimension of features is one, $x \in \mathbb{R}^1$, the distance between features is simply calculated and sorted, allowing for straightforward comprehension using the attention heat-maps. We already contain attention weights between context and target data points as seen in Fig 1(a) and Fig E.5 in the initial manuscript, but all heat-maps are too tiny to comprehend the meaning of axes, labels and ticks. We are revising the manuscript to increase the scale of each heat-map and add this analysis in the Appendix G. Additionally, we will provide a simplified heat-map that takes into account only target points with the same value in the context dataset.
>
> As a result, all figures in Appendix G in the revised manuscript compare baseline and ours in both clean and noisy 1D regression datasets. The x-axis in these graphs represents the value of features in the context dataset, while the y-axis represents the value of features in the target dataset. The best pattern for this heat-map is diagonal because all feature values are arranged in ascending order.
>
> Fig G.8 in the revised manuscript tells us that all methods exhibit identical patterns like the ideal case when trained on a clean dataset; nevertheless, when all models are trained on a noisy dataset, we see distinct phenomena as illustrated in Fig G.8. The attentive neural process(ANP) fails to capture contextual embeddings because the attention weights of all target points highlight on the lowest value or the maximum value in the context dataset. In the case of the Bootstrapping ANP, the quality of heat-map is slightly improved, but it still falls short of ours. Even in the noisy situation, the attention weights of target data points in ours still appear clearly.
>
> When comparing our model's heat-map pattern in clean and noisy environments, there is a noteworthy point. Our model is capable of learning adaptively how to focus on certain context data points depending on the extent of dirty data in the meta-train dataset. The model is trained on the clean dataset to take into consideration nearby points as well as the corresponding point in the context dataset, hence, the heat-map gradually changes. This is because the clean dataset has smooth values, $y_{i,j}$, near a certain feature $x_{i,j}$. Meanwhile, in the noisy dataset, the model is trained to focus exclusively on corresponding points to the context dataset. Hence, the heat-map in Fig G.9 in the revised manuscript indicates that the attention score of target data points that includes the context dataset has a high value, whilst the remaining points treat all context data points uniformly. This phenomenon occurs because there is less correlation between adjacent features $x_{i,j}$ and its labels $y_{i,j}$ in the noisy situation during the meta-training.
>
> **[Reference]**
> 1. Lee, Juho, et al. "Bootstrapping neural processes.", NeurIPS2020
> 2. Volpp, Michael, et al. "Bayesian Context Aggregation for Neural Processes.", ICLR2021
> 3. Yao, Huaxiu, et al. "Improving generalization in meta-learning via task augmentation." ICML2021
> 4. Yin, Mingzhang, et al. "Meta-learning without memorization." ICLR2020
> 5. Rajendran, et al. "Meta-learning requires meta-augmentation." NeurIPS2020
> 6. Fan, Xinjie, et al. "Bayesian attention modules." NeurIPS2020
> 7. Gordon, Jonathan, et al. "Convolutional conditional neural processes." ICLR2020
> 8. Foong, Andrew YK, et al. "Meta-learning stationary stochastic process prediction with convolutional neural processes." NeurIPS2020
> 9. Han, Soyeon Caren, et al. "GLocal-K: Global and Local Kernels for Recommender Systems." Proceedings of the 30th ACM International Conference on Information & Knowledge Management. 2021.

---

> ### Author Response · Authors · 2021-11-14
> **Response to TBTA's review (2/3)**
>
> ### Novelty (cont')
>
> **New metrics for NPs to be followed for proper context embedding via information theory** : As the reviewer uH8j and DEA4 stated as *pros*, we claim for the first time, using an information theory framework, that critical conditions for contextual embedding in NPs are independent of target features and close to contextual datasets as seen in Theorem 1. Previous studies[3,4,5] have dealt with information theory setups to reveal memorization phenomena under limited task sizes in the meta-training, meanwhile, nobody has investigated the points at which NPs retain their performance and quality of contextual embedding in severe situations. From this perspective, this work has a significant trajectory in NP domains.
>
> We theoretically and empirically prove that the proposed algorithm supports this statement. We demonstrate that the Bayesian Attention Module Address with key based prior[6] addresses this issue by leveraging amortized variational inference for attention weights in order to encourage NPs to pay more attention to the context dataset. Therefore, we empirically identify facts that the proposed method consistently improves predictive performance as well as the quality of contextual embeddings, especially in noisy environments.
>
> ### Experiments
>
> **The result of MovieLenz-10k** : The difference in RMSE scores between ANP and ours is 0.014 in Table 3. Due to the fact that the MovieLenz-10k is a regression problem, there is no standardized metric like accuracy in classification tasks. Rather than that, when we examine the rating values of MovieLenz-10k, we observe that they vary from 0 to 5, indicating that the RMSE score cannot have big values. Hence, recommendation systems that use the MovieLenz-10k dataset regard 0.001 improvements of the RMSE score as a noteworthy contribution. This fact was summarized by Han et al.[9]. We suggest that the discrepancy of 0.014 RMSE scores between ANP and ours is considered significant. This tendency is readily apparent when looking at Table H.5 in the revised manuscript. Except for the vanilla ANP, all other models showed a significant drop in performance, but the proposed model resulted in substantial gains in performance and outperformed all models including the vanilla ANP. These results are identical to those obtained in the preceding case of intentionally injecting noise to 1D regression and the Sim2Real problem. We believe that the proposed method is advantageous in the practical problem.
>
> **About role of the key-based prior** : To further understand the function of key-based prior, we suggest an information dropout model for the ANP. This approach makes use of a drop-out mask derived from amortized variational inference to a local presentation, $r_i$ At this point, the prior distribution employed is the standard normal distribution $N(0,1)$. The majority of experimental data show that this model does not improve performance except for the Sim2Real problem. From the perspective of quality of context embeddings, we also recognize that the attention with information drop-out is unable to properly capture the similarity between the context and target dataset. The next reply, **Analysis of the learned attention weights**, includes a full explanation about the quality of context embeddings. Additionally, It is stated in the Appendix G of the revised manuscript.
>
> As we state the response for the reviewer fqvy, we evaluate the proposed method in the absence of the regularization term; KL loss for the attention weights in order to see whether this model supports the Theorem1. These experimental results are included in the Appendix H.

---

> ### Author Response · Authors · 2021-11-14
> **Response to TBTA's review (1/3)**
>
> Thank you for thoughtful comments.
>
> ### Novelty
>
> **Difference from previous works, the Bayesian Attention Modules[6]** : The Bayesian Attention Modules examines only the role of the key-based prior via empirical experiments. The experiments of graph embeddings and language models show improvement in comparison to traditional attention. Regrettably, there is no theoretical justification about the role of key based prior.
>
> On the other hand, this paper establishes that the loss function and all components are constructed to fulfill Theorem 1. In addition, several heat-maps of associated attention weights in Appendix G and H indicate an increase in the quality of contextual embeddings, which confirms our claim. As a result, we prove that the Bayesian attention Module's cross-attention mechanism is better suited in terms of NP's aim. We believe that meaningful research achievement is when theoretical claims and all experimental outcomes coincide.
>
> **Main academic concern about Neural Processes** : The significance of NPs research is that the meaning of each component is explained in terms of the model's purpose, meanwhile, implementations of existing neural processes, including ours, are straightforward. The ANP shows why attention is required to capture context information in a clean dataset, and ConvCNP[7] and ConvNP[8] explain why a convolutional network is required for domain shift due to their translation invariant features. From this point, this study makes an adequate academic contribution. This is because we propose how context embedding should prioritize larger context datasets in order to preserve embedding quality in noisy environments.
>
> **Noisy environment impairs the embedding of context dataset in existing neural processes** : As the reviewer fqvy mentioned as an interesting research direction, the main concern of this paper is to examine how neural processes (NPs) learn to acquire suitable contextual embeddings in noisy situations that have received little attention. The preceding study, the Bootstrapping Neural Process[1], deals with training models on a clean dataset and testing them on a dirty dataset to demonstrate their model's robustness. However, no one has investigated whether NPs can be trained to acquire suitable contextual embeddings given dirty datasets. We discover through experimental results that existing NPs suffer from meaningless context embedding, hence, all existing methods do not perform well prediction in noisy situations. While modern approaches such as Bayesian Aggregation[2] and Bootstrapping[1] have marginally improved performance in 1D regression and Sim2Real experiments, these gains are not consistent across all experimental cases, and these cannot, therefore, be considered solutions.
>
> - The conventional attention mechanism's behavior in noisy environments.
>
> To improve embedding capacity, NPs incorporate an attention mechanism (called ANP). The attention mechanism is frequently used in large datasets such as natural language processing and computer vision. Regardless of how noisy a dataset is, irreducible noises may be ignored due to the law of large numbers. The authors of the ANP demonstrate significant improvement in 1D regression and image completion, such that this approach can be considered as the baseline of current NP studies. However, we notice that the vanilla attention does not perform as intended in few-shot and even worse in noisy conditions. Additionally, because the ANP employs a deterministic function attention mechanism as a part of context embedding, it deviates from the principle of amortized variational inference. As a result of these considerations, we believe that the ANP still falls short of the goal of NPs, Task adaptation. Therefore, we seek to conceptually imply, using information theory, that NPs satisfy the aim even in the few-shot regime. According to the above statement, we design a novel method that adheres to the principle of amortized variational inference without sacrificing the performance of context embeddings by introducing the Bayesian Attention Module with key based prior. The novelty is noteworthy that we theoretically establish for the first time the meaning of key based prior in terms of the purpose of NPs for the first time.
>
> This paper includes Figure 1 and Appendix G in the revised manuscript to substantiate this statement. The attention heat-maps for 1D regression and Sim2Real (the hare-lynx dataset) should ideally be a diagonal pattern because the provided features, $x_i \in \mathbb{R}^1$ are ascendingly ordered in both context and target dataset. While heat maps of the existing attentive neural processes are chaotic (including ANP with information dropout that uses amortized variational inference to the local variables, $r_{i}$ with the standard normal distribution $N(0,1)$ as the prior distribution), the proposed method appears to have a diagonal pattern, indicating that it is far better suited to the ideal case.

---

> ### Author Response · Authors · 2021-11-18
> **2nd Response to TBTA's review**
>
> ### Quick answers for the issued questions
>
> 1. **Comparison between including the contextual prior and not including the contextual prior** : First, if the model cannnot be trained (in this paper $k=50$), this regularization ensures that the models capture similarity between the context and target dataset. Second, this regularization plays a role in ensuring that the learned embedding takes into account all context data points equally. We can meet the objectives of NPs;Task adaptation, because we can consider all data areas without regard for training biases. The detailed explanation is shown in Appendix H and G of the revised manuscript.
>
> ---
>
> ### Ablation study for key contextual prior
>
> **The effect of the proposed regularization for training NPs** : To facilitate the ablation study, 1D regressions with noisy situations were introduced. As seen in Fig H.12 and Fig H.15 of the revised manuscript, we tested the proposed models varying the hyper-parameter $K$ and the presence of the regularization $I(Z, x_i|D)$. We recognize that the learned context embeddings vary according to hyper-parameter $k$ and $I(Z, x_i|D)$; however, the majority of models achieves comparable performance because 1D regressions do not require precise context embeddings, whereas the models with $K=10$, $K=50$ and $K=70$ failed. The noteworthy point is that while the model with $K=10, 50, 70$ can be adequately trained in the presence of $I(Z, x_i|D)$, it cannot be trained fully in the absence of $I(Z,x_i|D)$. Maximizing only log-likelihood alone is inefficient in a noisy situation. Rather than that, as suggested, we encourage the models to be trained in an intended manner by maximizing  likelihood values and decreasing dependency between context embeddings and the target feature $x_i$. Not only does this regularization improve model's learnability, but it also has a positive effect in terms of quality of context embeddings on the models with high numbers of hyper-parameter $K \in \{100,300,500\}$.
>
> **Relationship between the proposed regularization and attention scores** : The proposed regularization encourages attention weights of each target point and retains similar scales and magnitudes. In our experiment setting, we identified that the occurrence of deviation in attention scores was derived from wrong trained models. For example, in noisy situations, the attention weights of most of the target points focus on the lowest or highest value in the context dataset as seen in Fig G.8. At this moment, the features are aligned in ascending order. The highest attention score of each target point is listed as {0.88, 0.88, 0.75, 0.62, 0.5, 0.5}. However, the ANP trained on the clean dataset, has the {0.62, 0.5, 0.62, 0.62, 0.62} highest attention values of each target point. Thus, the best model, our model, in a noisy situation, has the value of the highest score like {0.55, 0.45, 0.51, 0.54, 0.55, 0.65}. From this fact, we identify that the wrong models have been taught to be excessively interested in either feature regions with small numbers or large numbers, with the intermediate section omitted.
> When comparing the stochastic attention with and without the proposed regularization, this phenomenon is readily apparent. The proposed method with $k = 40$ and without regularization in the RBF GP problem gets the maximum attention score, 0.64 and the lowest, 0.47. Meanwhile, the thing with regularization records the highest attention score, 0.52 and the minimum value, 0.43.
> This is an important aspect of the \textit{task adaptation}. The attention scores of all target points must not be influenced by feature magnitudes and their regions. Consequently, this method accomplishes to concentrate only on the similarity between features. The regularization aims to derive an appropriate attention mechanism that evenly weights across important regions and eliminates biases during the meta-training. It leads to task adaptation, as it equally prioritizes context data points regardless of features magnitude. These experimental results coincide the Matern GP and the Periodic GP.
>
> As a result, unlike the bayesian attention module[6], we analyze the key based conceptual prior from the perspective of NPs' purpose.
> The detailed explanation and figures are presented in Appendix H of the revised manuscript.

---

> ### Author Response · Authors · 2021-11-30
> **Response to TBTA's post rebuttal comment**
>
> We appreciate your time and effort in reviewing our paper, as well as of your constructive comments.
>
> To begin, we are relieved that the issues mentioned by reviewer TBTA about the experiment results appears to be resolved throughout the rebuttal procedure; comparison adding the contextual prior and excluding the contextual prior, as well as experimental accomplishment on the Movielens-100k dataset.
>
> Along with our earlier responses, we would like to include a description of our discussions with other reviewers to substantiate our claims and resolve the remaining questions.
>
> ### Novelty
>
> We demonstrate that when the models are trained on noisy situations, the typical neural processes degrade the quality of context embeddings. To address this issue, we provide a straightforward yet novel method for excelling at context embeddings by presenting both a theoretical claim and empirical proofs highlighting the importance of context embeddings. As stated by reviewer fqvy, this direction is fascinating and contributes to the practical use of neural processes.
>
> ### Theoretical analysis
>
> - Information theory in this study
>
> In this work, we provided an ideal strategy for context embedding in neural processes from the prospective of the information theory. Since there was rarely previous study in the field of neural processes that used information theory to analyze context embedding. as reviewer DEA4 and uH8j mentioned, this analysis helps readers grasp the advantages of the proposed method, which enhances their comprehension of the field of neural processes and opens the way for future works.
>
> - Amortized variational inference and stochastic attention
>
> The majority of neural processes incorporate the $ELBO$ loss, resulting in similar loss functions. Thus, utilizing the reparameterization trick, the bayesian attention modules also included amortized variational inference into the attention mechanism[1]. This leads in a loss function that is comparable to the majority of neural processes. Rather than that, what matters is to understand why proposed loss function are necessary and which elements enhance performance. Both appendix H in the revised manuscript and responses to reviewer fqvy's comments contain the evidences.
>
> Furthermore, the bayesian attention modules did not attempt to analyze deeply the influence that the contextual prior had on attention weights[1]. This paper was to examine empirical improvement on natural language processing and graph embeddings. On the other hand, our paper demonstrates how the key contextual prior regularizes the latent variable to pay more attention to the context dataset and then maintain similar attention weights even when a novel task is introduced. In the appendix, we provide evidences in the form of ablation studies. Consequently, we demonstrate that this regime is better suited for *task adaptation*, which is the purpose of neural processes.
>
> ### Implementation
>
> As noted by reviewer fqvy and uH8j and the responses to reviewer fqvy's comments, while our method is straightforward to implement and dose not need complicated network architectures or data augmentation, it delivers equivalent results to ConvCNP on the clean datasets, which uses functional representations (with RBF kernel function) and auxiliary data. In addition, the proposed method outperforms all baselines in noisy situations. We believe this is significant advantage of this study.
>
> ### The revised manuscript
>
> As shown in the revised manuscript, the majority of the updated points are placed in the appendix to provide ablation studies for clarifying misunderstandings throughout the rebuttal procedure. In response to reviewer DEA4, a few sentences in the introduction have been corrected, and an explanation of Figure 1 has been added. The experimental results of ConvCNP and ConvNP are additionally provided in Table 1,2. The main text has remained mostly unchanged. We believe that this modification might be viewed as a necessary step in the rebuttal procedure.
>
> Please let me know if there is anything you need to clarify. Because the final discussion period is coming to a close, we are going to try our best to rapidly respond if you have remaining concerns. Nonetheless, we would like to express our appreciation for your comments during the rebuttal period.
>
> [Reference]
> 1. Fan, Xinjie, et al. "Bayesian attention modules." NeurIPS2020

---

### Official Review · Reviewer_DEA4 · 2021-11-02

**Correctness:** 4
**Technical Novelty And Significance:** 2
**Empirical Novelty And Significance:** 3
**Recommendation:** 6
**Confidence:** 2

**Main Review:**

**Pros:**

- Even if, to me, the technical contribution is not major - integrating bayesian attention in the attention neural process - I enjoy the effort made to explained the advantages brought by the method in an information theory setup. This improves the understanding of the field and open up the path towards further directions.
- The experimental part show consistent improvement over multiple setups and several datasets

**Cons:**

- The paper mention that the model does not require exhausted computations. Can you also report the comparison between the proposed methods and the other baselines presented in the paper in terms of time efficiency and the number of parameters?
- For the experiments in Movielens-100K dataset, isn't it possible to integrate the graph structure in the NP approach for a fair and higher performant model?
- In Figure 1 a) it is not clear what it is depicted in the second row. I guess it represents the attention weights similar to the one in the supp material, but it should be clearly specified in the caption.

**Summary Of The Paper:**

This paper proposes a neural process enhanced with stochastic attention to focus more on the context dataset. The method replace the classical attention used in ANP with the Bayesian attention module, showing that this design choice improves the performance also in noisy scenarios or when target datasets mismatched (by changing the kernel used to generate the target dataset). Moreover, the paper offers an interpretation of the method from an information theory perspective, proving that the NP with stochastic attention can be seen as a regularization of the latent space such that it pays more attention to the context dataset. The method is tested on both synthetic and real-world datasets and improves the scores especially in noisy or more complicated scenarios.

**Summary Of The Review:**

To me, the paper seems interesting not necessarily because of the technical novelty, but because of the theoretical side: the information theory based explanation of the final loss. However, I want to mention that I do not have much experience on this particular field so it might be the case that I missed some similar prior work.

---

> ### Author Response · Authors · 2021-11-18
> **Response to DEA4's review**
>
> ### Comparison of computation
>
> **The number of parameters** : The initial manuscript already contained the number of parameters for all baselines including ours in Table D.2. For your convenience, we provide the simplified table in this response.
>
> | Models  |     1D regressions   |     Sim2Real  |     MovieLenz-100k  |     Image Completion  |
> |:----:|:------------:|:------------:|:------------:|:------------:|
> | CNP | 99,842 | 100,228 | 110,850 | 1,583,110 |
> | NP | 116,354 | 116,740 | 127,362 | 1,845,766 |
> | ConvCNP | 50,612 | 50,655 | N/A | N/A |
> | ANP | 595,714 | 596,228 | 617,730 | 9,466,886 |
> | Bootstrapping ANP | 628,610 | 629,124 | 650,626 | 9,991,686 |
> | Ours | 597,015 | 597,529 | 619,031 | 9,472,027 |
>
> - In terms of the number of parameters, ours is not considerably different from the ANP.
>
> **Inference time required for all baselines and ours** : The revised manuscript added a comparison of inference time between baselines and ours. The detailed information is presented in Table D.3, and this response contains a condensed version of the table. For one epoch, we track the inference time for all baselines. This measurement is carried out in the same settings.
>
> | Models  |     1D regressions   |     Sim2Real  |     MovieLenz-100k  | Image Completion (contexts: 100)  |Image Completion (contexts: 500) |
> |:----:|:------------:|:------------:|:------------:|:------------:|:------------:|
> | CNP | 0.86 | 0.896 | 0.829 | 1.321 | 1.390 |
> | NP | 1.411 | 1.422 | 1.356 | 1.990 | 2.081 |
> | ConvCNP | 2.788 | 4.128 | N/A | N/A |N/A |
> | ANP | 2.442 | 2.5 | 2.222 | 5.710 | 11.740 |
> | Bootstrapping ANP | 6.747 | 6.968 | 6.250 | 26.186 | 88.332 |
> | Ours | 3.15 | 5.848 | 3.121 | 11.350 | 95.031 |
> Unit : second
>
> - When the amount of context data points is limited, the difference between ANP and ours is insignificant. However, when the number of contexts grows, we discover that Image completions need more computation, especially in the case of the number of context data points is 500.
> - Multiple normalization operations and un-optimized functions such as a log-gamma function consume additional computation resources.
> We fix this claim as below.
> > “Particularly, this method significantly enhances performance without additional architectures and data augmentation compared to the attentive neural processes.”
>
>
>
> ### NPs for graph structures : new research direction
> **'Task adaptation'** -  **'recommendation system'** : We guess that the objectives of recommendation systems using graph embedding and NPs are slightly different at certain moments. The neural processes(NPs), one of the meta-learning algorithms, is proposed to improve generalization performance using a given context dataset, called Task adaptation. The purpose of NP is to use embedding $Z$ to derive an identifier suitable for a novel task. Meanwhile, the graph networks currently utilized in recommendation systems are intended to provide a suitable mapping for representing relationships between customers and other customers, as well as customers and items without loss of information. Because the MovieLenz-100k dataset has a relatively large number of features $x \in \mathbb{R}^{44}$, it demonstrates that task adaptation is capable of replacing the graph networks if small numbers of data points are provided as a context dataset. This situation is not universal among recommendation systems.
>
> Instead, we anticipate that NPs can be applied to graph generators when local subsets of the graph are visible, while the remainder is concealed. This is future work. We are grateful for you to suggest a new research direction involving the integration of NPs and graph structures.
>
> ### Clarify figures in the manuscript
>
> In the revised manuscript, we increased the size of all figures. We recognized that the reader might miss our academic contribution under the initial manuscript because all heat-maps are too tiny to comprehend the meaning of axes, labels and ticks. Additionally, we provide a simplified heat-map that takes into account only target points with the same value in the context dataset. Due to the fact that the original version has one-hundred target points, labels and ticks were necessarily small. By minimizing the number of target points, we are able to present the simplified version. We believe that the readers will instantly grasp what we convey by exhibiting a more distinct diagonal pattern. Plus, we will provide Sim2Real's attention heat-maps in both clean and noisy environments. As a result, our discovery appears to be consistent.
>
> Regarding to Fig 1 : We revised Fig 1 and added some comments as a caption as below.
> > (b) The horizontal axis indicates the value of features in the context dataset, while the vertical axis indicates the value of features in the target dataset in these heatmaps. The best pattern for this heat-map is diagonal because all feature values are arranged in ascending order. Among all models, ours comes closest to the ideal.

---

> > ### Comment · Reviewer_DEA4 · 2021-11-22
> > **Response to authors rebutal**
> >
> > I would like to thank the authors for the clarifications. The paper seems interesting in the current form and the authors put a lot of effort in responding to reviewers concerns. Taking these into consideration, I maintain my acceptance score (score 6-7).
> >
> > However I want to mention that I don’t have much experience on the NP side, so there is a chance that I might not be able to measure the novelty brought by this paper accurately.

---

> > > ### Author Response · Authors · 2021-11-22
> > > **Response to DEA4's comment**
> > >
> > > We sincerely appreciate that you acknowledge our effort in revising the manuscript and responding raised concerns. In particular, we feel thankful of your recognition that this work is valuable to be published in this venue.

---

### Official Review · Reviewer_uH8j · 2021-11-08

**Correctness:** 3
**Technical Novelty And Significance:** 3
**Empirical Novelty And Significance:** 2
**Recommendation:** 6
**Confidence:** 2

**Main Review:**

### Strengths
- The application of Bayesian attention to neural processes seems novel.
- The authors provide an information-theoretic perspective of the proposed method.

### Weaknesses
- The authors claim that conventional NPs are sensitive to noise, but this claim is not well-supported.
- Being sensitive to noise is caused by multiple factors, such as the model's flexibility, the amount of data, the number of training steps, etc. With enough data, therefore, larger sensitivity can result in better performance.
- I think claiming "superior" performance based on the reconstruction error is like claiming VAE is better than autoencoders in terms of the reconstruction error.

**Summary Of The Paper:**

This paper proposes to improve attentive neural processes (ANP; Kim et al.) by replacing deterministic attention with a Bayesian attention module.
Since the inference requires variational approximation, I think the proposed method is a variational counterpart of ANP, similar to the relation between VAEs and autoencoders.

**Summary Of The Review:**

I think the main idea is novel and worth publishing.
However, some of the claims are not grounded properly and should be toned down.

---

> ### Author Response · Authors · 2021-11-18
> **Response to uH8j's review (2/2)**
>
> ### Experiment settings for fair comparison to identify the sensitivity to noises.
>
> **Experiment setup** : To prevent unfair comparison, we provide adequate training steps to 200,000 and used identical datasets for all models. Although the meta-learning framework involves random sampling to create the context and target dataset, and some deviation from each model is unavoidable, it does not make a significant impact. All experimental setups are described in Appendix E, F, I and J.
>
> **Model architecture** : In this work, all baselines (with the exception of ConvCNP) and ours used the same decoder networks as an identifier for a novel task. The distinction is in the manner in which context embeddings are performed. The vanilla neural processes(NP), which are composed of basic MLP structures, are limited in their ability to capture context embeddings effectively. To improve embedding capacity, the ANP incorporates an attention mechanism despite the extra numbers of parameters required. This mechanism is frequently used in large datasets such as natural language processing and computer vision. In these domains, regardless of how noisy a dataset is, irreducible noises may be ignored due to the law of large numbers. Although the ANP demonstrates significant improvement in 1D regression and image completion, such that this approach can be considered as the baseline of current NP studies, we notice that the vanilla attention does not perform as intended in few-shot and even noisy conditions.
>
> ### Ablation study for the proposed regularization
>
> **The effect of the proposed regularization for training NPs** : To facilitate the ablation study, 1D regressions with noisy situations was introduced. As seen in Fig H.12 and Fig H.15 of the revised manuscript, we tested the proposed models varying the hyper-parameter $K$ and the presence of the regularization $I(Z, x_i|D)$. The learned context embeddings vary according to hyper-parameter $k$ and $I(Z, x_i|D)$; however, the majority of models achieves comparable performance because 1D regressions do not require precise context embeddings, whereas the models with $K=10$, $K=50$ and $K=70$ failed to train in intended manners. The noteworthy point is while the model with $K=10, 50, 70$ can be adequately trained in the presence of $I(Z, x_i|D)$, it cannot be trained fully in the absence of $I(Z,x_i|D)$. Maximizing only log-likelihood alone is inefficient in a noisy situation. Instead, as suggested, we encourage the models to be trained in the intended manner by maximizing likelihood and decreasing dependency between context embeddings and the target feature $x_i$. Not only does this regularization improve model's learnability, but it also has a positive effect in terms of quality of context embeddings on the models with high numbers of hyper-parameter $K \in \{100,300,500\}$.
>
> **Relationship between the proposed regularization and attention scores** : The proposed regularization encourages attention weights of each target point and retains similar scales and magnitudes. In our experiment setting, we identified that the occurrence of deviation in attention scores was derived from wrong trained models. For example, in noisy situations, the attention weights of the ANP focus on the lowest or highest value in the context dataset as seen in Fig G.8. The highest attention score of each target point is listed as {0.88, 0.88, 0.75, 0.62, 0.5, 0.5}. However, the ANP trained on the clean dataset, has the {0.62, 0.5, 0.62, 0.62, 0.62} highest attention values of each target point. Thus, the best model, our model, has the value of the highest score like {0.55, 0.45, 0.51, 0.54, 0.55, 0.65}. From this fact, we identify that the wrong models have been taught to be excessively interested in either feature regions with small numbers or large numbers, with the intermediate section omitted.
> When comparing stochastic attention with and without the proposed regularization, this phenomenon is readily apparent. The proposed method with $k = 40$ and without regularization in the RBF GP problem gets the maximum attention score, 0.64 and the lowest, 0.47. Meanwhile, the thing with regularization records the highest attention score, 0.52 and the minimum value, 0.43.
> This is an important aspect of the \textit{task adaptation}. The attention scores of all target points must not be influenced by feature magnitudes and their regions. Consequently, this method has to concentrate only on the similarity between features. The regularization aims to derive an appropriate attention mechanism that evenly weights across important regions and eliminates biases during the meta-training. It leads to achieving *task adaptation*, as it equally prioritizes context data points regardless of features magnitude.
> The detailed explanation is presented in Appendix H of the revised manuscript.

---

> ### Author Response · Authors · 2021-11-18
> **Response to uH8j's review (1/2)**
>
> Thank you for constructive comments.
>
> ### Quick answers for the issued questions
>
> 1. **The conventional NPs are sensitive to noises** : The findings of experiments in the manuscript and ablation studies in Appendix G indicate that the quality of context embeddings is critical for NP performance maintenance. We illustrate that the noises impair the quality of context embeddings in Appendix G
>
> 2. **Ablation studies for context embeddings and the proposed regularization (Others than completion performance)** : The ablation study in Appendix H demonstrates how the proposed method encourages to improve context embeddings. If the model cannnot be trained, this regularization ensures that the models capture the similarity between the context and target dataset. Additionally, this regularization plays a role in ensuring that the learned embedding takes into account all context data points equally. We can meet the objectives of NPs;Task adaptation, because we can consider all data areas without regard for training biases.
>
> ---
>
> ### Changes to the initial manuscript
>
> To compare the quality of context embeddings, the initial manuscript presented attention weights between context and target data points as seen in Fig 1 and Fig E.5 in the initial manuscript. However, all heat-maps of attention weights were too tiny to interpret the meaning of axes, labels and ticks. So, we revised the manuscript to increase the scale of each heat-map in Fig 1 and all figures in Appendix. In detail, Appendix G and H in the revised manuscript include empirical evidence and detailed explanation that the proposed method properly captures the similarity between the context and target dataset when trained on noisy situations. The detailed responses are as follows.
>
>
> ### Noise situation impairs context embeddings in conventional NPs
>
> **Role of context embeddings** : The neural processes(NPs), one of the meta-learning algorithms, is proposed to improve generalization performance using a given context dataset, called Task adaptation. The objective of NPs is to extract an identifier suited for a novel task by utilizing context embeddings $Z$. NPs can be trained in few-shot regimes. Individual data has a significant influence on this scenario, and also has an effect on the learning process.
>
> **Performance is dependent on the quality of context embeddings** : When clean datasets are utilized, the CNP model is capable of describing the full function values even if a small number of points are provided as a context dataset like Fig E.4. The mean values of prediction are very close to the ground truth.
> On the other hand, due to noises, if the models are not trained to capture context information, the predictive mean value does not only differ from a ground truth in Fig E.5. and the Sim2Real problem, but also they predict vaguely despite the presence of many context data points, as seen in Fig E.6 and 7. It does not appear to utilize context embeddings to predict associated target values in a novel task.
>
> **Quality of context embeddings in noisy environments** : If NPs employ an attention mechanism to enhance performance(ANP), attention weights can be used to explicitly represent quality of the context embeddings. The attention heat-maps for 1D regression and Sim2Real (the hare-lynx dataset) should ideally be a diagonal pattern because the provided features, $x_i \in \mathbb{R}^1$ are ascendingly ordered in both context and target dataset. When all models are trained on the clean dataset, the magnitude of attention score varies, but all models exhibit the diagonal patterns seen in Appendix G.7 and the left one of Fig G.9. However, in Fig G.8 and the right one of Fig.G9, the baselines trained on noisy situations do not contain significant context embeddings, even when the target and context data points are identical. Meanwhile, the proposed method retains a diagonal pattern, indicating that it is far better suited to the ideal case. The clear figures are provided in Appendix G of the revised manuscript.

---

### Author Response · Authors · 2021-11-21
**General Response**

Thank you all for helpful suggestions.

We thank all the reviewers for their thorough and helpful comments. If any of our responses to individual reviewers below is insufficient, please feel free to ask any additional questions.

---
### **List of concerns we addressed in the new manuscript**

- Ablation study of context embeddings → Appendix G and H (reviewer uH8j, TBTA,  fqvy)
- The role of *paying more attention to the context dataset* → Appendix H (reviewer fqvy, TBTA)
- Comparison of a family of ConvCNP → Table 1,2,E.4,F.5  (reviewer fqvy)
- Comparison of computation between baselines → Table D.2 and D3 (reviewer DEA4)
- Noise situations impair the context embeddings in NPs → Appendix G  (reviewer uH8j, TBTA)
- Increase figure sizes → all figures in revised manuscript  (reviewer DEA4, fqvy)
- Large size images completed by ours  → Appendix J (reviewer fqvy)

### **Discussion on reviews**
- MovieLenz-10k results → first response("The result of MovieLenz-10k")  (reviewer TBTA)
- Novelty of this work → first response  (reviewer fqvy, TBTA)
- Background → first response (reviewer fqvy)

### **Minors**
- Fix typos of eq.6  (reviewer fqvy)
- Revised the caption of Fig 1.  (reviewer DEA4)

---

### Author Response · Authors · 2021-11-21
**The end of the discussion phase is approaching**

Dear Reviewers and Area Chair,

Could you please go over our responses and the revision since we can have interactions with you only by this Monday (22nd)? We have responded to your comments and faithfully reflected them in the revision. We sincerely thank you for your time and efforts in reviewing our paper, and your insightful and constructive comments.

Thanks, Authors

---

### Decision · Program_Chairs · 2022-01-20

**Decision:**

Accept (Poster)

**Comment:**

This work receives mostly positive rates. Most reviewers agree that the use of Bayesian attention to neural processes is novel, and its interpretation is interesting. Since the reviewer TBTA requests a substantial revision of the submission and fortunately authors’ feedback is thoroughly satisfactory, we highly recommend the authors to prepare for a significantly improved camera-ready version that clarifies most of reviewers’ concerns.